 SciPost Phys. Lect. Notes 54 (2022)

# Fermionic Gaussian states:
# An introduction to numerical approaches

## Jacopo Surace[1*] and Luca Tagliacozzo[2]

**1** ICFO-Institut de Ciencies Fotoniques, The Barcelona Institute of Science and Technology, Castelldefels (Barcelona), 08860, Spain
**2** Departament de Física Quàntica i Astrofísica and Institut de Ciències del Cosmos (ICCUB), Universitat de Barcelona, Martí i Franquès 1, 08028 Barcelona, Catalonia, Spain

* jacopo.surace@icfo.eu

## Abstract

This document is meant to be a practical introduction to the analytical and numerical manipulation of fermionic Gaussian systems. Starting from the basics, we move to relevant modern results and techniques, presenting numerical examples and studying relevant Hamiltonians, such as the transverse field Ising Hamiltonian, in detail. We finish introducing novel algorithms connecting fermionic Guassian states with matrix product states techniques. All the numerical examples make use of the free Julia package `F_utilities`.

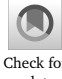
# 1 Introduction

In these notes, we review the properties of free fermionic systems. When we talk about free fermionic systems we naturally think of them as an idealisation of electrons weakly interacting

with the electro-magnetic field. This happens, for example, in vacuum, when an electron is separated from any other electron and moves slowly.

Free fermionic systems appear naturally even as emerging collective excitations in materials [1]. For example the Landau pseudo-particles in Fermi-liquids that describe metals behave as free. Most of the metal properties can be described through the band theory of these collective free electrons.

In one dimension the relevance of free fermions is even larger since they provide the natural language for describing some of the integrable spin chains, such as the Ising model and its derivates [2,3]. Furthermore, the possible phases of free fermionic systems in 1D have been fully characterized [4].

In higher dimensions, the analogous program is still ongoing, but by studying the topology of free fermionic bands, one can naturally identify and characterise a plaethora of topological materials [5].

Similarly to condensed matter, quantum chemistry often requires properly describing the orbitals of complicated molecules. In first approximation, one uses a Hartree Fock approach that is based on identifying the best set of free electronic orbitals to describe the molecule of interest [6].

As a result, just by properly understanding systems of free fermions, one can understand a great deal of modern physics. For these reasons, free fermionic systems have been widely used in advancing our understanding of new fields of physics such as e.g. the quantum information approach to many-body quantum systems [7].

More importantly, free fermionic systems are also at the base of approximate methods in solid-state physics and quantum chemistry. There the interacting multi-electron problem is solved by finding the best set of non-interacting orbitals that approximate the system [8].

Here we consider a different aspect of the free fermionic systems. We study them from the computational point of view, as a tool for benchmarking new ideas and new algorithms for solving many-body quantum systems. New approaches to quantum many-body systems inspired by quantum information and based on tensor networks, indeed, can be thought of as perturbation theories in terms of the entanglement in the system rather than around its interactions [9–11]. Roughly speaking, there is a consensus that the computational cost of simulating quantum many-body systems with classical computers increases as the entanglement increases (for a review on the topic of entanglement in many-body physics see [12] and references therein).

Free systems constitute an exception to this. In fact, no matter their entanglement structure, the Hamiltonian of a system of free fermions, can be diagonalised with a cost proportional to $N^3$, where $N$ is the number of fermionic modes in the system, and, as a result, most of the interesting states of the system can be obtained for relatively large mode sets. For example, a Fermi-sea in one dimension can have logarithmic violation to the area law, and we can build arbitrarily entangled free systems, across one partition (see e.g. the rainbow chain [13,14]).

We will review how to construct ground states, thermal states, diagonal ensembles, and states resulting from the out-of-equilibrium evolution resulting from quenches or time-dependent Hamiltonians.

Free systems provide the natural benchmark for algorithmic ideas on how to treat generic systems. The first example of this approach is presented in [15]. Successively, algorithms such as DMRG were introduced for free fermionic system [16].

This review describes a software package in Julia that can be used to experiment with those systems. The package is freely available in GitHub and provides the basic building blocks of any numerical simulation of free fermionic systems.

As specific examples, we review how to encode a one-dimensional system in a matrix product state at the level of correlation matrices, thus obtaining algorithms to encode optimal

Gaussian matrix product states [17] given the Hamiltonian of the system. The two most well-known algorithms used in this context, namely DMRG and the time-evolving block decimation are reviewed in this setting and explicitly implemented.

We also review how to perform real-time evolution using the correlation matrices, something that we have used for example in [18,19].

We also discuss how to encode a system of free fermions on a cylinder as a matrix product state. Using matrix product states for describing 2D systems is something that has found applications in detecting topological phases of matter [20,21]. Here we use the natural mixed coordinates momentum and real space, respectively in the periodic direction and in the open direction to describe the 2D system as a set of decoupled one-dimensional chains. By using the previously introduced time-evolving block decimation on each of those chains, we can build a matrix product state for the two-dimensional system.

## 2 Basics

### 2.1 The canonical anticommutation relations

Consider a set of operators $\{a_i\}_{i=1}^N$ acting on a Hilbert space $\mathcal{H}$. These operators satisfy the *canonical anticommutation relation* (CAR)

$$\{a_i, a_j^\dagger\} = \mathbb{I}\delta_{i,j}; \qquad \{a_i, a_j\} = 0, \tag{1}$$

where the curly brakets $\{a, b\} := ab + ba$ denote the anticommutator.

As shown in [22] a number of properties for the set of operators $\{a_i\}_{i=1}^N$ and for the Hilbert space $\mathcal{H}$ can be inferred just by the fact that such operators exist and obey the CAR.

The $a_i^\dagger a_i$ form a set of *commuting, Hermitian, positive operators* with eigenvalues $\{0, 1\}$. We denote with $\vec{x} \in \{0, 1\}^N$ a binary string of length $N$ with the $i$-th elements $x_i$. With $|\vec{x}\rangle$ we identify one of the $2^N$ states that is the simultaneous eigenstate of $a_i^\dagger a_i$ for all $i = 1, \dots, N$ with eigenvalues respectively $x_i$. The operator $a_i$ acts as a *lowering operator* for $a_i^\dagger a_i$ and $a_i^\dagger$ acts as a *raising operator* for $a_i^\dagger a_i$ in the sense that

1. If $a_i^\dagger a_i |\vec{x}\rangle = |\vec{x}\rangle$, that is, $|\vec{x}\rangle$ is an eigenvector of $a_i^\dagger a_i$ with eigenvalue equal to 1, then the action of $a_i$ on $|\vec{x}\rangle$ lowers the corresponding eigenvalue, meaning that $a_i^\dagger a_i(a_i|\vec{x}\rangle) = 0(a_i|\vec{x}\rangle)$.

2. If $a_i^\dagger a_i |\vec{x}\rangle = 0|\vec{x}\rangle$, that is, $|\vec{x}\rangle$ is an eigenvector of $a_i^\dagger a_i$ with eigenvalue equal to 0, then the action of $a_i^\dagger$ on $|\vec{x}\rangle$ raises the corresponding eigenvalue, meaning that $a_i^\dagger a_i(a_i^\dagger|\vec{x}\rangle) = 1(a_i^\dagger|\vec{x}\rangle)$.

We define an *ordering* by explicitly defining $|\vec{x}\rangle := (a_1^\dagger)^{x_1}(a_2^\dagger)^{x_2}\dots(a_N^\dagger)^{x_N}|\vec{0}\rangle$, where $\vec{0}$ is the string of $N$ zeros. The set $\{|\vec{x}\rangle\}_{\vec{x}\in\{0,1\}^N}$ forms an orthonormal basis. Since the dimension of the Hilbert space $\mathcal{H}$ is $2^N$, then $\{|\vec{x}\rangle\}_{\vec{x}\in\{0,1\}^N}$ is an orthonormal basis of $\mathcal{H}$.

The orthonormal basis $\{|\vec{x}\rangle\}_{\vec{x}\in\{0,1\}^N}$ is called *Fock basis*. The action of the raising and lowering operators on $|\vec{x}\rangle$ is then

$$a_i|\vec{x}\rangle = \begin{cases} -(-1)^{S_{\vec{x}}^i}|\vec{x'}\rangle \text{ with } x_i' = 0 \text{ and } x_{j\neq i}' = x_{j\neq i}, & \text{if } x_i = 1 \\ 0 & \text{if } x_i = 0 \end{cases}, \tag{2}$$

$$a_i^\dagger|\vec{x}\rangle = \begin{cases} 0 & \text{if } x_i = 1 \\ -(-1)^{S_{\vec{x}}^i}|\vec{x'}\rangle \text{ with } x_i' = 1 \text{ and } x_{j\neq i}' = x_{j\neq i}, & \text{if } x_i = 0 \end{cases}, \tag{3}$$

with $S_{\vec{x}}^i = \sum_{k=1}^{i-1} x_k$.

In appendix A we report some useful equalities valid for operators satisfying the CAR.

## 2.2 Dirac and Majorana representations

The raising and lowering operators $a_i^\dagger, a_i$ are called *Dirac operators* and they represent the action of adding and removing the $i$-th fermionic mode.

Both $a_i$ and its adjoint $a_i^\dagger$ are not Hermitian. The Hermitian combinations of the raising and lowering operators

$$x_i = \frac{a_i + a_i^\dagger}{\sqrt{2}}, \qquad p_i = \frac{a_i - a_i^\dagger}{i\sqrt{2}}, \tag{4}$$

are called *Majorana operators*.

The inverse transformations are:

$$a_i = \frac{x_i + i p_i}{\sqrt{2}}, \qquad a_i^\dagger = \frac{x_i - i p_i}{\sqrt{2}}. \tag{5}$$

In terms of Majorana operators the CARs read as

$$\{x_i, x_j\} = \{p_i, p_j\} = \delta_{i,j}, \qquad \{x_i, p_j\} = 0. \tag{6}$$

We remark that to Majorana operators labelled by $i$ correspond Dirac operators labelled by $i$. Moving between Majorana and Dirac operators does not mix modes.

**Vector notation**  We can collect the Dirac operators of a system with $N$ modes in the vector $\vec{\alpha}$ of length $2N$ defined as

$$\vec{\alpha} = \begin{pmatrix} a_0^\dagger \\ \vdots \\ a_{N-1}^\dagger \\ a_0 \\ \vdots \\ a_{N-1} \end{pmatrix}, \quad \vec{\alpha}^\dagger = \begin{pmatrix} a_0 & \dots & a_{N-1} & a_0^\dagger & \dots & a_{N-1}^\dagger \end{pmatrix}. \tag{7}$$

Analogously we can collect the Majorana operators in the vector $\vec{r}$ defined as

$$\vec{r} = \begin{pmatrix} x_0 \\ \vdots \\ x_{N-1} \\ p_0 \\ \vdots \\ p_{N-1} \end{pmatrix}, \tag{8}$$

in terms of $\vec{r}$ the CAR are conveniently written as

$$\{r_i, r_j\} = \delta_{i,j}. \tag{9}$$

We define the unitary matrix $\Omega$ as

$$\Omega = \frac{1}{\sqrt{2}} \begin{pmatrix} \mathbb{I} & \mathbb{I} \\ i\mathbb{I} & -i\mathbb{I} \end{pmatrix}, \quad \Omega^\dagger = \Omega^{-1} = \frac{1}{\sqrt{2}} \begin{pmatrix} \mathbb{I} & -i\mathbb{I} \\ \mathbb{I} & i\mathbb{I} \end{pmatrix}. \tag{10}$$

Such a matrix, applied to the vector of the Dirac operators $\vec{\alpha}$, returns the vector of Majorana operators $\vec{r} = \Omega\vec{\alpha}$.

**Fermionic transformation**   A transformation $\vec{r} \to \vec{s} = O\vec{r}$ is said to preserve the CAR in the Majorana representation if it maps a vector of Majorana operators $\vec{r}$ to a new one $\vec{s} = O\vec{r}$. If we explicitly impose $\vec{s} = O\vec{r}$ to preserve the CAR we obtain

$$\delta_{i,j} = \{s_i, s_j\} = \sum_{k,l} O_{i,k} O_{j,l} \{r_k, r_l\} = (OO^T)_{i,j}, \tag{11}$$

thus matrix $O$ must be an orthogonal matrix.

We call *fermionic transformation* any transformation $\vec{\alpha} \to \vec{\beta} = U\vec{\alpha}$ that preserves the CAR of the Dirac operators vectors. Matrix $U$ has the form of $U = \Omega^\dagger O \Omega$ with $O$ an orthogonal matrix. It has been shown in [23] that fermionic transformations are generated by fermionic quadratic Hamiltonian (to be defined later), thus have the general form $U = e^{-i\hat{H}}$, with $\hat{H}$ a generic fermionic quadratic Hamiltonian.

**Clifford Algebra**   The Majorana operators $\{r_i\}_{i=1,\dots,2N}$ are Hermitian, traceless and generate the Clifford algebra denoted by $\mathcal{C}_{2N}$.

Any arbitrary operator $X \in \mathcal{C}_{2N}$ can be represented as a polynomial of the Majorana operators as [23]

$$X = \alpha_0 \mathbb{I} + \sum_{p=1}^{2N} \sum_{1 \le q_1 < \dots < q_p \le 2N} \alpha_{q_1,\dots,q_p} r_{q_1} \dots r_{q_p}, \tag{12}$$

where $\mathbb{I}$ is the identity and the coefficients $\alpha_0$ and $\alpha_{q_1,\dots,q_p}$ are real. When the representation of $X \in \mathcal{C}_{2N}$ involves only even powers of Majorana operators, we call it an *even operator*. If the representation of $X$ involves only odd powers of Majorana operators, then $X$ is called *odd* operator.

We define the *parity operator* as

$$P = (i)^{2N} r_1 r_2 \dots r_{2N} = \prod_{i=1}^{N} (\mathbb{I} - 2a_i^\dagger a_i). \tag{13}$$

Every even operator $X$ commutes with the parity operator $P$. The parity $p_X$ of an operator $X$ is defined as $PX = p_X X$ and it can only assume the two values $p_X \in \{-1, 1\}$.

fermionic quadratic Hamiltonians (to be defined later) are even operators. For an $N$-mode fermionic system with orthonormal basis $\{|\vec{x}\rangle\}$, the matrices $|\vec{x}\rangle\langle\vec{x}|$ defined for every $\vec{x}$ have the polynomial representation

$$|\vec{x}\rangle\langle\vec{x}| = \left(\frac{\mathbb{I}}{2} - i(-1)^{x_1} r_1 r_2\right) \left(\frac{\mathbb{I}}{2} - i(-1)^{x_2} r_3 r_4\right) \dots \left(\frac{\mathbb{I}}{2} - i(-1)^{x_N} r_{2N-1} r_{2N}\right), \tag{14}$$

thus $\{|\vec{x}\rangle\langle\vec{x}|\}$ are all even operators with parity $p_{|\vec{x}\rangle\langle\vec{x}|} = -(-1)^{\sum_{i=1}^{N} x_i}$.

Mixed matrices $|\vec{x}\rangle\langle\vec{x}'|$ with $\vec{x} \ne \vec{x}'$ are odd operators if $\mathrm{mod}(d(\vec{x}, \vec{x}'), 2) = 1$, where $d(\vec{x}, \vec{y})$ is the Hamming distance[1] of $\vec{x}$ and $\vec{y}$, and they are even operators if $\mathrm{mod}(d(\vec{x}, \vec{x}'), 2) = 0$.

## 2.3   Fermionic Quadratic Hamiltonians

## 2.4   Dirac Representation

The general *fermionic quadratic Hamiltonians* (f.q.h.) on a finite lattice of $N$ sites in the Dirac operators representation can be written as

$$\hat{H} = \sum_{i,j=1}^{N} \left(A_{i,j} a_i^\dagger a_j - \bar{A}_{i,j} a_i a_j^\dagger + B_{i,j} a_i a_j - \bar{B}_{i,j} a_i^\dagger a_j^\dagger\right), \tag{15}$$

---

[1]The Hamming distance between two string is the minimum number of substitutions required to change one string into the other. For example, the Hamming distance between $\vec{x}$="000" and $\vec{y}$"=010" is 1 since flipping the bit $y_2$ is enough to change $\vec{y}$ to $\vec{x}$.

where $A$ is a *Hermitian* complex matrix, $A^\dagger = A$, and $B$ is a *skew-symmetric* complex matrix, $B^T = -B$ .

Defining the matrix

$$H = \begin{pmatrix} -\bar{A} & B \\ -\bar{B} & A \end{pmatrix}, \tag{16}$$

the *compact form* of equation (15) reads

$$\hat{H} = \vec{\alpha}^\dagger H \vec{\alpha}. \tag{17}$$

We will call *Hamiltonians* both $\hat{H}$ and $H$ as for a fixed choice of Dirac operators one completely identifies the other.

## 2.5 Majorana Representation

The Majorana representation of the generic f.q.h. reads as

$$\hat{H} = i \sum_{i,j=1}^{N} \left( h_{i,j}^{xx} x_i x_j + h_{i,j}^{pp} p_i p_j + h_{i,j}^{xp} x_i p_j + h_{i,j}^{px} p_i x_j \right) = i \vec{r}^\dagger h \vec{r}, \tag{18}$$

where

$$ih = \Omega H \Omega^\dagger = i \begin{pmatrix} \Im\{A+B\} & \Re\{A+B\} \\ \Re\{B-A\} & \Im\{A-B\} \end{pmatrix} = i \begin{pmatrix} h^{xx} & h^{xp} \\ h^{px} & h^{pp} \end{pmatrix}. \tag{19}$$

Where $\Im\{\cdot\}$ and $\Re\{\cdot\}$ are respectively the imaginary and the real part of their argument.

Using the properties of matrices $A$ and $B$, it is easy to see that matrix $h$ is real and skew-symmetric.

## 2.6 Diagonalisation

**Diagonal form of the Hamiltonian with Dirac operators**  Given a particular f.q.h. $\hat{H}$ in the general form (15) it is always possible to find a new set of Dirac operators $\{b_k\}_{k=1}^{N}$ such that $\hat{H}$ in terms of $\{b_k\}_{k=1}^{N}$ reads as

$$\hat{H} = \sum_{k=1}^{N} \epsilon_k (b_k^\dagger b_k - b_k b_k^\dagger), \tag{20}$$

with $\epsilon_k \in \mathbb{R}$ for all $k = 1, 2, \dots, N$ [24].

We call Hamiltonians in this form free-free fermion Hamiltonians.

In compact form

$$\hat{H} = \vec{\beta}^\dagger H_D \vec{\beta}, \tag{21}$$

with

$$H_D = U^\dagger H U = \begin{pmatrix} -\epsilon_1 & 0 & \dots & & \dots & 0 \\ 0 & \ddots & \ddots & & & \vdots \\ \vdots & \ddots & -\epsilon_N & & & \\ & & & \epsilon_1 & \ddots & \vdots \\ \vdots & & & \ddots & \ddots & 0 \\ 0 & \dots & & \dots 0 & \epsilon_N \end{pmatrix}, \tag{22}$$

where $\vec{\beta}$ is the collection of the Dirac operators $b_k, b_m^\dagger$ ordered as in $\vec{\alpha}$, and $U$ is the fermionic transformation that diagonalises the Hamiltonian.

We will always order the eigenvalues in descending order ($\epsilon_1 \geq \epsilon_2 \geq \cdots \geq \epsilon_N \geq 0$).

**Diagonal form of the Hamiltonian with Majorana operators**   In terms of Majorana operators the diagonal form of a generic f.q.h. reads as

$$\hat{H} = i \sum_{i=1}^{N} \lambda_i (\tilde{x}_i \tilde{p}_i - \tilde{p}_i \tilde{x}_i), \tag{23}$$

for a set of Majorana operators $\{\tilde{x}_i\}_i$, $\{\tilde{p}_i\}_i$. In compact form

$$\hat{H} = i\vec{s}^{\dagger} h_D \vec{s}, \tag{24}$$

where $\vec{s}$ is the collection of the Majorana operators $\tilde{x}_i, \tilde{p}_j$ ordered as (31) and where

$$h_D = O^T h O = \bigoplus_{i=1}^{N} \begin{pmatrix} 0 & \lambda_i \\ -\lambda_i & 0 \end{pmatrix} \tag{25}$$

is a block diagonal matrix and $O$ the orthogonal transformation that diagonalises the Hamiltonian in the Majorana operators representation. Substituting the definition of Majorana operators (4) into equation (23) and confronting with equation (20) we note that $\epsilon_k = \lambda_k$.

## 2.7   Numerical diagonalisation

As seen in subsection 2.6, diagonalising a general f.q.h. $\hat{H}$ reduces to diagonalising (or to block diagonalise in the case of Majorana representation) the matrix $H$ of its compact form.
We are thus interested in finding the fermionic transformation $U$ that maps $H$ and the vector of Dirac operators $\vec{\alpha}$ respectively to the diagonal matrix $H_D = U^{\dagger} H U$ and to the vector of Dirac operators $\vec{\beta} = U\vec{\alpha}$ such that, in terms of $\vec{\beta}$, the Hamiltonian is in the diagonal form (20).
Here we focus on the numerical approach, we diagonalise the Hamiltonian using standard matrix decomposition techniques. For a more physical approach we refer to [24].
First step in the diagonalisation procedure is moving to the Majorana representation of $H$

$$\hat{H} = \vec{\alpha}^{\dagger} H \vec{\alpha} = \vec{\alpha}^{\dagger} \Omega^{\dagger} \Omega H \Omega^{\dagger} \Omega \vec{\alpha} = $$
$$= i\vec{r}^{\dagger} h \vec{r}. \tag{26}$$

The following theorem is a standard result in matrix theory [25, 26]

**Theorem 1** (Block diagonal form of skew-symmetric matrices)
*Let $h$ be $2N \times 2N$ a real, skew-symmetric matrix. There exists a real special orthogonal matrix $O$ such that*

$$h = O h_D O^T, \tag{27}$$

*with $h_D$ a block diagonal matrix of the form*

$$h_D = \bigoplus_{i=1}^{N} \begin{pmatrix} 0 & \lambda_i \\ -\lambda_i & 0 \end{pmatrix} \tag{28}$$

*for real, positive-definite $\{\lambda_i\}_{i=1,\dots,N}$. The non-zero eigenvalues of matrix $h$ are the imaginary numbers $\{\pm i \lambda_i\}_{i=1,\dots,N}$.*
*For a more general form of this theorem see appendix A.1.5.*

Matrix $h$ in (18) is real, skew-symmetric, thus, using theorem (27) we know there exists an orthogonal transformation $O$ that diagonalises the matrix

$$\hat{H} = i\vec{r}^{\dagger} h \vec{r} = i\vec{r}^{\dagger} O O^{\dagger} h O O^{\dagger} \vec{r} \tag{29}$$

$$= i\vec{s}^{\dagger} \left( \bigoplus_{i=1}^{N} \begin{pmatrix} 0 & \lambda_i \\ -\lambda_i & 0 \end{pmatrix} \right) \vec{s} = i\vec{s}^{\dagger} h_D \vec{s}. \tag{30}$$

That is $\hat{H} = i\sum_{i=0}^{N-1} \lambda_i(\tilde{x}_i\tilde{p}_i - \tilde{p}_i\tilde{x}_i)$ once defined the new collection of Majorana operators $\vec{s} = O\vec{r}$ as

$$\vec{s} = \begin{pmatrix} \tilde{x}_0 \\ \tilde{p}_0 \\ \tilde{x}_1 \\ \tilde{p}_1 \\ \vdots \\ \tilde{x}_{N-1} \\ \tilde{p}_{N-1} \end{pmatrix}. \tag{31}$$

**Block-diagonal form of real skew-symmetric matrices**   The matrix decomposition (27) of theorem 1 is numerically obtained in 3 steps

1. Compute numerically a Schur decomposition (or Schur triangularisation as in [26]) of the skew-symmetric matrix $h$ such that: $h = \tilde{O}\tilde{h}_D\tilde{O}^T$. The matrix $\tilde{h}_D$ should be a block-diagonal matrix with each block in the anti-diagonal form

$$\begin{pmatrix} 0 & \tilde{\lambda}_i \\ -\tilde{\lambda}_i & 0 \end{pmatrix}. \tag{32}$$

   It is not guaranteed that the $\tilde{\lambda}_i$ are positive for each $i$. It is necessary to reorder them as in step 2.

2. Build the orthogonal matrix $S = \bigoplus_{i=1}^{\lfloor N/2 \rfloor} s_i$ with

$$s_i = \begin{pmatrix} 0 & 1 \\ 1 & 0 \end{pmatrix} \tag{33}$$

   if $\tilde{\lambda}_i < 0$ or

$$s_i = \begin{pmatrix} 1 & 0 \\ 0 & 1 \end{pmatrix}, \tag{34}$$

   if $\tilde{\lambda}_i > 0$.

3. The final orthogonal transformation is $O = \tilde{O}S$ such that $h = Oh_DO^T$.

---

**F_utilities  2.1: `Diag_real_skew(h)` $\to h_D, O$**

This function implements the algorithm for the block diagonalisation of $h$ a generic skew-symmetric real matrix. $h_D$ is the block-diagonal matrix of (27) and has the following property: it is in the block diagonal form, each $2 \times 2$ block is skew-symmetric with the upper-right element positive and real and $h_D$ is in ascending order for the upper diagonal. $O$ is an orthogonal matrix such that: $h = Oh_DO^T$.

---

**Moving back to Dirac representation**   The vector of Majorana operators $\vec{s}$ (31) has a different ordering with respect to the vector $\vec{r}$. We call the order of the operators in $\vec{s}$ an $xp$ ordering and the ordering of the operators in $\vec{r}$ and $xx$ ordering. The transformation matrix $\Omega^\dagger$ maps a vector of Majorana operators in $xx$ ordering to a vector of Dirac operators, thus, before being

able to move to the Dirac representation we have to reorder the element of vector $\vec{s}$. To do so we use the matrix

$$
F_{xp \to xx} =
\begin{array}{c}
i=0 \\
i=1 \\
\vdots \\
\vdots \\
i=N \\
i=N+1 \\
\vdots \\
i=2N+1
\end{array}
\begin{pmatrix}
1 & 0 & 0 & 0 & \dots & \dots & 0 & 0 \\
0 & \vdots & 1 & \vdots & & & \vdots & \vdots \\
\vdots & \vdots & \vdots & \vdots & & & 0 & \vdots \\
\vdots & 0 & 0 & 0 & & & 1 & \vdots \\
\vdots & \vdots & 1 & 0 & 0 & & 0 & \vdots \\
\vdots & 0 & \vdots & 1 & & & \vdots & \vdots \\
\vdots & \vdots & \vdots & 0 & & & \vdots & 0 \\
0 & 0 & 0 & \vdots & \dots & \dots & 0 & 1
\end{pmatrix}
\tag{35}
$$

that applied to a vector $\vec{s}$ with the $xp$ ordering returns a vector $\vec{r} = F_{xp \to xx}\vec{s}$ with $xx$ ordering. Mapping back to the Dirac representation we obtain the diagonal form of the Hamiltonian in the Dirac operators representation as

$$
\hat{H} = i\left(\vec{s} F_{xp \to xx}^{T} \Omega\right)\left(\Omega^{\dagger} F_{xp \to xx} h_D F_{xp \to xx}^{T} \Omega\right)\left(\Omega^{\dagger} F_{xp \to xx}\vec{s}\right)
\tag{36}
$$

$$
= \sum_{k=1}^{N} \epsilon_k (b_k^{\dagger} b_k - b_k b_k^{\dagger}) = \vec{\beta}^{\dagger} H_D \vec{\beta}.
\tag{37}
$$

The fermionic transformation $U$ that diagonalises the Hamiltonian H in the form (22) is

$$
U = \Omega^{\dagger} \cdot O \cdot F_{xp \to xx}^{\dagger} \cdot \Omega.
\tag{38}
$$

> **F_utilities  2.2: Diag_h($H$) $\to H_D, U$**
>
> This function diagonalises $H$. $H_D$ is the diagonal form with the first half diagonal negative and the second one positive ordered as (22). $U$ is the fermionic transformation such that: $H = U H_D U^{\dagger}$.

# 3 Fermionic Gaussian States

## 3.1 Fermionic Gaussian states

**Definition 1** (fermionic Gaussian state)
*A state $\rho$ is a fermionic Gaussian state (f.g.s.) if it can be represented as*

$$
\rho = \frac{e^{-\hat{H}}}{Z} = \frac{e^{-\vec{a}^{\dagger} H \vec{a}}}{Z},
\tag{39}
$$

*with $Z := Tr\left[e^{-\hat{H}}\right]$ a normalisation constant and $\hat{H}$ a fermionic Gaussian Hamiltonian called parent Hamiltonian of $\rho$.*
*Every possible value of the norm of the Hamiltonian is admitted, $\left\|\hat{H}\right\|_1 \in [0, +\infty]$. Both extremum values are reached with a single sided limit procedure in the definition of $\rho$.*
*All the information about the state is encoded in the $2N \times 2N$ matrix H at the exponential.*

Fermionic Gaussian states have an immediate interpretation as thermal Gibbs states of f.q.h.. One can even rescale the parent Hamiltonian as $\hat{\hat{H}} = \frac{1}{\beta}\hat{H}$ such that $\left\|\hat{\hat{H}}\right\| = 1$ and $\beta = \frac{1}{\|\hat{H}\|}$. In this way the state reads as $\rho = \frac{e^{-\beta\hat{\hat{H}}}}{Z}$ with $\beta \in [0, +\infty]$. Since f.g.s are exponential of f.g.h. and f.g.h. are even operator, it follows that f.g.s are even operator.

**Single mode Gaussian states** Consider the single mode parent Hamiltonian $\hat{H}_1 = \epsilon(b^\dagger b - b b^\dagger)$ of the f.g.s. $\rho = \frac{1}{Z}e^{-\hat{H}_1}$. The explicit representation of $\rho$ on the basis $\{b^\dagger|0\rangle, |0\rangle\}$ is

$$\rho = \begin{pmatrix} 1-f & 0 \\ 0 & f \end{pmatrix}, \tag{40}$$

where $f = \langle 0|\rho|0\rangle$ and the two coherences are 0 because we cannot have the odd terms $|0\rangle\langle 1|$ and $|1\rangle\langle 0|$ in the expansion of the even operator $\rho$ (see [27, 28] for a detailed and beautiful analysis of the admitted coherences). Using the polynomial expansion (14) we can see that $f = \mathrm{Tr}\left[\rho b^\dagger b\right] := \langle b^\dagger b\rangle$, that is the occupation of the fermionic mode, thus a single mode Gaussian state is completely characterised by the occupation $\langle b^\dagger b\rangle$.

## 3.2 Correlation Matrix

We have seen that for any f.q.h. $H$ it is always possible to find a fermionic transformation $U$ that diagonalises $H$ transforming the Dirac operators vector as $\vec{\beta} = U\vec{\alpha}$. Diagonalising the parent Hamiltonian of a f.g.s. $\rho$ we obtain its decomposition in terms of single-mode thermal states

$$\rho = \frac{e^{-\vec{\beta}^\dagger H_D \vec{\beta}}}{Z} = \frac{1}{Z}\bigotimes_{k=1}^{N} e^{-\epsilon_k(b_k^\dagger b_k - b_k b_k^\dagger)} = \bigotimes_{k=1}^{N} \frac{e^{-\epsilon_k(b_k^\dagger b_k - b_k b_k^\dagger)}}{Z_k}, \tag{41}$$

where $Z_k = \mathrm{Tr}\left[e^{-\epsilon_k(b_k^\dagger b_k - b_k b_k^\dagger)}\right]$.

Each single-mode thermal state is completely characterised by its occupation number, thus $\rho$ is completely characterised by the set of occupations $\{\langle b_i^\dagger b_i\rangle\}_{i=1}^{N}$. Expressing the occupations in terms of the operators $\vec{\alpha} = U^\dagger\vec{\beta}$, we find that every f.g.s. is *completely characterised* by the collection of all the correlators $\Gamma_{i,j}^{a^\dagger a} := \langle a_i^\dagger a_j\rangle$ and $\Gamma_{i,j}^{aa} := \langle a_i a_j\rangle$. We collect these correlators in the so called *correlation matrix*

$$\Gamma := \langle\vec{\alpha}\vec{\alpha}^\dagger\rangle = \begin{pmatrix} \Gamma^{a^\dagger a} & \Gamma^{a^\dagger a^\dagger} \\ \Gamma^{aa} & \Gamma^{aa^\dagger} \end{pmatrix} \tag{42}$$

with $\Gamma_{i,j}^{aa} = -\overline{\Gamma_{i,j}^{a^\dagger a^\dagger}}$ and $\Gamma_{i,j}^{aa^\dagger} = (\mathbb{I} - \Gamma^{a^\dagger a})_{i,j}^\dagger$, where $\overline{A}$ is the conjugate of $A$. The correlation matrix $\Gamma$ is Hermitian, $\Gamma^{aa}$ and $\Gamma^{a^\dagger a^\dagger}$ are skew-symmetric, and $\Gamma^{a^\dagger a}$ and $\Gamma^{aa^\dagger}$ are Hermitian. Expressed in terms of Majorana operator the correlation matrix is defined as

$$\Gamma^{maj} := \langle\vec{r}\vec{r}^\dagger\rangle = \Omega\Gamma\Omega^\dagger. \tag{43}$$

It is interesting observing that, since a f.g.s. is completely described by its correlation matrix, with the spirit of the maximum entropy principle (see [29, 30]), it is possible to equivalently define fermionic Gaussian states as the states that maximise the von Neumann entropy given the expectation values collected in the correlation matrix.

### 3.3   Covariance matrix

The *covariance matrix* of a f.g.s. is the real, skew-symmetric matrix defined as

$$\gamma := i \operatorname{Tr}\big[\rho[\vec{r}_i, \vec{r}_j]\big], \tag{44}$$

with $[\vec{r}_i, \vec{r}_j]$ the commutator of the two Majorana operators $\vec{r}_i$ and $\vec{r}_j$.
As for the correlation matrix, the covariance matrix of a f.g.s $\rho$ completely describes the states. In fact $\gamma$ and $\Gamma$ are related by the equality

$$\gamma = -i\Omega(2\Gamma - \mathbb{I})\Omega^\dagger = -i\left(2\Gamma^{maj} - \mathbb{I}\right). \tag{45}$$

We will use both the covariance matrix and the correlation matrix approach.

### 3.4   Wick's theorem

As mentioned, f.g.s. are fully characterised by their covariance matrix. This means that it must be possible to obtain the expectation value of every operator $X$ from $\gamma$ solely. To do so we just need to take the polynomial expansion (12) of $X$ and apply the celebrated Wick's theorem [31] to each monomial term. The Wick's theorem states that for a f.g.s. $\rho$ and a monomial of Majorana operators $r_{q_1} r_{q_2} \dots r_{q_p}$ one has

$$\operatorname{Tr}\left[\rho\, r_{q_1} r_{q_2} \dots r_{q_p}\right] = \operatorname{Pf}(\gamma_{|q_1, q_2, \dots, q_p}), \tag{46}$$

where $1 \le q_1 < q_2 < \dots < q_p \le 2N$ and $\gamma_{|q_1, q_2, \dots, q_p}$ is the restriction of the covariance matrix to all the two points correlators involving just the Majorana operators $\{r_{q_1}, r_{q_2}, \dots, r_{q_p}\}$ and $\operatorname{Pf}()$ is called the *Pfaffian*.

Since the Pfaffian is nonvanishing only for a $2N \times 2N$ skew-symmetric matrix [32], it is clear that the expectation value of any odd operators is always zero.

#### Example

Consider a system composed by 2 fermionic modes corresponding to the Dirac operators $a_1$ and $a_2$. The Majorana operators vector is $\vec{r} = (r_1, r_2, r_3, r_4)$, thus the covariance matrix takes the form

$$\gamma = \begin{pmatrix} 0 & \langle r_1, r_2 \rangle & \langle r_1, r_3 \rangle & \langle r_1, r_4 \rangle \\ -\langle r_1, r_2 \rangle & 0 & \langle r_2, r_3 \rangle & \langle r_2, r_4 \rangle \\ -\langle r_1, r_3 \rangle & -\langle r_2, r_3 \rangle & 0 & \langle r_3, r_4 \rangle \\ -\langle r_1, r_4 \rangle & -\langle r_2, r_4 \rangle & -\langle r_3, r_4 \rangle & 0 \end{pmatrix}, \tag{47}$$

where $\langle r_i, r_j \rangle := i \operatorname{Tr}\big[\rho[r_i, r_j]\big]$. Using Wick's theorem we have that

$$\operatorname{Tr}\left[\rho\, r_1 r_2 r_3 r_4\right] = \operatorname{Pf}\begin{pmatrix} 0 & \langle r_1, r_2 \rangle & \langle r_1, r_3 \rangle & \langle r_1, r_4 \rangle \\ -\langle r_1, r_2 \rangle & 0 & \langle r_2, r_3 \rangle & \langle r_2, r_4 \rangle \\ -\langle r_1, r_3 \rangle & -\langle r_2, r_3 \rangle & 0 & \langle r_3, r_4 \rangle \\ -\langle r_1, r_4 \rangle & -\langle r_2, r_4 \rangle & -\langle r_3, r_4 \rangle & 0 \end{pmatrix} \tag{48}$$

$$= \langle r_1, r_2 \rangle \langle r_3, r_4 \rangle - \langle r_1, r_3 \rangle \langle r_2, r_4 \rangle + \langle r_2, r_3 \rangle \langle r_1, r_4 \rangle, \tag{49}$$

and

$$\operatorname{Tr}\left[\rho\, r_2 r_4\right] = \operatorname{Pf}\begin{pmatrix} 0 & \langle r_2, r_4 \rangle \\ -\langle r_2, r_4 \rangle & 0 \end{pmatrix} = \langle r_2, r_4 \rangle, \tag{50}$$

and

$$\operatorname{Tr}\left[\rho\, r_1 r_2 r_4\right] = \operatorname{Pf}\begin{pmatrix} 0 & \langle r_1, r_2 \rangle & \langle r_1, r_4 \rangle \\ -\langle r_1, r_2 \rangle & 0 & \langle r_2, r_4 \rangle \\ -\langle r_1, r_4 \rangle & -\langle r_2, r_4 \rangle & 0 \end{pmatrix} = 0. \tag{51}$$

## 3.5 Diagonalisation of the correlation matrix

In subsection 3.2 we have seen that for any f.g.s. $\rho$ there exists a fermionic transformation $U$ that diagonalises its parent Hamiltonian . With the new Dirac operators $\vec{\beta}$ the state can be expressed as a tensor product of single mode thermal states

$$\frac{e^{-\vec{\beta}^{\dagger} H_D \vec{\beta}}}{Z} = \frac{1}{Z} \bigotimes_{k=1}^{N} e^{-\epsilon_k (b_k^{\dagger} b_k - b_k b_k^{\dagger})} = \bigotimes_{k=1}^{N} \frac{e^{-\epsilon_k (b_k^{\dagger} b_k - b_k b_k^{\dagger})}}{Z_k}, \tag{52}$$

with $Z_k = \text{Tr}\left[ e^{-\epsilon_k (b_k^{\dagger} b_k - b_k b_k^{\dagger})} \right]$.

Expressed with these operators the correlation matrix is diagonal. If we consider the Fock basis $\{|\vec{k}\rangle\}_{\vec{k} \in \{0,1\}^N}$ built with the action of the operators $\vec{\beta}$ on $|0\rangle$, we have that in this basis $\rho$ assumes a diagonal form. We call $U_\rho$ the unitary transformation that moves from the basis $\{|\vec{x}\rangle\}_{\vec{x} \in \{0,1\}^N}$ to the one of the modes $\{|\vec{k}\rangle\}_{\vec{k} \in \{0,1\}^N}$.

It is easy to see that expressed on this basis $\rho$ has the diagonal form

$$\rho^D = U_\rho^{\dagger} \rho U_\rho = \bigotimes_{i=1}^{N} \begin{pmatrix} \nu_i & 0 \\ 0 & 1 - \nu_i \end{pmatrix}. \tag{53}$$

The same fermionic transformation $U$ that diagonalises the parent Hamiltonian brings $\Gamma$ in the diagonal form

$$\Gamma^D = U^{\dagger} \Gamma U = \begin{pmatrix} \nu_1 & 0 & \dots & & \dots & 0 \\ 0 & \ddots & \ddots & & & \vdots \\ \vdots & \ddots & \nu_N & & & \\ & & & 1-\nu_1 & \ddots & \vdots \\ \vdots & & & \ddots & \ddots & 0 \\ 0 & \dots & & \dots 0 & 1-\nu_N \end{pmatrix}, \tag{54}$$

with $\nu_i \in [0,1]$ the occupation number of the $i$-th free mode. To numerically obtain the diagonal form of the correlation matrix we notice that the covariance matrix $\gamma$ is a real, skew-symmetric matrix, thus using theorem 1 we know that we can find an orthogonal transformation $O$ such that

$$\gamma = O \gamma^D O^T = O \left( \bigoplus_{i=1}^{N} \begin{pmatrix} 0 & \eta_i \\ -\eta_i & 0 \end{pmatrix} \right) O^T, \tag{55}$$

with $\eta_i \in [-\frac{1}{2}, \frac{1}{2}]$.

Following the same procedure of subsection 2.7, we can write the diagonal elements of $\Gamma^D$ as

$$\nu_i = \frac{1}{2} - \eta_i. \tag{56}$$

The elements of $H_D$ and $\Gamma^D$ are related by the following formulas

$$\epsilon_k = \frac{1}{2} \ln\left( \frac{1 - \nu_k}{\nu_k} \right), \tag{57}$$

$$\nu_k = \frac{1}{1 + e^{2\epsilon_k}}, \tag{58}$$

with $\nu_k \in [0,1]$ and $\epsilon_k \in [-\infty, +\infty]$, where the boundary values are taken with a limit. The complete calculation can be found in appendix A.1.1. In (22) we defined all the $\epsilon_k$ to be positive, to use the same notation, one just has to exchange $b$ with $\tilde{b}^{\dagger}$ and $b^{\dagger}$ with $\tilde{b}$, that is

exchanging occupations with vacancies for the mode with $\epsilon_k$ negative. This corresponds to switching $\nu_k$ with $1 - \tilde{\nu}_k$ and $1 - \nu_k$ with $\tilde{\nu}_k$.

In general the correlation matrix $\Gamma$ and the parent Hamiltonian $H$ are related by the formula [33–36]

$$\Gamma = \frac{1}{1 + e^{2H}}. \tag{59}$$

> **F_utilities  3.1: Diag_gamma($\Gamma$) $\rightarrow \Gamma^D, U$**
>
> This function returns $\Gamma^D$, the diagonal form of the Dirac correlation matrix $\Gamma$ and $U$ the fermionic transformation such that $\Gamma = U\Gamma^D U\dagger$.

**Phisicality of a state**    It is known that a matrix $\rho$ represents a valid physical density matrix if it is a positive semi-definite Hermitian matrix with trace equal to one. The condition for a matrix $\Gamma$ to represent a valid physical correlation matrix of a f.g.s. is

$$\Gamma^2 - \Gamma \leq 0, \tag{60}$$

or equivalently

$$\gamma\gamma^\dagger \leq -\mathbb{I}. \tag{61}$$

These conditions are equivalent to the request that all the eigenvalues $\nu_i$ of matrix $\Gamma$ have to belong to the interval $[0, 1]$.

**Ground states of fermionic quadratic Hamiltonians**    Suppose we have a f.q.h. $H$ and that we are interested in obtaining the correlation matrix $\Gamma_0$ associated to its ground state $|0\rangle$. In order to obtain $\Gamma_0$ we proceed by first finding the fermionic transformation $U$ that diagonalise $H$. Since our algorithm associates to each free mode of the diagonalised Hamiltonian a positive energy, in the diagonal basis the ground state is $|0\rangle\langle 0|$. The correlation matrix associated to the state $|0\rangle$ is the block matrix:

$$\Gamma_{|0\rangle\langle 0|} = \begin{pmatrix} 0 & 0 \\ 0 & \mathbb{I}_{N \times N} \end{pmatrix}. \tag{62}$$

To obtain the ground state $\Gamma_0$ we just need to move back to the original basis, thus

$$\Gamma_0 = U\Gamma_{|0\rangle\langle 0|}U^\dagger. \tag{63}$$

> **F_utilities  3.2: GS_gamma($H_D, U$) $\rightarrow \Gamma_0$**
>
> This function returns $\Gamma_0$, the ground state of the Hamiltonian $H = UH_D U^\dagger$.

**Thermal state of fermionic quadratic Hamiltonians**    Suppose we have a f.q.h. $H$ and that we are interested in obtaining the correlation matrix $\Gamma_0$ associated to the thermal state $\rho_\beta = \frac{e^{-\beta H}}{\text{Tr}[e^{-\beta H}]}$.

As we did for computing the ground state, we move to the diagonal basis with the fermionic transformation $U$. In the diagonal basis the thermal state has the correlation matrix

$$\Gamma_\beta^D = \begin{pmatrix} \frac{1}{1+e^{2\beta\epsilon_1}} & 0 & \cdots & & \cdots & & 0 \\ 0 & \ddots & \ddots & & & & \vdots \\ \vdots & \ddots & \frac{1}{1+e^{2\beta\epsilon_N}} & & & & \\ & & & \frac{1}{1+e^{-2\beta\epsilon_1}} & \ddots & & \vdots \\ \vdots & & & \ddots & \ddots & & 0 \\ 0 & \cdots & & & \cdots & 0 & \frac{1}{1+e^{-2\beta\epsilon_N}} \end{pmatrix}. \tag{64}$$

To obtain the thermal state $\Gamma_\beta$ we just need to move back to the original basis, thus

$$\Gamma_\beta = U\Gamma_\beta^D U^\dagger. \tag{65}$$

---

**F_utilities 3.3: Thermal_fix_beta($H_D, U, \beta$) $\to \Gamma_0$**

This function returns $\Gamma_\beta$, the termal state at inverse temperature $\beta$ of the Hamiltonian $H = UH_DU^\dagger$.

---

**F_utilities 3.4: Thermal_fix_energy($H_D, U, E$) $\to \Gamma_{\beta(E)}, \beta(E), \Delta(E)$**

This function variationally computes and then returns $\Gamma_{\beta(E)}$, the thermal state at inverse temperature $\beta(E)$ of the Hamiltonian $H = UH_DU^\dagger$, and $\beta(E)$ the temperature such that $\text{Tr}\left[\rho_{\beta(E)}H\right] = E$ and $\Delta(E)$ the difference between the required energy $E$ and the actual energy of the state $\Gamma_{\beta(E)}$. It outputs the precision $\Delta(E)$ and $\beta(E)$.

---

**Energy of a fermionic Gaussian state**   Consider a f.q.h $H$ and a f.g.s. $\Gamma$. The energy of $\Gamma$ with respect to $H$ is the expectation value $\text{Tr}\left[\hat{H}\rho\right]$ of the associated $\hat{H}$ computed on the associated state $\rho$.

In order to compute this expectation value one just needs to find the fermionic transformation $U$ that diagonalises $H$. With this, one is able to find the energies $\epsilon_k$ and the occupations $\langle b_k^\dagger b_k \rangle$ and $\langle b_k b_k^\dagger \rangle$. The correlation matrix $\Gamma$ is not diagonal in the diagonal basis $\vec{\beta}$ of $H$, but we are just interested in its diagonal elements.

The energy $E_H(\Gamma)$ of $\Gamma$ is thus

$$E_H(\Gamma) = \sum_k \epsilon_k(\langle b_k^\dagger b_k \rangle - \langle b_k b_k^\dagger \rangle). \tag{66}$$

---

**F_utilities 3.5: Energy($\Gamma, H_D, U$) $\to E_H(\Gamma)$**

This function returns $E_H(\Gamma)$ the energy of the state $\Gamma$ calculated with $H$. Matrices $H_D$ and $U$ are the output of `Diag_h(H)`.

---

## 3.6 Eigenvalues of $\rho$ and eigenvalues of $\Gamma$

We have seen that the diagonal form of the correlation matrix $\Gamma$ and of the density matrix $\rho$ of a f.g.s. can be obtained respectively with a fermionic transformation $U$ and a unitary operation

$U_\rho$. The Fock basis in which $\rho$ is diagonal is the one generated by the set of operators that expresses $\Gamma$ in diagonal form.

In these two basis $\rho$ and $\Gamma$ assume the forms

$$
\Gamma^D = \begin{pmatrix}
v_1 & 0 & \cdots & & \cdots & 0 \\
0 & \ddots & \ddots & & & \vdots \\
\vdots & \ddots & v_N & & & \\
& & & 1-v_1 & \ddots & \vdots \\
\vdots & & & \ddots & \ddots & 0 \\
0 & \cdots & & \cdots 0 & 1-v_N
\end{pmatrix},
$$

$$
\rho^D = \bigotimes_{k=1}^{N} \begin{pmatrix} v_k & 0 \\ 0 & 1-v_k \end{pmatrix} = \begin{pmatrix} \pi_{\vec{0}} & 0 & \cdots \\ 0 & \ddots & \vdots \\ \vdots & \cdots & \pi_{\vec{1}} \end{pmatrix}. \tag{67}
$$

Thus if we denote each of the $2^N$ eigenvalues $\pi_{\vec{x}}$ of $\rho$ with a binary string $\vec{x} \in \{0,1\}^N$ we have that

$$
\pi_{\vec{x}} = \prod_{k=1}^{N} (\vec{x}_k v_i + (1-\vec{x}_k)(1-v_k)) . \tag{68}
$$

For example in the case of $N = 2$ one has the four eigenvalues:

$$
\pi_{00} = (1-v_1)(1-v_2), \qquad \pi_{01} = (1-v_1)v_2, \qquad \pi_{10} = v_1(1-v_2), \qquad \pi_{11} = v_1 v_2.
$$

It is evocative changing the order of the Dirac operators in the representation of $\Gamma^D$

$$
\vec{\beta} = \begin{pmatrix} b_0^\dagger \\ \vdots \\ b_{N-1}^\dagger \\ b_0 \\ \vdots \\ b_{N-1} \end{pmatrix} \qquad \rightarrow \qquad \vec{\tilde{\beta}} = \begin{pmatrix} b_0^\dagger \\ b_0 \\ b_1^\dagger \\ b_1 \\ \vdots \\ b_N^\dagger \\ b_N \end{pmatrix}, \tag{69}
$$

this can be easily done with the fermionic transformation $\tilde{\Gamma}^D = F^\dagger_{xp \to xx} \Gamma^D F_{xp \to xx}$. With this ordering we have

$$
\tilde{\Gamma}^D = \bigoplus_{k=1}^{N} \begin{pmatrix} v_k & 0 \\ 0 & 1-v_k \end{pmatrix}, \qquad \rho^D = \bigotimes_{k=1}^{N} \begin{pmatrix} v_k & 0 \\ 0 & 1-v_k \end{pmatrix}. \tag{70}
$$

To the tensor product of density matrices corresponds a direct sum of correlation matrices.

---

**F_utilities 3.6: Eigenvalues_of_rho($\Gamma$)$\to \vec{v}$**

This function returns the eigenvalues of the correlation matrix $\rho$ associated to the fermionic Gaussian state with Dirac correlation matrix $\Gamma$.

---

## 3.7 Reduced density matrix and tensor product of fermionic Gaussian states

Trying to define a partial trace over fermionic modes subspaces one soon faces what is often called the "partial trace ambiguity" [27, 28] (see also the end of appendix A.1.8).

In the case of fermionic Gaussian states, though, this is a much simpler task. Any reduced state formalism has to satisfy the simple criterion that the reduced density operator must contain all the information about the subsystem that can be obtained from the global state when measurements are performed only on the respective subsystem alone [27, 28].

With Wick's theorem in mind it is easy to see that the correlation matrix of the reduced state on the modes $i_1, \ldots, i_m$ is just the correlation matrix $\Gamma|_{\{i_1,\ldots,i_m\}}$ and that the reduced state of a f.g.s. is a f.g.s. too.

---

**F_utilities 3.7: Reduce_gamma($\Gamma$,m,i_1)$\rightarrow \Gamma|_{\{i_1,\ldots,i_m\}}$**

This function takes a Dirac correlation matrix $\Gamma$, a dimension of the partition $m$ and the initial site of the partition $i_1$ and return $\Gamma|_{\{i_1,\ldots,i_m\}}$, the reduced correlation matrix on the contiguous modes $\{i_1, \ldots, i_m\}$ where $i_m = i_1 + m$ and periodic boundary conditions are always assumed.

Examples: the green elements of the matrix $M_{6\times6}$ are the ones returned by the function calls.

$$\text{Reduce\_gamma}(M_{6\times6}, 2, 1) \rightarrow$$

$$\text{Reduce\_gamma}(M_{6\times6}, 2, 3) \rightarrow$$

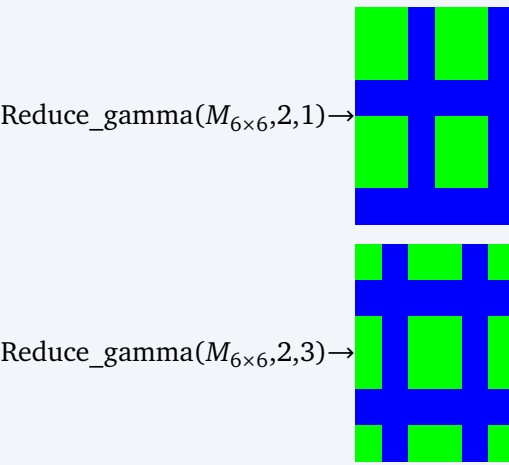

---

The correlation matrix $\Gamma_{A,B}$ of the tensor product of two f.g.s. $\Gamma_A$ and $\Gamma_B$ is obtained simply by collecting all the elements of $\Gamma_A$ and $\Gamma_B$ in a single well ordered correlation matrix $\Gamma_{A,B}$. The code for obtaining $\Gamma_{A,B}$ from $\Gamma_A$ and $\Gamma_B$ is

```
D_A = size(Gamma_A,1);
D_B = size(Gamma_B,1);
D   = D_A+D_B;

Gamma_AB = zeros(Complex{Float64}, D,D);
Gamma_AB = Inject_gamma(Gamma_AB,Gamma_A,1);
Gamma_AB = Inject_gamma(Gamma_AB,Gamma_B,D_A+1);
```

This code makes use of the function `Inject_gamma`.

---

**F_utilities  3.8: Inject_gamma($\Gamma$, $\Gamma_{inj}$, i)$\to \Gamma_{comp}$**

This function takes a $2N \times 2N$ matrix $\Gamma$ and a $2n \times 2n$ matrix $\Gamma_{inj}$ with $n \leq N$ and an injection point $i$. It returns the $2N \times 2N$ matrix $\Gamma_{comp}$ as shown in the pictures.

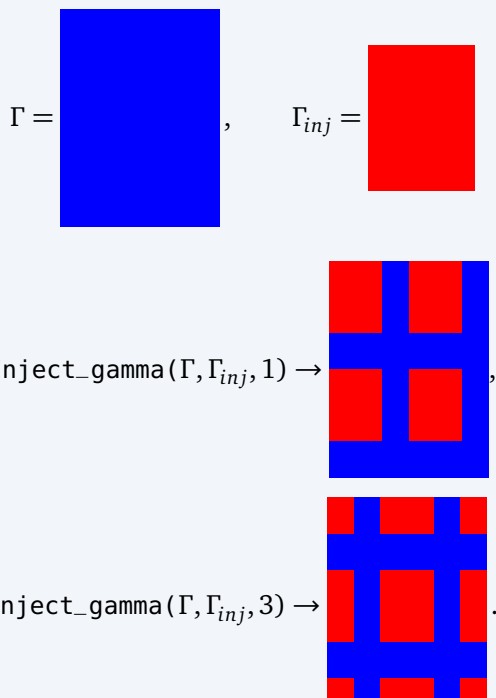

In the last example it is clear the systems behave with periodic boundary conditions. If $\Gamma$ is the correlation matrix of a f.g.s $\rho$ and $\Gamma_{inj}$ is the correlation matrix of a f.g.s $\rho_{inj}$ then $\Gamma_{comp}$ is the correlation matrix of the state $\text{Tr}_{\texttt{i,...,i+n-1}}[\rho] \otimes \rho_{inj}$.

---

It is clear that with the ordering (69), the tensor product of two f.g.s. corresponds to the direct sum of their correlation matrices

$$\rho_{A,B} = \rho_A \otimes \rho_B \ \to \ \tilde{\Gamma}_{A,B} = \tilde{\Gamma}_A \oplus \tilde{\Gamma}_B \,. \tag{71}$$

### 3.8   Correlation matrices of translational invariant states

We consider a state $\rho$ of a system of $N$ sites and all its reduced density matrices $\rho_A$, where $A$ is any possible set of sites of the system. We denote with $A+m$ the set of sites $A+m = \{j = i+m | i$ is a site of $A\}$, that is a translation of all site of $A$ by $m$ sites. When we will assume Periodic Boundary Conditions (PBC) we will allow for translations "over the border" of the system, in the sense that when $i+m > N$ (or $i+m < 1$) we will substitute it with $i+m \to \text{mod}(i+m-1, N+1)+1$. This is interpreted as connecting the first site with the last site of the system. Thus for PBC all translations are allowed. When we will assume Open Boundary Conditions (OBC) only translations within the system. This means that if $i \in A$ and $i+m > N$ (or $i+m < 1$), then the subset $A+m$ is not an allowed subset of sites.

A *translational invariant state* is a state such that for every $A$ we have $\rho_A = \rho_{A+m}$ for every allowed $m$.

This property easily translates to correlation matrices of states. For the two point correlators of a translational invariant state we have that $\langle a_j^\dagger a_l \rangle = \langle a_{j+m}^\dagger a_{l+m} \rangle$ and $\langle a_j a_l \rangle = \langle a_{j+m} a_{l+m} \rangle$ for every $m$. The specific correlator is thus individuated just by the difference of the sites of

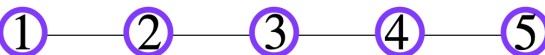

Figure 1: System with Open Boundary Conditions. If the state is translational invariant, then the reduced density matrix on sites 1 and 2 is the same as the one on sites 3 and 4, but is different from the one on sites 5 and 1

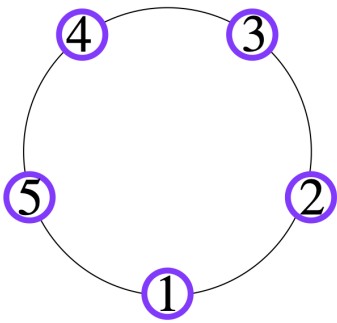

Figure 2: System with Periodic Boundary Conditions. If the state is translational invariant then the reduced density matrix on sites 1 and 2 or sites 3 and 4 or even sites 5 and 1 are all the same

the first and second operator $\Delta := l - j$ with $\Delta \in [-(N-1), N-1]$. Using this, we substitute $\langle a_j^\dagger a_l \rangle \to \langle a^\dagger a \rangle_\Delta$ and analogously $\langle a_j a_l \rangle \to \langle aa \rangle_\Delta$. We now focus on $\Gamma^{a^\dagger a}$, explicitly expressing it, we have

$$
\Gamma^{a^\dagger a}
\begin{pmatrix}
\langle a^\dagger a \rangle_0 & \langle a^\dagger a \rangle_1 & \langle a^\dagger a \rangle_2 & \dots & \langle a^\dagger a \rangle_{N-1} \\
\langle a^\dagger a \rangle_{-1} & \langle a^\dagger a \rangle_0 & \langle a^\dagger a \rangle_1 & \dots & \langle a^\dagger a \rangle_{N-2} \\
\langle a^\dagger a \rangle_{-2} & \langle a^\dagger a \rangle_{-1} & \langle a^\dagger a \rangle_0 & \dots & \langle a^\dagger a \rangle_{N-3} \\
\vdots & \vdots & \vdots & \ddots & \vdots \\
\langle a^\dagger a \rangle_{-(N-1)} & \langle a^\dagger a \rangle_{-(N-2)} & \langle a^\dagger a \rangle_{-(N-3)} & \dots & \langle a^\dagger a \rangle_0
\end{pmatrix}.
\tag{72}
$$

Matrix with this structure are called *Toeplitz matrices*.

If we further require the system to have PBC, we have that the parameter $\Delta$ is restricted to the range $[0, N-1]$. Consider for example the specific instance of the correlator $\langle a^\dagger a \rangle_{1-N} = \langle a_N^\dagger a_1 \rangle$, because of the translational invariance property of the system and because of the PBC we know that $\langle a_N^\dagger a_1 \rangle = \langle a_{N+1}^\dagger a_{1+1} \rangle = \langle a_1^\dagger a_2 \rangle = \langle a^\dagger a \rangle_1$ (see figure 2).

With PBC, $\Gamma^{a^\dagger a}$ has the form

$$
\Gamma^{a^\dagger a} =
\begin{pmatrix}
\langle a^\dagger a \rangle_0 & \langle a^\dagger a \rangle_1 & \langle a^\dagger a \rangle_2 & \dots & \langle a^\dagger a \rangle_{N-1} \\
\langle a^\dagger a \rangle_{N-1} & \langle a^\dagger a \rangle_0 & \langle a^\dagger a \rangle_1 & \dots & \langle a^\dagger a \rangle_{N-2} \\
\langle a^\dagger a \rangle_{N-2} & \langle a^\dagger a \rangle_{N-1} & \langle a^\dagger a \rangle_0 & \dots & \langle a^\dagger a \rangle_{N-3} \\
\vdots & \vdots & \vdots & \ddots & \vdots \\
\langle a^\dagger a \rangle_1 & \langle a^\dagger a \rangle_2 & \langle a^\dagger a \rangle_3 & \dots & \langle a^\dagger a \rangle_0
\end{pmatrix}.
\tag{73}
$$

We see that $\Gamma^{a^\dagger a}$ is the circulant matrix (see appendix A.1.4) characterised by the circulant vector $\langle \vec{a^\dagger a} \rangle = (\langle a^\dagger a \rangle_0, \langle a^\dagger a \rangle_1, \langle a^\dagger a \rangle_2, \dots, \langle a^\dagger a \rangle_{N-1})$. Following the same reasoning, we see that $\Gamma^{aa^\dagger}$ is a circulant matrix characterised by the circulant vector $\langle \vec{aa^\dagger} \rangle = (\langle aa^\dagger \rangle_0, \langle aa^\dagger \rangle_1, \langle aa^\dagger \rangle_2, \dots, \langle aa^\dagger \rangle_{N-1})$. Matrices $\Gamma^{aa}$ and $\Gamma^{a^\dagger a^\dagger}$ are circulant skew-symmetric matrices, often called *skew-circulant matrices*. If $N$ is even $\Gamma^{aa}$ and $\Gamma^{a^\dagger a^\dagger}$ are specified by the circulant vectors

$$\langle \vec{aa} \rangle = \left( \langle aa \rangle_0, \langle aa \rangle_1, \langle aa \rangle_2, \ldots, \langle aa \rangle_{\frac{N}{2}-1}, 0, -\langle aa \rangle_{\frac{N}{2}-1}, -\langle aa \rangle_{\frac{N}{2}-2}, \ldots, \langle aa \rangle_1 \right)$$

and

$$\langle \vec{a^\dagger a^\dagger} \rangle = \left( \langle a^\dagger a^\dagger \rangle_0, \langle a^\dagger a^\dagger \rangle_1, \langle a^\dagger a^\dagger \rangle_2, \ldots, \langle a^\dagger a^\dagger \rangle_{\frac{N}{2}-1}, 0, -\langle a^\dagger a^\dagger \rangle_{\frac{N}{2}-1}, -\langle a^\dagger a^\dagger \rangle_{\frac{N}{2}-2}, \ldots, \langle a^\dagger a^\dagger \rangle_1 \right).$$

If $N$ is odd $\Gamma^{aa}$ and $\Gamma^{a^\dagger a^\dagger}$ are specified by the circulant vectors

$$\langle \vec{aa} \rangle = \left( \langle aa \rangle_0, \langle aa \rangle_1, \langle aa \rangle_2, \ldots, \langle aa \rangle_{\frac{N-1}{2}}, -\langle aa \rangle_{\frac{N-1}{2}-1}, -\langle aa \rangle_{\frac{N-1}{2}-2}, \ldots, \langle aa \rangle_1 \right)$$

and

$$\langle \vec{a^\dagger a^\dagger} \rangle = \left( \langle a^\dagger a^\dagger \rangle_0, \langle a^\dagger a^\dagger \rangle_1, \langle a^\dagger a^\dagger \rangle_2, \ldots, \langle a^\dagger a^\dagger \rangle_{\frac{N-1}{2}}, -\langle a^\dagger a^\dagger \rangle_{\frac{N-1}{2}-1}, -\langle a^\dagger a^\dagger \rangle_{\frac{N-1}{2}-2}, \ldots, \langle a^\dagger a^\dagger \rangle_1 \right).$$

**Eigenvalues using the properties of circulant-matrices** In appendix A.1.4 we show the general form of the eigenvalues of a circulant matrix. For $\Gamma^{aa}, \Gamma^{a^\dagger a^\dagger}$ matrices $\Gamma^{a^\dagger a}, \Gamma^{aa^\dagger}$ we have that their respective eigenvalues $\lambda_k^{a^\dagger a}, \lambda_k^{aa^\dagger}$ are

$$\lambda_k^{a^\dagger a} = \sum_{\Delta=0}^{N-1} e^{i \frac{2\pi}{N} \Delta k} \langle a^\dagger a \rangle_\Delta, \qquad\qquad \lambda_k^{aa^\dagger} = \sum_{\Delta=0}^{N-1} e^{i \frac{2\pi}{N} \Delta k} \langle aa^\dagger \rangle_\Delta. \tag{74}$$

For matrices $\Gamma^{aa}, \Gamma^{a^\dagger a^\dagger}$ we have that their respective eigenvalues $\lambda_k^{aa}, \lambda_k^{a^\dagger a^\dagger}$ are

$$\lambda_k^{aa} = \begin{cases} 2 \sum_{\Delta=0}^{\frac{N}{2}-1} e^{i \frac{2\pi}{N} k \Delta} \langle aa \rangle_\Delta & \text{if } N \text{ even} \\ (1 + e^{-i \frac{\pi}{N}}) \sum_{\Delta=0}^{\frac{N}{2}-1} e^{i \frac{2\pi}{N} k \Delta} \langle aa \rangle_\Delta & \text{if } N \text{ odd} \end{cases},$$

$$\lambda_k^{a^\dagger a^\dagger} = \begin{cases} 2 \sum_{\Delta=0}^{\frac{N}{2}-1} e^{i \frac{2\pi}{N} k \Delta} \langle a^\dagger a^\dagger \rangle_\Delta & \text{if } N \text{ even} \\ (1 + e^{-i \frac{\pi}{N}}) \sum_{\Delta=0}^{\frac{N}{2}-1} e^{i \frac{2\pi}{N} k \Delta} \langle a^\dagger a^\dagger \rangle_\Delta & \text{if } N \text{ odd} \end{cases}. \tag{75}$$

We notice that the eigenvalues of $\Gamma^{aa}, \Gamma^{a^\dagger a^\dagger}$ comes in pairs $\lambda_k^{aa} = -\lambda_{k+\lceil \frac{N}{2} \rceil}^{aa}$ and $\lambda_k^{a^\dagger a^\dagger} = -\lambda_{k+\lceil \frac{N}{2} \rceil}^{a^\dagger a^\dagger}$ as expected from the property of skew-symmetric matrices (see appendix A.1.5).

**Eigenvalues using the Fourier transform on a linear lattice** We introduce the Fourier transforms on a linear lattice

$$f_k = \frac{1}{\sqrt{N}} \sum_{j=1}^{N} e^{i \frac{2\pi}{N} kj} a_j, \qquad f_k^\dagger = \frac{1}{\sqrt{N}} \sum_{j=1}^{N} e^{-i \frac{2\pi}{N} kj} a_j^\dagger, \tag{76}$$

with inverse transformations

$$a_j = \frac{1}{\sqrt{N}} \sum_{k=1}^{N} e^{-i \frac{2\pi}{N} kj} f_k, \qquad a_j^\dagger = \frac{1}{\sqrt{N}} \sum_{k=1}^{N} e^{i \frac{2\pi}{N} kj} f_k^\dagger. \tag{77}$$

It is easy to see that the Fourier modes $\{f_k, f_k^\dagger\}_k$ obey to the CAR and are valid Dirac operators.

Now we perform the substitutions (5.1) in the expression of $\Gamma^{a^\dagger a}$ and we further exploit the translational invariance ($\langle a^\dagger a \rangle \Delta = \frac{1}{N} \sum_{j=1}^{N} \langle a_j^\dagger a_{j+\Delta} \rangle$) to obtain

$$\langle a^\dagger a \rangle_\Delta = \frac{1}{N^2} \sum_{j}^{N} \sum_{k,k'} e^{i \frac{2\pi}{N} k' \Delta} e^{i \frac{2\pi}{N} (k-k')j} \langle f_k^\dagger f_{k'} \rangle. \tag{78}$$

Collecting the Kronecker delta (see appendix A.3.4) we can express the elements of $\Gamma^{a^\dagger a}$ as

$$\langle a^\dagger a \rangle_\Delta = \frac{1}{N} \sum_{k=1}^{N} e^{-i\frac{2\pi}{N}k\Delta} \langle f_k^\dagger f_k \rangle. \tag{79}$$

With the same procedure we obtain

$$\langle a^\dagger a \rangle_\Delta = \frac{1}{N} \sum_{k=1}^{N} e^{-i\frac{2\pi}{N}k\Delta} \langle f_k^\dagger f_k \rangle, \qquad \langle aa^\dagger \rangle_\Delta = \frac{1}{N} \sum_{k=1}^{N} e^{-i\frac{2\pi}{N}k\Delta} \langle f_k f_k^\dagger \rangle,$$

$$\langle aa \rangle_\Delta = \frac{1}{N} \sum_{k=1}^{N} e^{-i\frac{2\pi}{N}k\Delta} \langle f_k f_{N-k} \rangle, \qquad \langle a^\dagger a^\dagger \rangle_\Delta = \frac{1}{N} \sum_{k=1}^{N} e^{-i\frac{2\pi}{N}k\Delta} \langle f_{N-k}^\dagger f_k^\dagger \rangle, \tag{80}$$

with inverse transformations

$$\langle f_k^\dagger f_k \rangle = \sum_{\Delta=1}^{N} e^{i\frac{2\pi}{N}k\Delta} \langle a^\dagger a \rangle_\Delta, \qquad \langle f_k f_k^\dagger \rangle = \sum_{\Delta=1}^{N} e^{i\frac{2\pi}{N}k\Delta} \langle aa^\dagger \rangle_\Delta,$$

$$\langle f_k f_{N-k} \rangle = \sum_{\Delta=1}^{N} e^{i\frac{2\pi}{N}k\Delta} \langle a^\dagger a \rangle_\Delta, \qquad \langle f_k^\dagger f_{N-k}^\dagger \rangle = \sum_{\Delta=1}^{N} e^{-i\frac{2\pi}{N}k\Delta} \langle aa^\dagger \rangle_\Delta. \tag{81}$$

We can easily identify

$$\lambda_k^{a^\dagger a} = \langle f_k^\dagger f_k \rangle, \qquad \lambda_k^{aa^\dagger} = \langle f_k f_k^\dagger \rangle,$$

$$\lambda_k^{aa} = \langle f_k f_{N-k} \rangle, \qquad \lambda_k^{a^\dagger a^\dagger} = \langle f_k^\dagger f_{N-k}^\dagger \rangle. \tag{82}$$

At last, we note that the Fourier transform does not mix creation and annihilation operators and can be implemented directly on the vector of Dirac operators $\vec{\alpha}$ with the fermionic transformation $U_\omega$ that has the block diagonal form

$$U_\omega = \begin{pmatrix} W & 0 \\ 0 & \bar{W} \end{pmatrix}, \tag{83}$$

where $W$ is the matrix implementing the discrete Fourier transform (see appendix A.1.4) and it acts separately on the creation and annihilation operators sectors of $\vec{\alpha}$.

For an example of diagonalisation of translational invariant matrices see e.g. subsection 4

> **F_utilities 3.9: Build_Fourier_matrix($N$) $\to U_\omega$**
>
> This function returns the fermionic transformation $U_\omega$ for a system of $N$ sites.

## 3.9 Product Rule

It will result useful to compute the product $\rho = \rho_1 \rho_2$ of the density matrices of two fermionic Gaussian states. We observe that the commutator of two quadratic terms of Majorana operators $\vec{r}$ is always again a quadratic operator or zero

$$[r_i r_j, r_k r_l] = \delta_{k,i} r_l r_j + \delta_{k,j} r_i r_l - \delta_{i,l} r_k r_j - \delta_{l,j} r_i r_k. \tag{84}$$

This is also valid for Dirac operators. We say that the commutator of two monomials of Dirac operators of degree at most 2 is a polynomial of Dirac operators of degree at most 2. Using

this observation together with the Baker-Campbell-Hausdorff formula ( equation B.C.H.0 in appendix A.3.4), it is easy to see that $\rho$, the product of two f.g.s., is always a f.g.s.

$$\rho = \frac{e^{-\hat{H}}}{Z}, \tag{85}$$

with $\hat{H}$ given by the B.C.H.0.

It is possible to derive the covariance matrix $\gamma$ of $\rho$ directly from the covariance matrices $\gamma_1$ and $\gamma_2$ of the states $\rho_1$ and $\rho_2$. This formula appears in [37] where a more detailed description, considering even pathological cases, is given. If we assume that $\mathbb{I} - \gamma_1$ and $\mathbb{I} - \gamma_2$ are invertible then we have

$$\gamma = \mathbb{I} - (\mathbb{I} - \gamma_2) \frac{1}{\mathbb{I} + \gamma_1 \gamma_2} (\mathbb{I} - \gamma_1). \tag{86}$$

> **F_utilities  3.10: Product($\Gamma_1, \Gamma_2$) $\to \Gamma$**
>
> This function returns the correlation matrix $\Gamma$ corresponding to the f.g.s $\rho = \rho_1 \rho_2$, where $\rho_1$ and $\rho_2$ are characterised by the correlation matrices $\Gamma_1$ and $\Gamma_2$.

## 3.10  Information measures

**Von Neumann Entropies**   The von Neumann entropy of a quantum state described by the density matrix $\rho$ is

$$S(\rho) = -\mathrm{Tr}\left[\rho \ln(\rho)\right]. \tag{87}$$

In terms of the eigenvalues $\lambda$ of $\rho$, the von Neumann entropy reads as

$$S(\rho) = -\sum_{\lambda} \lambda \ln(\lambda). \tag{88}$$

If $\rho$ is a f.g.s. of a system with $N$ sites, since the von Neumann entropy is invariant under unitary transformation of the state, substituting in (87) the product form (70) and using the fact that the von Neumann entropy is additive for product states, the von Neumann entropy becomes a function of the eigenvalues $\nu_i$ of the correlation matrix $\Gamma$ and it is the sum of just $2N$ terms

$$S(\Gamma) \equiv S(\rho) = -\sum_{k=1}^{N} \left[\nu_k \ln(\nu_k) + (1 - \nu_k) \ln(1 - \nu_k)\right]. \tag{89}$$

> **F_utilities  3.11: VN_entropy($\Gamma$) $\to S$**
>
> This function returns $S$, the `Float64` value of the von Neumann Entropy of the state described by the Dirac correlation matrix $\Gamma$.

**Purity**   A state is pure if its correlation matrix $\Gamma$ is such that [38]

$$\Gamma^2 = \Gamma, \tag{90}$$

or, equivalently,

$$\gamma^2 = -\mathbb{I}. \tag{91}$$

The purity of a state $\rho$ is defined as

$$\mathrm{Purity}(\rho) \equiv Tr\left[\rho^2\right]. \tag{92}$$

We have that:

$$\text{Purity}(\rho) = \prod_{i=1}^{N-1} \frac{1}{2}\left(1 + \tanh(\epsilon_i)^2\right), \tag{93}$$

$$\text{Purity}(\Gamma) = \prod_{i=1}^{N-1} \left(2\left(\nu_i - 1\right)\nu_i + 1\right), \tag{94}$$

$$\text{Purity}(\gamma) = \prod_{i=1}^{N-1} \left(2\eta_i^2 + \frac{1}{2}\right), \tag{95}$$

the value of the purity is the same if computed with any of these equations. For more details see appendix A.1.2.

> **F_utilities 3.12: Purity($\Gamma$)$\to p$**
>
> This function returns $p$ the purity of the fermionic Gaussian state with Dirac correlation matrix $\Gamma$.

**Entanglement Contour**   In 2014 Chen and Vidal [39] introduced the entanglement contour "a tool for identifyng which real-space degrees of freedom contribute, and how much, to the entanglement of a region A with the rest of the system B". We consider the state of a system on a chain of $N$ sites, we divide the chain into two complementary partitions, partition $A$ and partition $B$. Now suppose partitions $A$ and $B$ are entangled and that there exists a measure $\mathcal{E}(A,B)$ that quantifies the amount of entanglement between $A$ and $B$. The entanglement contour $c_A(i)$ of partition $A$ tells us how much each site $i$ of partition $A$ contributes to the total amount of entanglement betwen $A$ and $B$. Furthermore summing $c_A(i)$ over all the sites of $A$ one should obtain exactly $\mathcal{E}(A,B)$.

Chen and Vidal state five reasonable properties that define when a function is a contour function. In the same paper they show that these five properties do not identify a unique contour function, but instead a class of functions. Here we are going to focus on a specific entanglement contour defined for fermionic Gaussian states. First of all we restrict to pure states. For a pure state, it is known that a good measure of entanglement between two complementary partition $A$ and $B$ is the entanglement entropy, that is the von Neumann entropy $\mathcal{E}(A,B) = S(A)$ of the reduced state on $A$.

We consider an Hilbert space $\mathcal{H}$ divided in the two complementary partitions $\mathcal{H} = \mathcal{H}_A \otimes \mathcal{H}_B$, each partition with $N_A$ and $N_B$ sites respectively. The Schmidt decomposition (see section of a pure state $|\psi^{A,B}\rangle$ in $\mathcal{H}$ is

$$|\psi^{A,B}\rangle = \sum_i \sqrt{p_i}|\psi_i^A\rangle \otimes |\psi_i^B\rangle, \tag{96}$$

with $p_i \geq 0$, $\sum_i p_i = 1$ and

$$\rho^A \equiv Tr_B\left[|\psi^{A,B}\rangle\langle\psi^{A,B}|\right] = \sum_i p_i|\psi_i^A\rangle\langle\psi_i^A|. \tag{97}$$

The entanglement entropy for this choice of partition is thus $S(A) = -\sum_i p_i ln(p_i)$.
Factorising the Hilbert space $\mathcal{H}_A$ in its tensor product structure $\mathcal{H}_A = \bigotimes_{j\in A} \mathcal{H}_j$, we individuate in each local Hilbert space $\mathcal{H}_j$ a site of the partition $A$. We remind that $\rho^A$ cannot be expressed as a product state over this factorisation of $\mathcal{H}_A$ and that the von Neumann entropy is not additive. Thus the von Neumann entropy computed on each site is not a good entanglement contour function.

We know from 3.7 that $\rho_A$ is a f.g.s., thus we can express the entanglement entropy $S(A)$ as the sum of the von Neumman entropy of each mode in $A$,

$$S(A) = \sum_{k=1}^{N_A} S_k = -\sum_{k=1}^{N_A} \left[ \nu_k \ln(\nu_k) + (1 - \nu_k) \ln(1 - \nu_k) \right]. \tag{98}$$

Each mode $k$, associated to the Dirac operators $\beta_k = b_k^\dagger$, $\beta_{k+N_A} = b_k$, is connected to the real space modes associated the Dirac operators $\alpha_i = a_i^\dagger$, $\alpha_{i+N_A} = a_i$ by the fermionic transformation $U$ such that

$$\beta_k = \sum_{i=1}^{N_A} U_{k,i} \alpha_i. \tag{99}$$

We want to use this equation to find how much a fixed mode $k$ contributes to a fixed site $i$. We call this contribution $p_i(k)$ and we define it as

$$p_i(k) := \frac{1}{2} \left[ |U_{k,i}|^2 + |U_{k+N_A,i+N_A}|^2 + |U_{k,i+N_A}|^2 + |U_{k+N_A,i}|^2 \right]. \tag{100}$$

The entanglement contour for partition $A$ is thus defined as

$$c_A(i) := \sum_{k=1}^{N_A} p_i(k) S_k. \tag{101}$$

It is easy to see that each of the $p_i(k)$ is positive and that

$$\sum_{k=1}^{N_A} p_i(k) = 1, \tag{102}$$

as

$$\sum_{k=1}^{N_A} p_i(k) = \frac{1}{2} \left[ \sum_{l=1}^{2N_A} U_{i,l} U_{l,i}^* + U_{i+N_A,l} U_{l,i+N_A}^* \right] = \frac{1}{2} \left( (UU^\dagger)_{i,i} + (UU^\dagger)_{i+N_A,i+N_A} \right) = 1,$$

since $U$ is unitary. Thus one has the desired property

$$\sum_{i=1}^{N_A} c_A(i) = S(A). \tag{103}$$

> **F_utilities  3.13: Contour$(\Gamma_A) \to \vec{c}_A$**
>
> This function returns the vector $\vec{c}_{Ai} = c_A(i)$ of the entanglement contour of the correlation matrix $\Gamma_A$.

## 3.11  Examples

We will use the function

> **F_utilities  3.14: Random_NNHamiltonian(N)$\to H$**
>
> Generates a random f.q.h. Hamiltonian for a system of $N$ sites with just nearest neighbour interactions.

SciPost Phys. Lect. Notes 54 (2022)

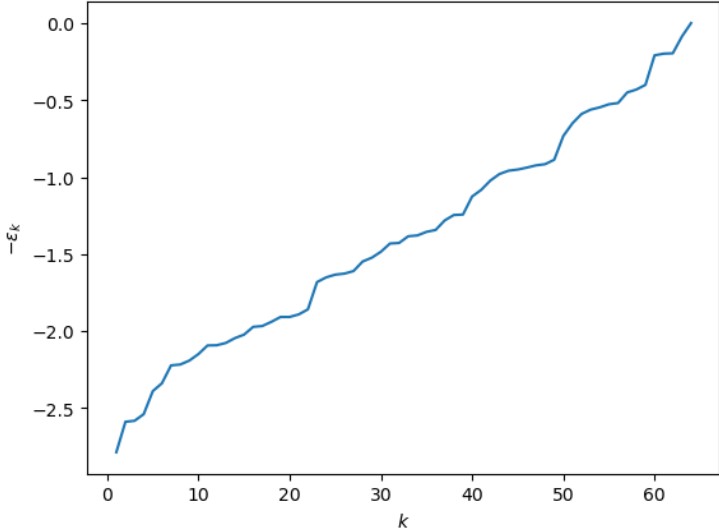

Figure 3: Output of examples 3.11. In the plot are represented the eigenvalues of a random nearest-neighbour Hamiltonian computed with the code `Random_NNHamiltonian(N)` for a system with $N = 64$ sites.

**Computing the energies of H** In this program we compute the energies $\epsilon_k$ of a random nearest neighbours Hamiltonian on a linear lattice of $N = 64$ sites generated with the function `Random_NNHamiltonian(64)`. The program generates the output figure 3 .

```julia
using F_utilities;
using PyPlot;
using LinearAlgebra;

const Fu = F_utilities;

N = 64;

#Generate and diagonalise the hamiltonian
H = Fu.Random_NNhamiltonian(N)
H_D, U_D = Fu.Diag_h(H)

#Print the energy modes epsilon_k
figure("Energies")
plot(1:N,diag(H_D)[1:N])
xlabel(L"$k$")
ylabel(L"$-\epsilon_k$")
```

**Computing the entanglement contour of a partition of a ground state** In this program we compute the entanglement contour and the entropy of a partition of $N_A = 32$ sites in the bulk of the ground state of a random nearest neighbours Hamiltonian on a linear lattice of $N = 64$ sites. The program generates the output figure 4.

```julia
using F_utilities;
using PyPlot;
using LinearAlgebra;
```

```
const Fu = F_utilities;

N = 64;
H = Fu.Random_NNhamiltonian(N);
H_D, U_D = Fu.Diag_h(H);

Gamma = Fu.GS_Gamma(H_D, U_D);
println("The energy of the ground state is: ", Fu.Energy_fermions(Gamma,
    ↪ H_D, U_D));

N_A = 32;
#I consider the reduced state over the sites 17,18,...,48
Gamma_A = Fu.Reduce_gamma(Gamma,N_A,17);
#I compute the entangement entropy
S_A = Fu.VN_entropy(Gamma_A);
#I compute the contour of partition A
c_A = Fu.Contour(Gamma_A);

lbl_title   = string(L"$S(A) = $", S_A);
lbl_legend  = string(L"$\sum_{i=1}^{N_A} c_A(i) = $", sum(c_A));
figure("Contour of A")
title(lbl_title)
plot(1:N_A, c_A, marker="o", label=lbl_legend);
xlabel("i")
ylabel(L"$c_A(i)$")
legend();
```

Output:

The energy of the ground state is:   -83.1750144099933

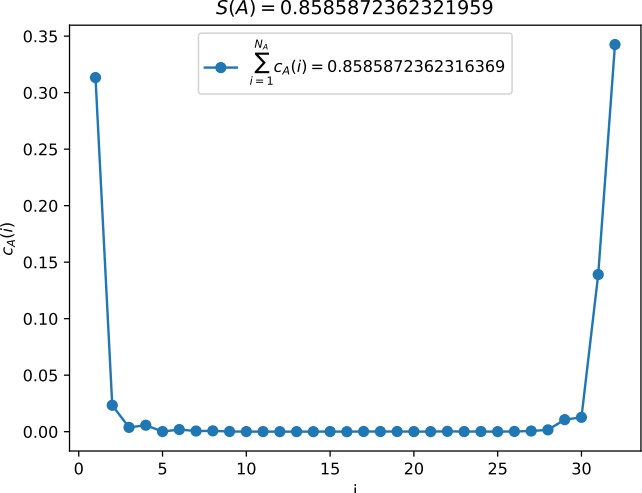

Figure 4: Output of examples 3.11. In the plot it is represented the entanglement contour of the reduced density matrix of the first 32 contiguous site of a linear chain, when the global state is the ground state of a nearest-neighbour Hamiltonian on a linear chain of $N = 64$ sites. Note how for a nearest neighbour Hamiltonian, the contour is higher on the boundary of the partition.

### 3.12 Time Evolution

We learned about Hamiltonians and states. Now it is time to put these two ingredients together and finally talk about the unitary evolution of fermionic Gaussian states.

We start stating that *the space of fermionic Gaussian states is closed under evolution induced by fermionic quadratic Hamiltonians*.

The best way for seeing this is using the Majorana operators representation. We consider a general f.q.h. $\hat{H}$ and a generic f.g.s. $\rho = \frac{e^{-\hat{H}_\rho}}{Z}$ with both $\hat{H}$ and $\hat{H}_\rho$ of the form (18). Using standard notation we call $\rho(t)$ the state $\rho$ at time $t$ defined as

$$\rho(t) := e^{-it\hat{H}}\rho e^{it\hat{H}} = \frac{e^{-it\hat{H}}e^{-\hat{H}_\rho}e^{it\hat{H}}}{Z}. \tag{104}$$

As already observed in subsection 3.9, the commutator of two quadratic monomial of Dirac operators is a polynomial at most quadratic in Dirac operators. Using this observation together with the Baker-Campbell-Hausdorff formula ( equation B.C.H.0 in appendix A.3.4), it is easy to see that $\rho(t)$ has the form

$$\rho(t) = \frac{e^{-\hat{H}_{\rho(t)}}}{Z}, \tag{105}$$

with $\hat{H}_{\rho(t)}$ a fermionic quadratic Hamiltonian. Thus $\rho(t)$ is again a Gaussian state proving that the space of Gaussian states is closed under evolution induced by fermionic quadratic Hamiltonians.

We will now compute an explicit formula for the time evolution of the correlation matrix $\Gamma(t)$ of the f.g.s. state $\rho(t)$. The first step is computing the time evolution of the creation and annihilation operators in the Heisenberg picture. We denote with $\vec{\beta}$ the vector of Dirac operators that diagonalises $H$. The annihilation and creation operators $b_k$ and $b_k^\dagger$ evolved at time $t$ are (see appendix A.1.3)

$$b_k(t) = e^{-i\hat{H}t}b_k e^{i\hat{H}t} = e^{-i2\epsilon_k t}b_k, \tag{106}$$

$$b_k^\dagger(t) = e^{-i\hat{H}t}b_k^\dagger e^{i\hat{H}t} = e^{i2\epsilon_k t}b_k^\dagger. \tag{107}$$

In compact form this can be written as

$$\vec{\beta}(t) = e^{i2H_D t}\vec{\beta}. \tag{108}$$

It is easy now to compute the time evolution of the correlators $\langle \vec{\beta}\vec{\beta}^\dagger \rangle$

$$\langle \vec{\beta}(t)\vec{\beta}^\dagger(t) \rangle = \langle e^{i2H_D t}\vec{\beta}\vec{\beta}^\dagger e^{-i2H_D t} \rangle. \tag{109}$$

Thus, if $U$ is the fermionic transformation such that $\vec{\beta} = U\vec{\alpha}$, the fermionic transformation implementing the time evolution of $\vec{\alpha}$ is $U^\dagger e^{i2H_D t}U = e^{i2Ht}$. We finally obtain that the correlation matrix $\Gamma$ evolves with $H$ as

$$\Gamma(t) = e^{i2Ht}\Gamma e^{-i2Ht}. \tag{110}$$

---

> **F_utilities 3.15: Evolve(Γ, H_D, U, t)→ $\Gamma(t)$**
>
> This function returns the correlation matrix $\Gamma$ evolved at time $t$ with $H$. Matrices $H_D$ and $U$ are the output of `Diag_h(H)`.

# 4 Hopping model

We consider the translational invariant hopping Hamiltonian for a system of $N$ sites

$$\hat{H} = \sum_{i=1}^{N-1}[a_i^\dagger a_{i+1} - a_i a_{i+1}^\dagger] + \delta[a_N^\dagger a_1 - a_N a_1^\dagger], \tag{111}$$

with $\delta = 1$ for periodic boundary conditions and $\delta = 0$ for open boundary conditions.
The compact form (16) of $\hat{H}$ is specified by the two circulant matrices (see A.1.4)

$$A = \begin{pmatrix} 0 & \frac{1}{2} & 0 & \dots & 0 & \delta\frac{1}{2} \\ \frac{1}{2} & 0 & \frac{1}{2} & 0 & \dots & 0 \\ 0 & \frac{1}{2} & 0 & \frac{1}{2} & 0 & \dots \\ \vdots & \vdots & \ddots & \ddots & \ddots & \vdots \\ \delta\frac{1}{2} & 0 & 0 & \dots & \frac{1}{2} & 0 \end{pmatrix}, \qquad B = \begin{pmatrix} 0 & \dots & 0 \\ \vdots & \ddots & \vdots \\ 0 & \dots & 0 \end{pmatrix}. \tag{112}$$

As seen in subsection 3.8 and appendix A.1.4) we know that $H$ is diagonalised with a Fourier transformation. Indeed, if we express the hopping Hamiltonian (111) in terms of the Fourier modes (5.1) we obtain

$$\hat{H} = \sum_{k=1}^{N} \phi_k(f_k^\dagger f_k - f_k f_k^\dagger), \tag{113}$$

where

$$\phi_k = \cos(\frac{2\pi}{N}k). \tag{114}$$

The fermionic transformation that diagonalises the Hamiltonian is $U_\omega$ as defined in (83).

> **F_utilities  4.1: Build_hopping_Hamiltonian($N$, PBC=true)$\rightarrow H$**
>
> This functions return the Hamiltonian $H$ of dimension $2N \times 2N$ for the hopping model.
> If PBC=false it return the hopping Hamiltonian with open boundary conditions

## 4.1 Numerical diagonalisation

In the following code we show how to initialise and diagonalise the hopping Hamiltonian using functions of F_utilities. We perform these calculations using two different methods. The energies computed with both methods are reported in figure 5, the ground states correlations matrices are identical.

```
using F_utilities;
using PyPlot;
using LinearAlgebra;

const Fu = F_utilities;

N =127;

H   = Fu.Build_hopping_hamiltonian(N,PBC=true);

U_omega = Fu.Build_Fourier_matrix(N);
D_omega = U_omega'*H*U_omega;
```

```
D,U = Fu.Diag_h(H);

figure("Energies")
plot(diag(real.(D_omega))[(N+1):(2*N)],label="Method Fourier");
plot(real.(diag(D))[(N+1):(2*N)], label="Method Diag_h");
xlabel(L"$k$");
ylabel(L"$\epsilon_k$");
legend();

Gamma_omega = Fu.GS_gamma(D_omega,U_omega);
Gamma       = Fu.GS_gamma(D,U);
println("")
println("Energy GS Method Fourier:         ",Fu.Energy(Gamma_omega,(D_omega,
    ↪ U_omega)))
println("En GS Method Diag_h:              ",Fu.Energy(Gamma,(D,U)))
```

Output:
Energy GS Method Fourier:  -80.85277253991693
En GS Method Diag_h:  -80.85277253997737

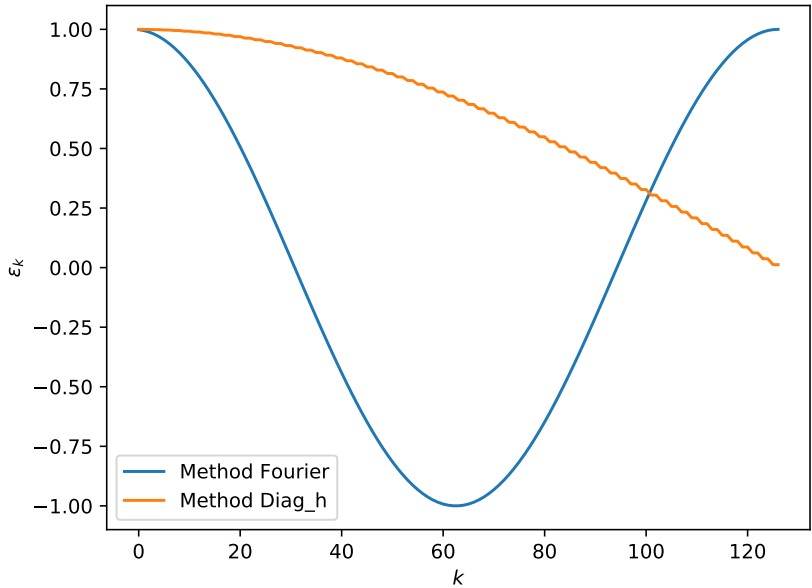

Figure 5: In this example we diagonalised the Hamiltonian with two different methods. Using the Fourier transform method we obtain the energies specified in (114). These energies are both positive and negative. Using the `Diag_h` method of `F_utilities` we obtain just positive energies. The difference in the diagonal energies comes from the fact that `Diag_h`, for every eigenmode with negative energy, substitutes creation and annihilation operators in order to redefine the energy as positive, and then reorders the modes in order to have the energies in descending order. If we diagonalise with the Fourier transform then the ground state is obtained filling all the modes with negative energy. If we diagonalise with `Diag_h` then the ground state corresponds to the empty state.

For the numerical diagonalisation of the Hamiltonian we used two methods, the analytical one using the Fourier modes, and the numerical one introduced in the previous subsection. These methods return the hopping Hamiltonian in the diagonal forms

$$\hat{H}_{\omega} = \sum_{k=1}^{N} \phi_k (f_k^{\dagger} f_k - f_k f_k^{\dagger}), \qquad \hat{H} = \sum_{k=1}^{N} \epsilon_k (b_k^{\dagger} b_k - b_k b_k^{\dagger}), \tag{115}$$

where the differences in the energies are due to the fact that `Diag_h` considers all the energies $\epsilon_k$ positive, thus defines $b_k = f_k^{\dagger}$ and $b_k^{\dagger} = f_k$ for each $k$ such that $\phi_k < 0$ (that corresponds to flipping the sign of $\phi_k$ when it is negative, such that the corresponding $\epsilon_k = -\phi_k$), and then reorder the modes such that to modes with smaller $k$ correspond biggest energies.

## 4.2  Time Evolution

For the hopping model we analytically obtained the fermionic transformation $U_{\omega}$ that diagonalises the Hamiltonian. This allows us to give an analytical expression for the time evolution of the correlation matrix.

Expressing the correlation matrix $\Gamma$ in terms of the operators $\vec{\phi}$ and computing the time evolution with the diagonal Hamiltonian (113) we obtain

$$\langle a_l^{\dagger} a_m \rangle(t) = \frac{1}{N} \sum_{k,k'} \sum_{x,y} e^{i2(\phi_k - \phi_{k'})t} e^{i\frac{2\pi}{N}(k(l-x) - k'(m-y))} \langle a_x^{\dagger} a_y \rangle, \tag{116}$$

$$\langle a_l a_m \rangle(t) = \frac{1}{N} \sum_{k,k'} \sum_{x,y} e^{-i2(\phi_k + \phi_{k'})t} e^{-i\frac{2\pi}{N}(k(l-x) + k'(m-y))} \langle a_x a_y \rangle. \tag{117}$$

Because of the block diagonal structure of $U_{\omega}$ there is not mixing of the two types of correlators during the evolution of the correlation matrix.

**Time evolution of translational invariant states**   Let us consider a translational invariant state $\Gamma$. In subsection 3.8 we expressed $\Gamma$ in terms of the Fourier modes $f_k^{\dagger}, f_k$. Using the diagonal form (115) of the Hopping Hamiltonian to compute the time evolution of the correlators of $\Gamma$ expressed as in (80) we have that $\Gamma$ evolves as

$$\langle a^{\dagger} a \rangle_{\Delta}(t) = \frac{1}{N} \sum_{k=1}^{N} e^{-i\frac{2\pi}{N}k\Delta} \langle f_k^{\dagger} f_k \rangle, \qquad \langle a a^{\dagger} \rangle_{\Delta}(t) = \frac{1}{N} \sum_{k=1}^{N} e^{-i\frac{2\pi}{N}k\Delta} \langle f_k f_k^{\dagger} \rangle,$$

$$\langle a a \rangle_{\Delta}(t) = \frac{1}{N} \sum_{k=1}^{N} e^{i4\phi(k)t} e^{-i\frac{2\pi}{N}k\Delta} \langle f_k f_{N-k} \rangle, \quad \langle a^{\dagger} a^{\dagger} \rangle_{\Delta}(t) = \frac{1}{N} \sum_{k=1}^{N} e^{-i4\phi(k)t} e^{-i\frac{2\pi}{N}k\Delta} \langle f_{N-k}^{\dagger} f_k^{\dagger} \rangle,$$

$$\tag{118}$$

in the following program we numerically compute the time evolution induced by a hopping Hamiltonian on a random translational invariant gaussian state with exponentially decaying correlation functions. We consider a linear system of $N = 50$ sites and evolve it for $N_{\text{steps}} = 100$ steps of $\delta_{steps} = 0.1$. The program generates the output figures 6 and 7.

```
using F_utilities;
using PyPlot;
using LinearAlgebra;

const Fu = F_utilities;
```

```julia
N=50;
N_steps = 100;
delta_steps = 0.1;

#Build the circulant vector for the adaa part of the Gamma with
    ↪ exponential decaying correlations
adaa = zeros(Complex{Float64},N);#
for i=1:div(N,2)
    adaa[i] = exp(-i*0.15)*(rand()+im*rand())
end
adaa[((div(N,2))+1):N]= reverse(adaa[1:div(N,2)]);
#Build the translational invariant adaa part of the Gamma
Gamma_adaa = Fu.Circulant(adaa);
Gamma_adaa = (Gamma_adaa+Gamma_adaa')/2.

#Build the circulant vector for the aa part of the Gamma
aa = zeros(Complex{Float64},N);
aa[2] = rand()+im*rand();
aa[3] = rand()+im*rand();
#Build the translational invariant aa part of the Gamma
Gamma_aa = Fu.Circulant(aa)
Gamma_aa = (Gamma_aa-transpose(Gamma_aa))/2.;

#Build the translational invariant Gamma
Gamma= zeros(Complex{Float64}, 2N,2N);
Gamma[(1:N),(1:N)]          = Gamma_adaa;
Gamma[(1:N).+N,(1:N).+N]    = (I-Gamma_adaa)';
Gamma[(1:N).+N,(1:N)]       = Gamma_aa;
Gamma[(1:N),(1:N).+N]       = -conj(Gamma_aa);
Fu.Print_complex_matrix("Gamma",Gamma)

H   = Fu.Build_hopping_hamiltonian(N,PBC=true);
D,U = Fu.Diag_h(H);

Gamma_evolved   = copy(Gamma);
adaa        = zeros(Complex{Float64}, N_steps)
aa          = similar(adaa);

#Start the time evolution cycle
#at each cycle it saves the value of two correalotors
adaa[1]     = Gamma_evolved[1,2];
aa[1]       = Gamma_evolved[N+1,2];
for t=2:N_steps
    global Gamma_evolved = Fu.Evolve(Gamma_evolved,(D,U),delta_steps);
    adaa[t]     = Gamma_evolved[1,2];
    aa[t]       = Gamma_evolved[N+1,2];
end
```

```
figure("Evolutions")
plot(real.(adaa), color="black", label=L"$\mathfrak{R}(\langle a_1^{\
    ↪ dagger}a_2 \rangle)(t)$");
plot(imag.(adaa), color="black",linestyle="--", label=L"$\mathfrak{I}(\
    ↪ langle a_1^{\dagger} a_2 \rangle)(t)$");
plot(real.(aa), color="purple", label=L"$\mathfrak{R}(\langle a_1a_2 \
    ↪ rangle)(t)$");
plot(imag.(aa), color="purple", linestyle="--", label=L"$\mathfrak{I}(\
    ↪ langle a_1 a_2 \rangle)(t)$");
legend()
xlabel("timestep")
```

Output:

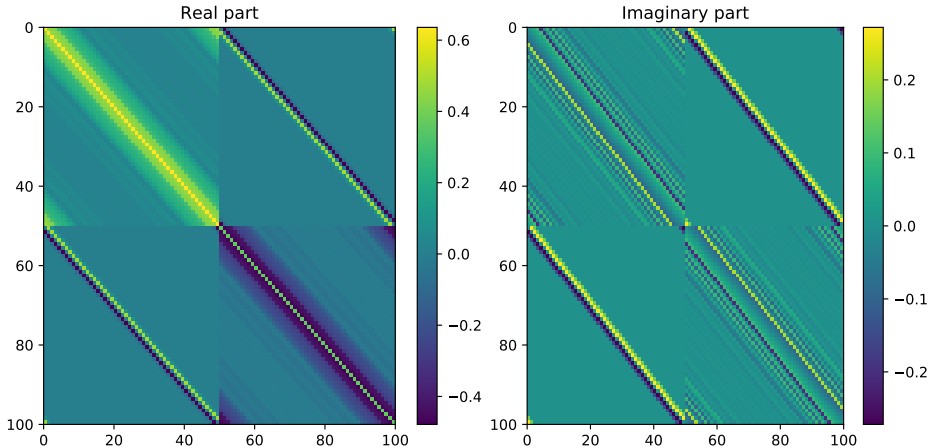

Figure 6: This is a representation of the real and imaginary part (left and right plots) of the elements of the correlation matrix $\Gamma$ of a translational invariant state with exponentially decaying correlations. The element $(i, j)$ corresponds to $\Gamma_{i,j}$. The exponential decay of the correlation is evident from the fading of the colours moving to matrix elements farther from the diagonal.

## 5 Transverse Field Ising Model

The Hamiltonian of the Transverse Field Ising model (TFI) has the form

$$\hat{H} = -\sum_{n=1}^{N-1} \sigma_n^x \sigma_{n+1}^x - g_I \sigma_N^x \sigma_1^x - \cot(\theta) \sum_{n=1}^{N} \sigma_n^z, \tag{119}$$

where $N$ is the number of sites, $\sigma_i^\alpha$ with $\alpha = x, y, z$ are the Pauli matrices at the $i$-th site and $\cot(\theta)$ is the magnetic field, with $0 < \theta < \frac{\pi}{2}$.

The parameter $g_I$ encodes the boundary conditions of the Ising model: here we consider $g_I = -1, 0, +1$, corresponding, respectively, to anti-periodic, open and periodic boundary conditions.

The model is called *transverse* field Ising model because the field interacts with the spins with $\sigma_i^z$, while the spins interact between each others with $\sigma_i^x \sigma_{i+1}^x$.

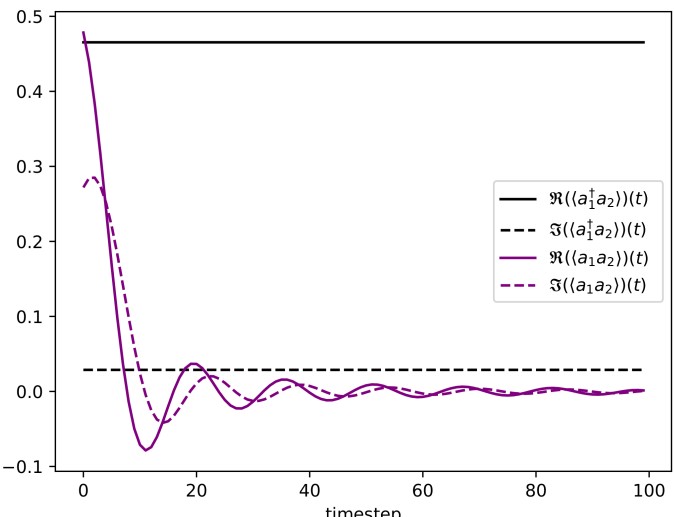

Figure 7: The time evolution induced by the Hopping Hamiltonian of the real and imaginary part of $\langle a_1^\dagger a_2 \rangle$ and $\langle a_1 a_2 \rangle$ of the translational invariant state specified in the code. The expectation values evolve as predicted by equations (118).

The TFI Hamiltonian can be exactly diagonalised using a Jordan-Wigner transformation (see appendix A.1.8) mapping spin operators to spinless fermions [24, 40–46].
in terms of fermions the Hamiltonian has the form

$$\hat{H} = -\sum_{n=1}^{N-1}(a_n^\dagger - a_n)(a_{n+1} + a_{n+1}^\dagger) + g_I P(a_N^\dagger - a_N)(a_1 + a_1^\dagger) - \cot(\theta)\sum_{n=1}^{N}(a_n^\dagger a_n - a_n a_n^\dagger), \quad (120)$$

where $P = \prod_{n=1}^{N}(1 - 2a_n^\dagger a_n)$ is the parity operator introduced in 2.2.
We notice that because of the presence of the parity operator $P$, the TFI Hamiltonian cannot be directly mapped to a f.q.h.. Nonetheless, since the Hamiltonian $\hat{H}$ commutes with the parity operator $P$ we can diagonalise $\hat{H}$ and $\hat{P}$ simultaneously. On the diagonal basis of $P$ we have that $\hat{H}$ has the block diagonal form $\hat{H} = \hat{H}_e \oplus \hat{H}_o$, where $\hat{H}_e$, called even sector Hamiltonian, corresponds to the eigenvalue $+1$ of $P$ and $\hat{H}_o$, the odd sector Hamiltonian, corresponds to the eigenvalue $-1$ of $P$. We can then proceed to diagonalise the Hamiltonian on the two sectors independently.
The Hamiltonians of the two sectors are:

$$\hat{H}_e^e = -\sum_{n=1}^{N-1}(a_n^\dagger - a_n)(a_{n+1} + a_{n+1}^\dagger) \pm g_I(a_N^\dagger - a_N)(a_1 + a_1^\dagger) - \cot(\theta)\sum_{n=1}^{N}(a_n^\dagger a_n - a_n a_n^\dagger). \quad (121)$$

Finally, we see that on each parity sector the TFI Hamiltonian is mapped to a f.q.h..
A bit of confusion can raise from considering the boundary conditions. The boundary conditions of the TFI Hamiltonian do not correspond to the boundary conditions of the fermionic Hamiltonian. In fact, let us consider the three f.q.h.

$$\hat{H}(g_F) = -\sum_{n=1}^{N-1}(a_n^\dagger - a_n)(a_{n+1} + a_{n+1}^\dagger) - g_F(a_N^\dagger - a_N)(a_1 + a_1^\dagger) - \cot(\theta)\sum_{n=1}^{N}(a_n^\dagger a_n - a_n a_n^\dagger), \quad (122)$$

where the boundary conditions are encoded by the parameter $g_F = -1, 0. + 1$ and corresponds respectively to antiperiodic, open and periodic boundary conditions of the fermionic Hamiltonian. The correspondences between spin model and fermionic model are collected in table 1.

Table 1: Corrspondences between spin models and fermionic models. In the case of periodic and antiperiodic boundary conditions of the TFI, the boundary conditions of the fermionic model depend both on the boundary condition of the spin model and on the parity of the number of particles in the chain. In the case of open boundary conditions of the TFI, the boundary condition of the associated fermionic model are independent from the the number of particles in the chain.

| $g_I = 1$ (TFI periodic) | Even sector | $g_F = -1$, (f.q.h. antiperiodic) |
|---|---|---|
| | Odd Sector | $g_F = +1$, (f.q.h periodic) |
| $g_I = 0$ (TFI open) | Even/Odd sector | $g_F = 0$, (f.q.h. open) |
| $g_I = -1$ (TFI antiperiodic) | Even sector | $g_F = +1$, (f.q.h. periodic) |
| | Odd Sector | $g_F = -1$, (f.q.h antiperiodic) |

Written in compact form (17), Hamiltonian (122), is specified by the matrices

$$A = -\frac{1}{2}\begin{pmatrix} 2\cot(\theta) & 1 & 0 & \dots & 0 & g_F \\ 1 & 2\cot(\theta) & 1 & 0 & \dots & 0 \\ 0 & 1 & 2\cot(\theta) & 1 & 0 & \dots \\ \vdots & \vdots & \ddots & \ddots & \ddots & \vdots \\ g_F & 0 & 0 & \dots & 1 & 2\cot(\theta) \end{pmatrix} \tag{123}$$

and

$$B = -\frac{1}{2}\begin{pmatrix} 0 & -1 & 0 & \dots & 0 & g_F \\ 1 & 0 & -1 & 0 & \dots & 0 \\ 0 & 1 & 0 & -1 & 0 & \dots \\ \vdots & \vdots & \ddots & \ddots & \ddots & \vdots \\ -g_F & 0 & 0 & \dots & 1 & 0 \end{pmatrix}. \tag{124}$$

We already know how to numerically diagonalise this Hamiltonian. We will present here the standard method for analytical diagonalisation, introducing the Bogoliubov transformations, and we will compare the results with the numerical diagonalisation.

## 5.1 Analytical diagonalisation of the TFI Hamiltonian

In this subsection we will see how to diagonalise the three Hamiltonians (122) analytically. For a complete and detailed treatment we refer to [24, 43] or the more recent review [46].

**Antiperiodic and periodic boundary condition fermionic Hamiltonian**  Let us first consider the case of antiperiodic and periodic boundary conditions, $g_F = -1, +1$ (APBC and PBC respectively). Both cases can be brought to the form:

$$\hat{H} = -2\sum_{k>0}\left[(\cot(\theta) + \cos(\frac{2\pi}{N}k))(f_k^\dagger f_k - f_k f_k^\dagger) + i\sin(\frac{2\pi}{N}k)(f_k^\dagger f_{-k}^\dagger - f_{-k}f_k)\right] +$$
$$- (1 + \cot(\theta))(f_0^\dagger f_0 - f_0 f_0^\dagger) - (\cot(\theta) - 1)(f_{-\frac{N}{2}}^\dagger f_{-\frac{N}{2}} - f_{-\frac{N}{2}}f_{-\frac{N}{2}}^\dagger), \tag{125}$$

where

$$f_k = \frac{1}{\sqrt{N}}\sum_{j=1}^{N} e^{i\frac{2\pi}{N}kj}a_j, \qquad f_k^\dagger = \frac{1}{\sqrt{N}}\sum_{j=1}^{N} e^{-i\frac{2\pi}{N}kj}a_j^\dagger, \tag{126}$$

with inverse transformations

$$a_j = \frac{1}{\sqrt{N}} \sum_k e^{-i\frac{2\pi}{N}kj} f_k, \qquad a_j^\dagger = \frac{1}{\sqrt{N}} \sum_k e^{i\frac{2\pi}{N}kj} f_k^\dagger, \tag{127}$$

and

$$k = \frac{-N+1}{2}, \frac{-N+1}{2}+1, \ldots, \frac{N-1}{2} \qquad \text{, for } g_F = -1 \text{ and } N \text{ even or } g_F = 1 \text{ and } N \text{ odd},$$

$$k = \frac{-N}{2}, \frac{-N}{2}+1, \ldots, \frac{N}{2}-1 \quad \text{, for } g_F = +1 \text{ and } N \text{ even or } g_F = -1 \text{ and } N \text{ odd}. \tag{128}$$

Terms with $f_0^\dagger, f_0$ and $f_{-\frac{N}{2}}, f_{-\frac{N}{2}}^\dagger$ in (125) are present just when $k = 0$ and $k = -\frac{N}{2}$ are allowed.
The different quantisations of the $k$ in the two cases are justified in [24] and can be understood by intuition noting that with the first quantisation one would have $a_{N+1} = -a_1$, while with the second $a_{N+1} = a_1$. We can write the Hamiltonian in the compact form

$$\hat{H} = \sum_{k>0} \begin{pmatrix} f_k^\dagger & f_{-k} \end{pmatrix} h_k \begin{pmatrix} f_k \\ f_{-k}^\dagger \end{pmatrix} - (1 + \cot(\theta)(f_0^\dagger f_0 - f_0 f_0^\dagger), \tag{129}$$

with

$$h_k = \begin{pmatrix} -2(\cot(\theta) + \cos(\frac{2\pi}{N}k)) & 2i\sin(\frac{2\pi}{N}k) \\ -2i\sin(\frac{2\pi}{N}k) & 2(\cot(\theta) + \cos(\frac{2\pi}{N}k)) \end{pmatrix}. \tag{130}$$

This divides the modes space in sectors that couple each $k$ with $-k$. For each of these sectors we have the unitary transformation

$$U_k = \begin{pmatrix} s_k & -it_k \\ -it_k & s_k \end{pmatrix}, \tag{131}$$

such that it diagonalises $h_k$ as

$$U_k^\dagger h_k U_k = \begin{pmatrix} \epsilon_k & 0 \\ 0 & -\epsilon_k \end{pmatrix}, \tag{132}$$

with eigenvalues

$$\epsilon_k = 2\sqrt{1 + \cot(\theta)^2 + 2\cot(\theta)\cos(\frac{2\pi}{N}k)}. \tag{133}$$

The elements of $U_k$ are defined as

$$s_k = \frac{\sin(\frac{2\pi}{N}k)}{\sqrt{\epsilon_k(\epsilon_k/2 + \cot(\theta) + \cos(\frac{2\pi}{N}k))}}, \tag{134}$$

$$t_k = \frac{\epsilon_k/2 + \cot(\theta) + \cos(\frac{2\pi}{N}k)}{\sqrt{\epsilon_k(\epsilon_k/2 + \cot(\theta) + \cos(\frac{2\pi}{N}k))}}. \tag{135}$$

This defines the fermionic transformation of all the Fourier modes that read as

$$f_k = s_k b_k - it_k b_{-k}^\dagger, \tag{136}$$

$$f_{-k}^\dagger = s_k b_{-k}^\dagger - it_k b_k. \tag{137}$$

This transformation is called Bogoliubov-Valatin transformation [47, 48], and sometimes $s_k$ and $t_k$ are expressed respectively as $\cos(\phi_k)$ and $\sin(\phi_k)$, with $\phi_k$ called Bogoliubov angle. One has that for PBC and APBC each f.q.h. of the form (125) is characterised by the choice of the quantisation of $k$ and a particular Bogoliubov angle.
We finally obtain the diagonal form of the Hamiltonian

$$\hat{H} = \sum_{k \neq -\frac{N}{2}, 0} \frac{\epsilon_k}{2}(b_k^\dagger b_k - b_k b_k^\dagger) - (1 - \cot(\theta))(f_0^\dagger f_0 - f_0 f_0^\dagger) - (\cot(\theta) - 1)(f_{-\frac{N}{2}}^\dagger f_{-\frac{N}{2}} - f_{-\frac{N}{2}} f_{-\frac{N}{2}}^\dagger). \tag{138}$$

**Open boundary condition fermionic Hamiltonian** For the open boundary conditions (OBC) form of Hamiltonian (122) there is not a clear meaning for the term $a_{N+1}$, thus we will not apply any Fourier transform. We will not show here the procedure for the diagonalisation, we refer to [24,43] or the more recent [49] for it. We have that the energies $\epsilon_k$ of the Hamiltonian in diagonal form will be

$$\epsilon_k = \sqrt{1 + \cot(\theta)^2 + 2\cot(\theta)\cos(\phi_k)}, \tag{139}$$

with $\{\phi_k\}_{k=1}^N$ the roots of equation

$$\frac{\sin((N+1)\phi)}{\sin(N\phi)} = -\frac{1}{\cot(\theta)}, \tag{140}$$

in the interval $0 \leq \phi_k \leq \pi$.

## 5.2 Ground state

In the case of OBC the ground state is easily found using the function GS_gamma() with the Hamiltonian (122) imposing $g_F = 0$.

For computing the ground state in the case of APBC or PBC we need to know if the ground state is even or odd or if it is a superposition of states with different parities. It is known that, at finite dimension, with $N$ even, for the antiperiodic Ising model the ground state is in the odd sector, while for the periodic Ising model the ground state is in the even sector. When $N$ is odd, for the antiperiodic Ising model the ground state is in the even sector and for the periodic Ising model the ground state is in the odd sector [45].

In the thermodynamic limit, the energy difference between the two sectors goes to zero, the ground state becomes degenerate. Here we present a program for finding the correct sector of the ground state and for verifying that as $N$ grows the energy difference between the ground state of the two sectors goes to zero. The program generates the output figure 8.

```julia
using F_utilities;
using PyPlot;
using LinearAlgebra;

const Fu = F_utilities;

N     = 10;
theta   = pi/8;

#Initialise the TFI with antiperiodic boundary conditions
H_APBC           = Fu.TFI_Hamiltonian(N, theta; PBC=-1);
HD_APBC, U_APBC = Fu.Diag_h(H_APBC);
#Numerical energies for the TFI with antiperiodic boundary conditions
NE_APBC          = diag(HD_APBC);

#Initialise the TFI with periodic boundary conditions
H_PBC           = Fu.TFI_Hamiltonian(N, theta; PBC=+1);
HD_PBC, U_PBC   = Fu.Diag_h(H_PBC);
#Numerical energies for the TFI with periodic boundary conditions
NE_PBC          = diag(HD_PBC);
```

```julia
#Initialise the TFI with open boundary conditions
H_OBC           = Fu.TFI_Hamiltonian(N, theta; PBC=0);
HD_OBC, U_OBC   = Fu.Diag_h(H_OBC,2);
#Numerical energies for the TFI with open boundary conditions
NE_OBC          = diag(HD_OBC);

AE_APBC = zeros(Float64, 2*N);
AE_PBC  = similar(AE_APBC);
AE_OBC  = similar(AE_APBC);

#Analitical energies for the TFI with antiperiodic, periodic and open
    ↪ boundary conditins
#The solutions of equation sin((N+1)*phi)/sin(N*phi)=-1/cot(theta)
phi = [0.293377974249272,0.586547314382234,
       0.879273168816649,1.17126278605144,
       1.46212217642804,1.75129510871389,
       2.03798675412035,2.32111594487769,
       2.59947341172037,2.87247738375037]
for n=1:N
    AE_APBC[n]  = sqrt(1+cot(theta)^2+2*cot(theta)*cos(2*pi*((1-N)/2+n-1)/
        ↪ N));
    AE_PBC[n]   = sqrt(1+cot(theta)^2+2*cot(theta)*cos(2*pi*((-N)/2+n-1)/N
        ↪ ));
    AE_OBC[n]   = sqrt(1+cot(theta)^2+2*cot(theta)*cos(phi[n]));
end

AE_APBC[(N+1):(2*N)]    = -AE_APBC[1:N];
AE_PBC[(N+1):(2*N)]     = -AE_PBC[1:N];
AE_OBC[(N+1):(2*N)]     = -AE_OBC[1:N];

#Begin plot instruction
fig = plt.figure("Comparison Analitical and Numerical Results",figsize
    ↪ =(10, 6), dpi=80)
plt.subplots_adjust(wspace=0, hspace=0)
ax1 = plt.subplot2grid((21,10), (0,0), colspan=10, rowspan=7);
ax1.set_title("Comparison Analitical and Numerical")
ax1.plot(sort(AE_APBC)[11:20], color="black",  marker="o",
        markersize=10, mfc="none" ,   label="Analytical APBC");
ax1.plot(sort(NE_APBC)[11:20], color="red", marker="+",
        markersize=10,  linestyle="None", label="Numerical APBC");
ax1.xaxis.set_ticklabels([])
yticks([1.5,2,2.5,3],[L"$1.5$",L"$2$",L"$2.5$",L"$3$"],fontsize=15)
ax1.set_ylabel(L"$\epsilon_k$",fontsize=18)
legend();
ax2 = plt.subplot2grid((21,10), (7,0), colspan=10, rowspan=7);
ax2.plot(sort(NE_PBC)[11:20],  color="purple",  marker="o",
        markersize=10, mfc="none" , label="Analytical PBC" );
```

```julia
ax2.plot(sort(AE_PBC)[11:20], color="green", marker="+",
         markersize=10,  linestyle="None", label="Numerical PBC");
ax2.xaxis.set_ticklabels([])
yticks([1.5,2,2.5,3],[L"$1.5$",L"$2$",L"$2.5$",L"$3$"],fontsize=15)
ax2.set_ylabel(L"$\epsilon_k$",fontsize=18)
legend();
ax3 = plt.subplot2grid((21,10), (14,0), colspan=10, rowspan=7);
ax3.plot(sort(NE_OBC)[11:20],  color="blue",  marker="o",
         markersize=10, mfc="none" , label="Analytical OBC" );
ax3.plot(sort(AE_OBC)[11:20],  color="orange", marker="+",
         markersize=10,  linestyle="None", label="Numerical OBC");
ax3.set_ylabel(L"$\epsilon_k$",fontsize=18)
ax3.set_xlabel(L"$k$",fontsize=18)
xticks([0,1,2,3,4,5,6,7,8,9],[L"$1$",L"$2$",L"$3$",L"$4$",L"$5$",L"$6$",L"
    ↪ $7$",L"$8$",L"$9$",L"$10$"],fontsize=15)
yticks([1.5,2,2.5,3],[L"$1.5$",L"$2$",L"$2.5$",L"$3$"],fontsize=15)
legend();
tight_layout();
#End plot instruction

#Numerical energies for the ground states
GS_APBC     = Fu.GS_gamma(HD_APBC,U_APBC);
GS_PBC      = Fu.GS_gamma(HD_PBC,U_PBC);
E_GS_APBC   = Fu.Energy(GS_APBC,(HD_APBC,U_APBC));
E_GS_PBC    = Fu.Energy(GS_PBC,(HD_PBC,U_PBC));

println("Ground State Energies");
println("G_F=-1 :        ", E_GS_APBC);
println("G_F=+1 :        ", E_GS_PBC);
```

Output:
```
Ground State Energies
G_F=-1 :  -25.18934650837823
G_F=+1 :  -25.189223629491178
```
We have $N = 10$ so the ground state is expected to be in the $g_F$=-1 sector. The computed energies confirm this.

**Degenerancy of the ground state**   In the following program we check that the ground state energies of the TFI hamiltonians (122) with $g_F = \pm 1$ converge to the same value as the dimension of the system grows. The program generates the output figure 9.

```julia
using F_utilities;
using PyPlot;

const Fu = F_utilities;

theta   = pi/8;
Delta_E = zeros(Float64, 47)

for N=4:50
```

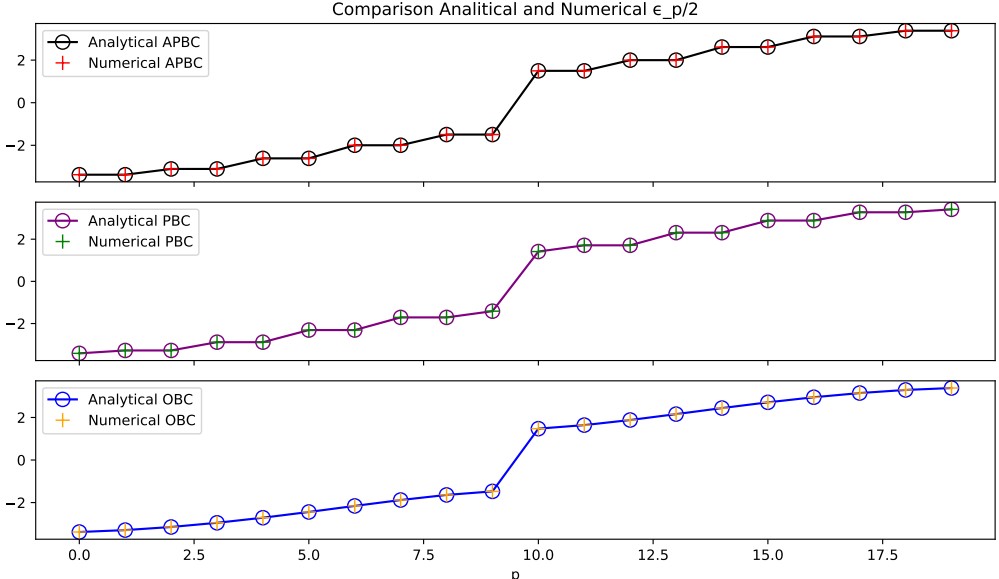

Figure 8: Output of the code 5.2. The three plots represent the analytical and numerical values of the free mode energies $\epsilon_k$ of the Hamiltonian (122) computed with antiperiodic, periodic, and free boundary conditions. We see that the energies computed with `F_utilities` correspond to the one computed analytically.

```
        H_APBC          = Fu.TFI_Hamiltonian(N, theta; PBC=-1);
        HD_APBC, U_APBC = Fu.Diag_h(H_APBC);
        H_PBC           = Fu.TFI_Hamiltonian(N, theta; PBC=+1);
        HD_PBC, U_PBC   = Fu.Diag_h(H_PBC);

        E_GS_APBC   = Fu.Energy(Fu.GS_gamma(HD_APBC,U_APBC),(HD_APBC,
            ↪ U_APBC));
        E_GS_PBC    = Fu.Energy(Fu.GS_gamma(HD_PBC,U_PBC),(HD_PBC,U_PBC));

        global Delta_E[N-3]= abs(E_GS_APBC-E_GS_PBC);
end

figure("|E_GS(GF=+1)-E_GS(GF=-1)|")
plot(4:50, log10.(abs.(Delta_E)));
xlabel(L"$N$");
ylabel(L"$\log_{10}|E_{GS}(g_F=+1,N)-E_{GS}(g_F=-1,N)|$");
tight_layout();
```

## 5.3 Time Evolution

As in the case of the Hopping model, even for the Hamiltonian (122) it is possible to explicitly compute the time evolution of the correlation matrix elements. We focus here on the case of $g_F = -1$ and $N$ even in order to simplify the analitical form. The principal difference with the Hopping model is that, in the case of the fermionic TFI, the transformation that diagonalises the Hamiltonian is not a simple Fourier transform, but it is a composition of a Fourier transform and a Bogoliubov transformation. We exemplify how to obtain an analitical form for the time evolution of the term $\langle a_1^\dagger a_1 \rangle$ of a translational invariant correlation matrix. As a first step,

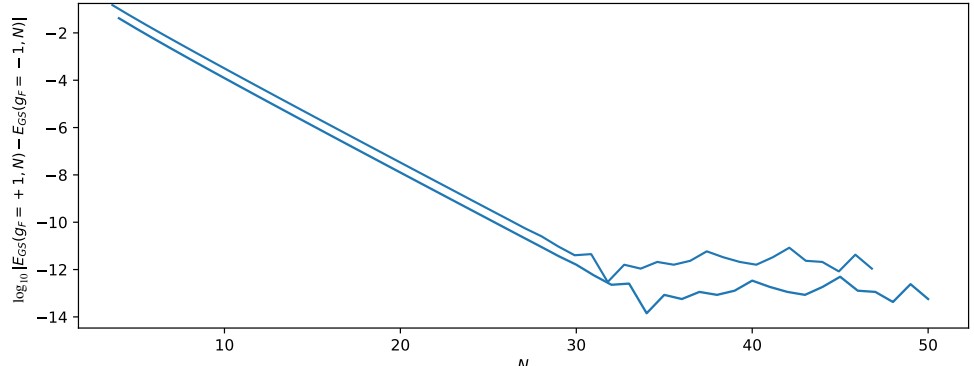

Figure 9: The ground states of Hamiltonians (122) for $g_F = \pm 1$ converge exponentially to the same value. The ground state of the antiperiodic and of the periodic transverse field Ising model is degenerate in the thermodynamic limit.

exploiting the translational invariance of the state and moving to the Fourier modes, we write

$$
\langle a_1^\dagger a_1 \rangle = \frac{1}{N} \sum_{n=1}^{N} \langle a_n^\dagger a_n \rangle = \frac{1}{N^2} \sum_{n=1}^{N} \sum_{k=\frac{-N+1}{2}}^{\frac{N+1}{2}} \sum_{k'=\frac{-N+1}{2}}^{\frac{N+1}{2}} e^{-i\frac{2\pi}{N}n(k-k')} \langle f_k^\dagger f_{k'} \rangle =
$$

$$
= \frac{1}{N} \sum_{k=\frac{-N+1}{2}}^{\frac{N+1}{2}} \langle f_k^\dagger f_k \rangle. \tag{141}
$$

We then move to the Bogoliubov modes with the transformation (136) obtaining

$$
\langle a_1^\dagger a_1 \rangle = \frac{1}{N} \sum_{k=\frac{-N+1}{2}}^{\frac{N+1}{2}} \left[ s_k^2 \langle b_k^\dagger b_k \rangle + t_k^2 \langle b_{-k} b_{-k}^\dagger \rangle + i s_k t_k (\langle b_{-k} b_k \rangle - \langle b_k^\dagger b_{-k}^\dagger \rangle) \right]. \tag{142}
$$

In this basis the Hamiltonian is diagonal, thus the time evolution easily computed as

$$
\langle a_1^\dagger a_1 \rangle(t) = \frac{1}{N} \sum_{k=\frac{-N+1}{2}}^{\frac{N+1}{2}} \left[ s_k^2 \langle b_k^\dagger b_k \rangle + t_k^2 \langle b_{-k} b_{-k}^\dagger \rangle + i s_k t_k (e^{i2\epsilon_k t} \langle b_{-k} b_k \rangle - e^{-i2\epsilon_k t} \langle b_k^\dagger b_{-k}^\dagger \rangle) \right]. \tag{143}
$$

To obtain the expression of $\langle a_1^\dagger a_1 \rangle(t)$ in terms of correlators of the operators $a^\dagger, a$ we just have to map the $b^\dagger, b$ to $a^\dagger, a$.

# 6 Benchmarking with fermionic Gaussian states

Fermionic Gaussian states can be used as a tool for benchmarking algorithms. We will see how tools developed for general quantum states can be translated to the language of correlation matrices. To understand the idea behind the benchmarking, let us take an explicit example. In the next subsection we are going to see the imaginary time evolution of fermionic Gaussian states. One usually uses the imaginary time evolution of a state for computing the ground state of an Hamiltonian. Of course, in the case of f.g.s., computing the ground state is not the main purpose, as we know already how to compute it for any f.q.h.. Knowing already the exact results allows us to compare the algorithm for the imaginary time evolution with the exact results and get good insight in what we should expect in a context where the exact result is

not known.

If we have an algorithm acting on some generic quantum state, we can try to translate it in the formalism of f.g.s. and benchmark it to the exact results.

In this subsection we present the translation of some well known algorithm in the language of correlation matrices. The purpose of this subsection is not to benchmark these algorithm, but instead, to translate some important existing algorithms. This will provide us the translations of the possible building blocks of any novel and more complex algorithm.

## 6.1 Imaginary-time evolution

In order to find the ground state of a non-degenerate Hamiltonian $H$ one can use the following equality

$$|GS\rangle = \lim_{\tau \to \infty} \frac{e^{-H\tau}|\psi\rangle}{||e^{-H\tau}|\psi\rangle||}, \tag{144}$$

starting from a generic state $|\psi\rangle$ with $\langle GS|\psi\rangle \neq 0$.

To see this, let us consider the orthononormal basis $\{|E_i\rangle\}_i$ generated by the collection of the eigenvectors of $H$, with eigenvalues $\{E_i\}_i$ such that $0 \leq E_0 \lneqq E_1 \leq E_2 \leq$, where $\mathcal{H}$ is the Hilbert space on which $H$ act.

Expanding $|\psi\rangle$ on this basis one obtains $|\psi\rangle = \sum_i c_i |E_i\rangle$, with $c_0 \neq 0$ from the fact that $\langle GS|\psi\rangle \neq 0$. One can thus see that eq (144) is just a projection to the ground state:

$$\lim_{\tau \to \infty} \frac{e^{-H\tau}|\psi\rangle}{||e^{-H\tau}|\psi\rangle||} = \lim_{\tau \to \infty} \sum_i \frac{e^{-E_i \tau} c_i}{\sqrt{\sum_i e^{-2E_i \tau}|c_i|^2}} |E_i\rangle \tag{145}$$

$$= \lim_{\tau \to \infty} \sum_i \frac{e^{-\frac{E_i}{E_0}\tau} c_i}{\sqrt{\sum_i e^{-2\frac{E_i}{E_0}\tau}|c_i|^2}} |E_i\rangle = |E_0\rangle, \tag{146}$$

and thus that $\lim_{\tau \to \infty} \frac{e^{-H\tau}}{||e^{-H\tau}||}$ is the projector on the ground state:

$$\lim_{\tau \to \infty} \frac{e^{-H\tau}}{||e^{-H\tau}||} = \lim_{t \to \infty} \frac{\sum_i e^{-E_i \tau}|E_i\rangle\langle E_i|}{\sqrt{\sum_i e^{-2E_i \tau}}} \tag{147}$$

$$= \lim_{\tau \to \infty} \frac{\sum_i e^{-\frac{E_i}{E_0}\tau}|E_i\rangle\langle E_i|}{\sqrt{\sum_i e^{-2\frac{E_i}{E_0}\tau}}} = |E_0\rangle\langle E_0|. \tag{148}$$

The imaginary-time evolution is directly related to the power method presented in section A.1.6. The eigenvenvector associated to the smallest eigenvalue $E_0$ of $H$ is the eigenvector associated to the biggest eigenvalue of $e^{-H}$ and this can be approximately obtained using the power method by computing $(e^{-H})^N|\psi\rangle$, a procedure that in the limit of $N \to \infty$ is analogous to equation (144).

Applying the same method to the density matrix one can obtain the ground state $\rho_{GS}$ of a non degenerate Hamiltonian $H$ from a general density matrix $\rho$ such that $Tr[\rho\rho_{GS}] \neq 0$ as

$$\rho_{GS} = \lim_{\tau \to \infty} \frac{e^{-H\tau}\rho e^{-H\tau}}{Tr[\rho e^{-2H\tau}]}. \tag{149}$$

We refer to the method for obtaining the ground state using (144) as performing an *imaginary time evolution*.

This is the case because, if for the time evolution operator $U(t) = e^{-iHt}$ for the Hamiltonian $H$, we select $t = -i\tau$ we obtain the operator $e^{-H\tau}$ that is the one of eq (144). One can thus

write in a non-formal way $|GS\rangle = \lim_{t\to-i\infty} \frac{|\psi(t)\rangle}{||\psi(t)\rangle||}$.

It is important to keep in mind that the operator $e^{-H\tau}$ is not unitary and for this reason it does not preserve the norm of the state and one has to renormalise it.

## 6.2 Numerical Imaginary time evolution

In the numerical approach to imaginary time evolution one faces some difficulties.

Almost in all cases one is forced to evolve the state step by step renormalising every time, performing a discrete imaginary time evolution.

This procedure does not allow to reach infinite time in a finite amount of time steps, thus one has to find a criterion to stop the evolution when the convergence is accurate up to some confidence parameter. To check if the reached state is the expected state is tricky and theoretically impossible in most of the cases since one does not always have the exact value of the energy of the ground state.

A method for checking the convergence is to check the energy difference between two steps of the discrete imaginary time evolution. Once the difference in energy between two steps is lower than an acceptable value $\epsilon$, one decides that the algorithm converged.

It is not always the case though. It is also possible that the approximate imaginary time evolution stops at some plateaux and thus it tricks the algorithm in believing in a false convergence to the ground state. Possible escape tricks exists. Perhaps one can slightly perturb the state and then compute the value of the energy after another step of the evolution, this can help in escaping from local minima. In general such tricks have to be adapted to the particular needs.

**Imaginary time evolution of the correlation matrix**   The imaginary time evolution of the correlation matrix is defined as

$$\Gamma_{i,j}(\tau) = Tr\left[\rho(\tau)\vec{\alpha}_i\vec{\alpha}_j^\dagger\right] \tag{150}$$

$$= \frac{Tr\left[e^{-\hat{H}\tau}\rho e^{-\hat{H}\tau}\vec{\alpha}_i\vec{\alpha}_j^\dagger\right]}{Tr\left[e^{-\hat{H}\tau}\rho e^{-\hat{H}\tau}\right]} . \tag{151}$$

Obtaining an explicit form for $\Gamma(\tau)$ just in terms of $H$ and $\Gamma(0)$ is not easy. Following the reasoning made for the real time evolution, one can compute the imaginary time evolution in Heisenberg picture with $e^{-\hat{H}\tau}$ of the operator $\vec{\alpha}_i\vec{\alpha}_j^\dagger$. Using the Baker-Campbell-Hausdorff formula (i.e. $e^A Be^A = \sum_{n=0}^\infty \frac{1}{n!} \underbrace{\{A,\dots\{A}_n, B \underbrace{\}\dots\}}_n$ i.e. *B.C.H.2* in Appendix A.3.4) and moving in the diagonal basis with Dirac operators $\vec{\beta}$ one can write the Hamiltonian as $\hat{H} = \sum_k \epsilon(k)\left(b_k^\dagger b_k - b_k b_k^\dagger\right)$. Thus one has

$$e^{-\hat{H}\tau}\vec{\beta}_l\vec{\beta}_j^\dagger e^{-\hat{H}\tau} = \sum_{n=0}^\infty \frac{-\tau^n}{n!} \underbrace{\{\hat{H},\dots\{\hat{H}}_n, \vec{\beta}_l\vec{\beta}_j^\dagger \underbrace{\}\dots\}}_n . \tag{152}$$

Since $b_l^\dagger b_j\hat{H} = (\hat{H} + 2\Delta_{l,j})b_l^\dagger b_j$, we cannot simplify this expression as in the case of real time evolution.

To obtain a numerical algorithm for the imaginary time evolution one has to realise that, for each value of $\tau$, $\Gamma(\tau)$ is just the correlation matrix of the f.g.s.

$$\rho(\tau) = \frac{e^{-\hat{H}\tau}\rho e^{-\hat{H}\tau}}{Tr\left[e^{-\hat{H}\tau}\rho e^{-\hat{H}\tau}\right]} , \tag{153}$$

and this can be seen as the state obtained by correctly normalising the matrix product of the density matrices of three states. The trick for obtaining the correlation matrix $\Gamma(\tau)$ is thus using the product rule (see subsection 3.9) of the initial f.g.s. $\rho$ and the thermal state $\rho_{\beta=\tau} = \frac{e^{-H\tau}}{\text{Tr}[e^{-\hat{H}\tau}]}$. This allows us to compute the imaginary time evolution of fermionic Gaussian states.

---

**F_utilities** 6.1: `Evolve_imag(`$\Gamma$`, H_D, U, `$\tau$`)`$\rightarrow \Gamma(\tau)$

This function returns the correlation matrix $\Gamma$ evolved at imaginary time $\tau$ with $H$. Matrices $H_D$ and $U$ are the output of `Diag_h(H)`.

---

### 6.3 Fermionic Gaussian States with Fixed Bond Dimension

The compression of correlation matrices of f.g.s. in a similar fashion of matrix product states (MPS, see appendix A.2) has been introduced in [50].

Let us consider a pure fermionic Gaussian state completely described by the $N \times N$ matrix $\Lambda_{i,j} = \langle a_i^\dagger a_j \rangle$. Since $\Lambda$ corresponds to a pure state, its eigenvalues are either 1 (the mode is occupied) or 0 (the mode is unoccupied). This high degeneracy can be exploited mixing occupied (or unoccupied) eigenstate for finding a basis in which these modes are localised. In systems with a limited entanglement structure we expect to be able to find a basis in which eigenstates are localised. This fact can be justified as follows. Let $\Lambda$ be the correlation matrix of the ground state of a 1D local Hamiltonian. We consider the partition of the first $\ell$ sites of the system. The state of the partition is described by the $\ell \times \ell$ correlation matrix $\Lambda_\ell$, and it is in general a mixed state. In ground states of 1D local Hamiltonians, the von Neumann entropy of partitions of the systems of dimension $\ell$ grows at most as $\log(\ell)$. Since the contributes to the von Neumann entropy are given by the eigenvalues of the correlation matrix different from 0

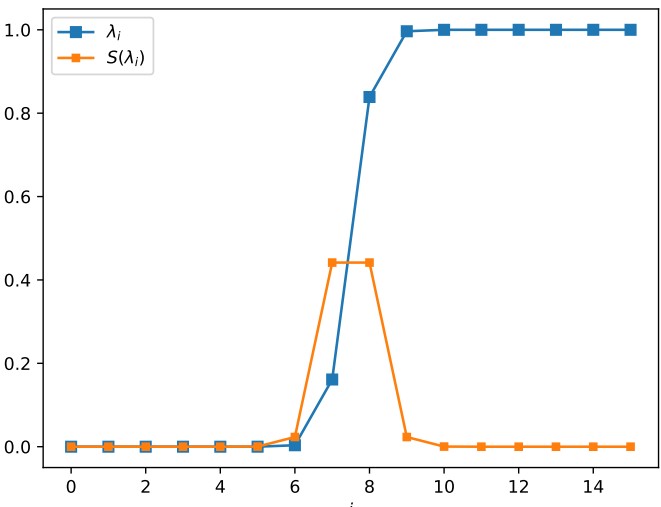

Figure 10: In blue the eigenvalues $\lambda_i$ of the reduced state $\Lambda_{\ell=16}$ of the ground state of a hopping Hamiltonian with $N = 500$. In orange the von Neumann entropy $S(\lambda_i)$ of the mode associated to each eigenvalue $\lambda_i$. The total von Neumann entropy of the partition $\Lambda_{\ell=16}$ is given by the sum of the entropies of each mode (see (89)). Since the entropy of a partition $\Lambda_\ell$ is bounded by $\log(\ell)$, with growing $\ell$ the added modes must have associated eigenvalues close to 0 or 1.

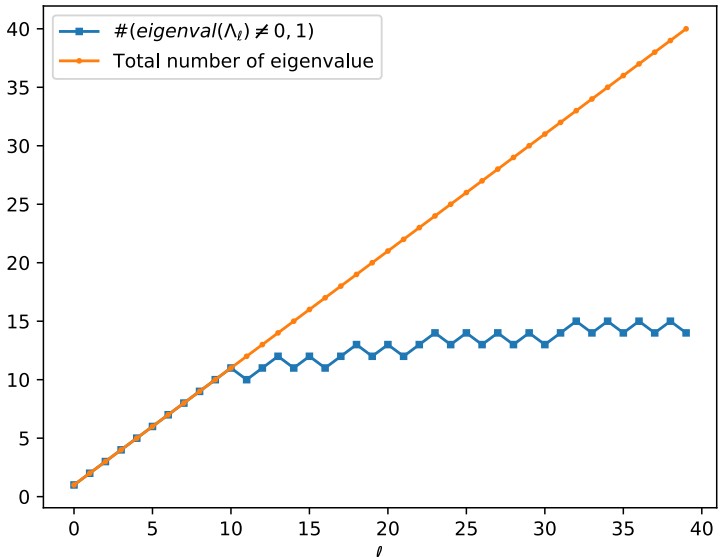

Figure 11: In blue the number of eigenvalues of $\Lambda_\ell$ that are not 0 or 1 up to machine precision, in orange the total number of eigenvalues of $\Lambda_\ell$ in the ground state of a hopping Hamiltonian with $N = 500$. When the dimension of the partition is $\ell > 10$ the entanglement grows logarithmically and the number of eigenvalues equal to 1 or 0 starts growing linearly.

and 1, we have that, for growing values of $\ell$, we expect to find most eigenvalues of $\Lambda_\ell$ closer and closer to 1 or 0 (see figure 10 and figure 11). In the limit of $\ell = N$ each the eigenvalue of $\Lambda_{\ell=N}$ are or 0 or 1. Now suppose that diagonalising $\Lambda_{\ell<N}$, the eigenvalue associated to the eigenvector $\vec{v}$ is $\sim 1$. This makes $\vec{v}$ also an approximate eingenvector of $\Lambda$. In fact we know that the eigenvalues of $\Lambda$ are 0 or 1, thus it is sufficient to extend the dimension of the eigenvector $\vec{v}$ adding to it $N - \ell$ zeros to obtain an eigenvector of $\Lambda$.

In [50], developing on this idea, the authors are able to construct a compression algorithm for correlation matrices and directly map it to the MPS representation of the state.
Here we will illustrate a method for obtaining the correlation matrix of a f.g.s. expressed as an MPS with fixed bond dimension $D$.

Let us consider a pure f.g.s. $|\psi\rangle$ on a system with $N$ sites with associated $N \times N$ correlation matrix $\Lambda$. We denote with $|\psi^D\rangle$ the state obtained representing $|\psi\rangle$ with an MPS of fixed bond dimension $D$. We are interested in the correlation matrix $\Lambda^D$ of the state $|\psi^D\rangle$.
For a bipartition having bond dimension $D$ corresponds to having Schmidt rank $D$ [34]. If a state $|\psi\rangle$ has bond dimension $D' > D$, we can approximate it at bond dimension $D$ by setting to 0 the lower Schmidt coefficients and renormalising the state. With the formalism of correlation matrices we cannot directly manipulate the single Schmidt coefficients, but we can approximate low entangled modes with product modes. Approximating an entangled mode with a product mode corresponds to setting half of the Schmidt coefficients to zero. With this insight we can devise the following algorithm for obtaining $\Lambda^D$.
We proceed as follows. We consider $\Lambda_{1,...,m+1}$, the correlation matrix of the partition of the first $m + 1$ sites, where $m = \lceil \log_2(D) \rceil$. We will refer to $m$ as the bond dimension of the correlation matrix.
We diagonalise it as $\Lambda_{1,...,m+1} = U_1 D^1_{1,...,m+1} U_1^\dagger$, the diagonal elements of $D^1_{1,...,m+1}$ are organised such that, the top left element $\lambda_1^1$ is the closest to 0 or 1. Suppose $\lambda_1^1 \sim 1$. We expand $U_1$ to

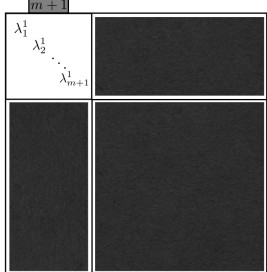

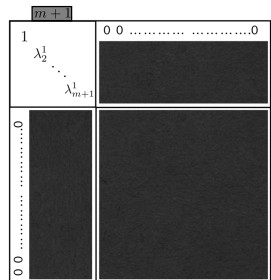

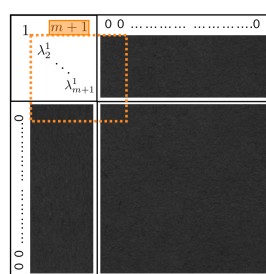

(a) Step 1: Diagonalise the subsystem $\Lambda^1_{1,\dots,m+1}$. The eigenvalues are ordered such that $\min(|1 - \lambda^1_1|, |\lambda^1_1|) \leq \min(|1 - \lambda^1_2|, |\lambda^1_2|) \leq \dots \leq \min(|1 - \lambda^1_{m+1}|, |\lambda^1_{m+1}|)$.

(b) Step 2: Since $\lambda^1_1 \sim 1$ ($\lambda^1_1 \sim 0$) we set it to 1 (0). We set to zero all the other elements of the first row and first column of $\Lambda$. This will be the an eigenvalue of $\Lambda$.

(c) Step 3: We move to the second subsystem $\Lambda^2_{2,\dots,m+2}$. We note that now the correlation matrix is represented in a mixed basis and the lower indices do not exactly represent the sites of the system.

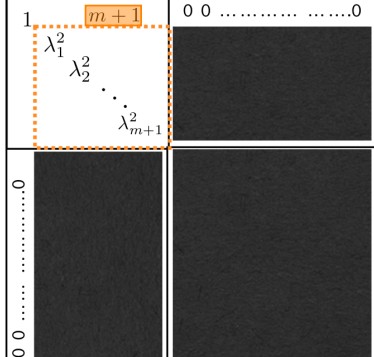

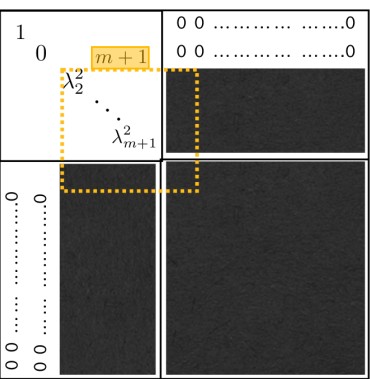

(d) Step 4: We diagonalise $\Lambda^2_{2,\dots,\ell+1}$

(e) Step 5: After resetting $\lambda^2_1$, we move to $\Lambda^3_{3,\dots,m+3}$.

Figure 12: Steps of the algorithm for reducing the bond dimension of a fermionic Gaussian state. The big squares represent the correlation matrix $\Lambda$. We repeat this procedure $(N-m)$ times then we continue for $m$ steps reducing by one the dimension of the reduced system at each step. At the end one obtains a diagonal matrix with diagonal elements equal to 1 or 0.

be $N \times N$ adding ones on the diagonal and we have that the top left element of $\Lambda^1 = U_1^\dagger \Lambda U_1$ is $\lambda_1^1$. We set the first column and first row of $\Lambda^1$ to 0 and $(\Lambda^1)_{1,1}$ to 1 (because $\lambda_1^1 \sim 1$). We then proceed diagonalising $\Lambda^1_{2,\ldots,m+2} = U_2 D^2_{2,\ldots,m+2} U_d^\dagger$. Suppose this time the top left element of $D^2$ is $\lambda_1^2 \sim 0$. We set the second column and second row of $\Lambda^2 = U_2^\dagger \Lambda^1 U_2$ to 0 and $(\Lambda^2)_{2,2}$ to 0 (because $\lambda_1^2 \sim 0$). Iterating this procedure $N - m - 1$ times we obtain a correlation matrix $\Lambda^{(N-m+1)}$ with $N - m$ diagonal elements equal to 0 or 1. We proceed with the same procedure decreasing the dimension of the reduced system everytime until after $N$ steps we obtain a diagonal matrix with diagonal elements equal to 0 or 1. Returning to the original basis applying all the transformation $\{U_i\}_{i=1,\ldots,N}$ to $\Lambda^N$ we obtain the correlation matrix associated to the state $|\psi^D\rangle$. We report a schematic representation of the algorithm in figures 12.

In figure 13 we plot the number of eigenvalues different from 0 and 1 for different partitions of the system for the ground state $\Gamma$ of a hopping Hamiltonian of a system of $N = 200$ sites, and for $\Gamma$ with bond dimension reduced to $m$.

We know that for ground states of $1D$ local Hamiltonians the amount of entanglement (measured by the von Neumann entropy $S$) of any region of an MPS of bond dimension $D$ is bounded by $S \leq \log(D)$. In figure 14 we plot the value of the entropy of different regions of the ground state of a random Hamiltonian. The entropy is indeed bounded by $S \leq \log(D)$ with $D = 2^m$.

Combining this method with the imaginary time evolution algorithm one can construct the time evolving block decimation algorithms on the space of correlations matrices.

Together with the algorithm for reducing the bond dimension of one dimensional systems on the space of correlation matrices, in F_utilities we include the algorithm for reducing the bond dimension of specific two dimensional systems. In particular we focussed on two dimensional systems where the Hamiltonian can be sectorised. This algorithm differs from the one for one dimensional systems in the way it handles the lowest Schmidt's coefficients of different sectors. Taking advantage of the sectorisation of the Hamiltonian, the algorithm becomes more complex, but at the same time more efficient.

---

**F_utilities 6.2: RBD($\Gamma$, $m$)$\rightarrow \Gamma(m)$**

This function returns the correlation matrix $\Gamma(m)$ obtained reducing the bond dimension of $\Gamma$ to $m$.

---

```
using F_utilities;
using PyPlot;
using LinearAlgebra;

const Fu     = F_utilities;
const LinA   = LinearAlgebra;

N            = 400;
H            = Fu.Build_hopping_hamiltonian(N);
HD, U        = Fu.Diag_h(H);
Gamma        = Fu.GS_gamma(HD,U);
Gamma_RBD    = Fu.RBD(Gamma,5)

prod_modes_border        = zeros(Int64, N);
prod_modes_RBD_border    = zeros(Int64, N);
prod_modes_bulk          = zeros(Int64, div(N,2));
prod_modes_RBD_bulk      = zeros(Int64, div(N,2));
for l=1:N
```

```
        DA,UA               = Fu.Diag_gamma(Fu.Reduce_gamma(Gamma,l,1));
        DA_RBD,UA_RBD       = Fu.Diag_gamma(Fu.Reduce_gamma(Gamma_RBD,l,1));
        prod_modes_border[l]    = count(i->i!=0, round.(real.(LinA.diag(DA)
            ↪ [1:l]),digits=14));
        prod_modes_RBD_border[l] = count(i->i!=0, round.(real.(LinA.diag(
            ↪ DA_RBD)[1:l]),digits=14));
end
for l=1:div(N,2)
        DA,UA                   = Fu.Diag_gamma(Fu.Reduce_gamma(Gamma,l, div
            ↪ (N,2)));
        DA_RBD,UA_RBD           = Fu.Diag_gamma(Fu.Reduce_gamma(Gamma_RBD,l,
            ↪  div(N,2)));
        prod_modes_bulk[l]      = count(i->i!=0, round.(real.(LinA.diag(DA)
            ↪ [1:l]),digits=14));
        prod_modes_RBD_bulk[l]  = count(i->i!=0, round.(real.(LinA.diag(
            ↪ DA_RBD)[1:l]),digits=14));
end
figure("prod_eigenvalues");
plot(1:N,prod_modes_border, marker="s", linewidth=0.5, markersize=0.7,
    ↪ label=L"$\#(eigenval(\Lambda_{1,\dots,\ell}\neq 0,1)$");
plot(1:div(N,2),prod_modes_bulk, marker="s", linewidth=0.5, markersize
    ↪ =0.7, label=L"$\#(eigenval(\Lambda_{\frac{N}{2},\dots,\frac{N}{2}+\
    ↪ ell})\neq 0,1)$");
plot(1:N,prod_modes_RBD_border, marker="s", linewidth=0.5, markersize=0.7,
    ↪  label=L"$\#(eigenval(\Lambda(m=5)_{1,\dots,\ell}\neq 0,1)$");
plot(1:div(N,2),prod_modes_RBD_bulk, marker="s", linewidth=0.5, markersize
    ↪ =0.7, label=L"$\#(eigenval(\Lambda(m=5)_{\frac{N}{2},\dots,\frac{N
    ↪ }{2}+\ell}\neq 0,1)$");
axvline(div(N,2), linestyle="--", linewidth=0.5, color="gray")
xlabel(L"$\ell$")
grid(axis="y", linestyle="--")
legend();
```

Output:

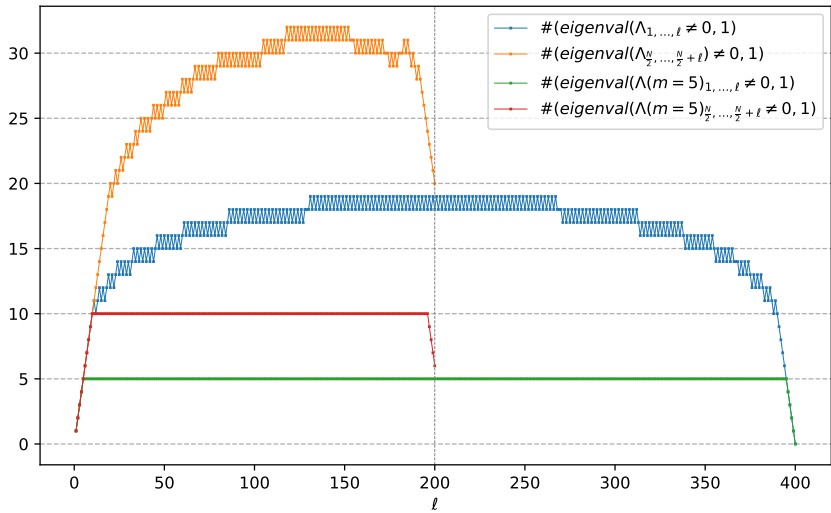

Figure 13: Eigenvalues different from 0 and 1 for different partitions of the system for the ground state $\Gamma$ of a hopping Hamiltonian of a system of $N = 400$ sites, and for $\Gamma$ with bond dimension reduced to $m$. The blue dots correspond to partitions of $\Gamma$ with first site at the boundary of the system and with dimension $\ell$. The orange dots correspond to partitions of $\Gamma$ with first site at the boundary of the system and with dimension $\ell$. The green dots are anologous to the blue dots, but computed for the state $\Gamma(m = 5)$ obtained reducing the bond dimension of $\Gamma$ to $m = 5$. Red dots are analogous to the orange dots, but computed for $\Gamma(m = 5)$. As expected the number of eigenvalues different from 0 and 1 are bounded as $\#eigenval(\neq 0, 1) \leq 2m$.

```julia
using F_utilities;
using PyPlot;
using LinearAlgebra;

const Fu   = F_utilities;
const LinA = LinearAlgebra;

function Random_hamiltonian(N)
  A   = rand(N,N)+im*rand(N,N);
  A   = (A+A')/2.;
  bd  = rand(N-1).+im*rand(N-1);
  B   = Tridiagonal(bd, zeros(Complex{Float64}, N), -bd);
  H   = zeros(Complex{Float64}, 2*N, 2*N);
  H[(1:N),(1:N)]          = -conj(A);
  H[(1:N).+N,(1:N)]       = -conj(B);
  H[(1:N),(1:N).+N]       = B;
  H[(1:N).+N,(1:N).+N]    = A;

  return H;
end

N             = 400;
H             = Random_hamiltonian(N);
HD, U         = Fu.Diag_h(H);
Gamma         = Fu.GS_gamma(HD,U);
Gamma_RBD     = Fu.RBD(Gamma,10)
S_modes_border       = zeros(Float64, N);
S_modes_RBD_border   = zeros(Float64, N);
S_modes_bulk         = zeros(Float64, div(N,2));
S_modes_RBD_bulk     = zeros(Float64, div(N,2));
for l=1:N
    S_modes_border[l]     = Fu.VN_entropy(Fu.Reduce_gamma(Gamma,l,1));
    S_modes_RBD_border[l] = Fu.VN_entropy(Fu.Reduce_gamma(Gamma_RBD,l,1)
        ↪ );
end
for l=1:div(N,2)
    S_modes_bulk[l]           = Fu.VN_entropy(Fu.Reduce_gamma(Gamma,l, div
        ↪ (N,2)));
    S_modes_RBD_bulk[l]       = Fu.VN_entropy(Fu.Reduce_gamma(Gamma_RBD,l,
        ↪ div(N,2)));
end
```

```
figure("Entropies");
plot(1:N,log.(abs.(S_modes_border)), marker="s", linewidth=0.5, markersize
    ↪ =0.7, label=L"$F=S(\Lambda_{1,\dots,\ell}\neq 0,1)$");
plot(1:div(N,2),log.(abs.(S_modes_bulk)), marker="s", linewidth=0.5,
    ↪ markersize=0.7, label=L"$F=S(\Lambda_{\frac{N}{2},\dots,\ell+\frac{
    ↪ N}{2}})\neq 0,1)$");
plot(1:N,log.(abs.(S_modes_RBD_border)), marker="s", linewidth=0.5,
    ↪ markersize=0.7, label=L"$F=S(\Lambda(m=10)_{1,\dots,\ell}\neq 0,1)$
    ↪ ");
plot(1:div(N,2),log.(abs.(S_modes_RBD_bulk)), marker="s", linewidth=0.5,
    ↪ markersize=0.7, label=L"$F=S(\Lambda(m=10)_{\frac{N}{2},\dots,\ell
    ↪ +\frac{N}{2}}\neq 0,1)$");
axvline(div(N,2), linestyle="--", linewidth=0.5, color="gray")
axhline(log(log(2^9)), linestyle="-.", color="red", label="F=log(D)")
axhline(log(2*log(2^9)), linestyle="-.", color="red", label="F=2*log(D)")
xlabel(L"$\ell$")
ylabel(L"$\log(F)$")
# grid(axis="y", linestyle="--")
legend();
```

Output:

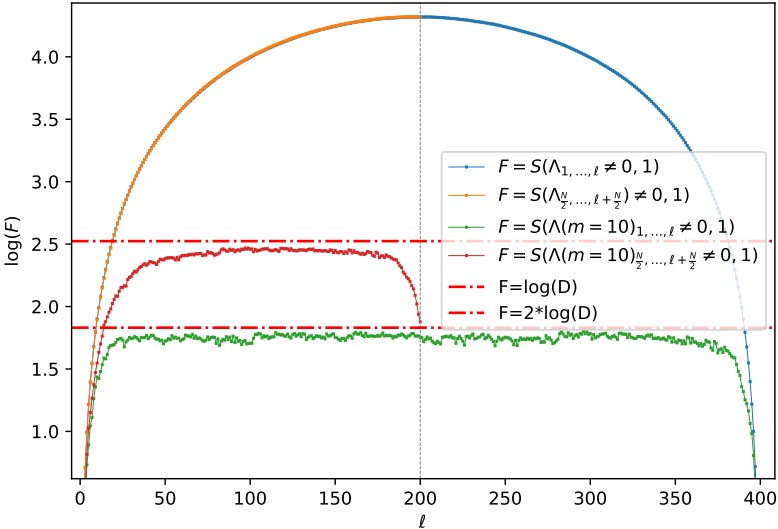

Figure 14: The entropy of different regions of the ground state of a random Hamiltonian. The blue dots correspond to partitions of dimension $\ell$ with first site at the boundary of the chain. The orange dots correspond to partitions of dimension $\ell$ with first site in the middle of the chain. The green dots are analogous to the blue dots but computed for the state $\Gamma(m=1)$ obtained reducing the bond dimension of $\Gamma$ to $m=1$. Red dots are anologous to the orange dots, but computed for $\Gamma(m=1)$. As expected since the Hamiltonian is random and long range the entropy has an extensive scaling with the subsystem size. The red dash-dotted horizontal lines represent the upper bound for the entropy of a partition (starting at the border or not respectively for $\log(D)$ and $2\log(D)$). As we can see the entropy is always bounded by $S \leq \log(D)$ with $D = 2^m$ as expected.

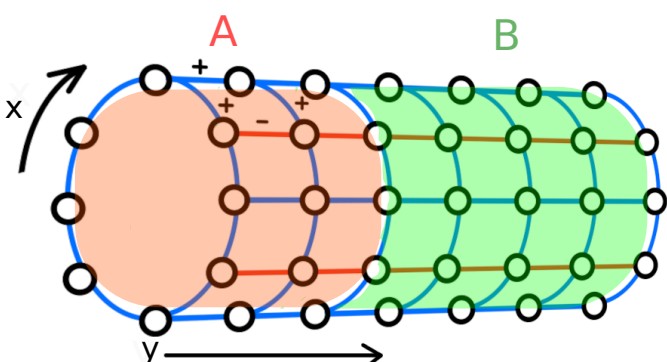

Figure 15: Lattice with periodic boundary conditions along the $x$ direction. Red lines correspond to negative couplings, blue lines correspond to positive couplings. Two possible partitions $A$ and $B$ are higlighted in red and green respectively.

## 6.4 Reduction of the bond dimension of a two dimensional system divided in sectors

We consider the fermionic quadratic Hamiltonian

$$\hat{H} = \sum_x \sum_y \left[ a^\dagger_{x,y} a_{x+1,y} + a^\dagger_{x+1,y} a_{x,y} + (-1)^x \left( a^\dagger_{x,y} a_{x,y+1} + a^\dagger_{x,y+1} a_{x,y} \right) \right], \tag{154}$$

defined on the on an $L \times L$ lattice on a cylinder with periodic boundary conditions along the $x$ direction and open boundary conditions along the $y$ direction as in figure 15. Because of the boundary conditions we have that $L$ must be even.

Substituting the Fourier operators

$$a^\dagger_{x,y} = \frac{1}{\sqrt{L}} \sum_k e^{-i \frac{2\pi}{L} kx} c^\dagger_{k,y}, \tag{155}$$

where, because of the boundary conditions, we choose $k$ as

$$k = -\frac{L}{2}, -\frac{L}{2} + 1, \ldots, \frac{L}{2} - 1, \tag{156}$$

the Hamiltonian becomes as

$$\hat{H} = \sum_{k<0} \left[ \sum_y 2\cos(\frac{2\pi}{L} k) \left( c^\dagger_{k,y} c_{k,y} - c^\dagger_{k+\frac{L}{2},y} c_{k+\frac{L}{2},y} \right) \right. \tag{157}$$
$$\left. + c^\dagger_{k,y} c_{k+\frac{L}{2},y+1} + c^\dagger_{k+\frac{L}{2},y} c_{x,y+1} + c^\dagger_{k+\frac{L}{2},y+1} c_{k,y} + c^\dagger_{k,y+1} c_{k+\frac{L}{2},y} \right].$$

In this form the Hamiltonian is divided in $\frac{L}{2}$ sectors, each one corresponding to the couples of values of $\{k, k+\frac{L}{2}\}$. This means that the eigenstates of this Hamiltonain are product states of states defined on each $\{k, k+\frac{L}{2}\}$ sector. Thus, these eigenstates, instead of being described by a $2L^2 \times 2L^2$ correlation matrix, are instead described by just a collection of $\frac{L}{2}$ correlation matrices of dimension $2L \times 2L$, where each of these correlation matrices corresponds to a stripe of the cylinder in the $(k, y)$ space.

We call states of this kind, sectorised states.

Performing the reduction of the bond dimension of a sectorised state, one can expolit the sectorisaton property in order to reduce the computational cost of the operation.

Consider for example the ground state of Hamiltonian (154). Once we move to the Fourier basis along the $x$ direction this becomes a sectorised state. Mimicking the encoding of this

quantum state with a tensor network corresponds to fixing its Schmidt rank relatively to some iterative partition scheme. We choose a partition scheme that increasingly cuts the cylinder perpendicularly to $y$. Since the Fourier transformation we applied mixes only Dirac operator corresponding to the same value of $y$, this partition scheme is a geometric partition scheme (it is equivalent on the $(x, y)$ space and the $(k, y)$ space). Step $l$ of the partition scheme divides the system in a partition $A$ consisting of all the elements corresponding to $y = \{1, \dots, l\}$ and a partition $B$ corresponding to all the elements corresponding to $y \in \{l+1, \dots, L\}$ analogously of the iterative partition scheme used for the one dimensional system of the last section. Choosing this partition scheme allows us to exploit a parallel implementation of the algorithm for the reduction of the bond dimension of the state. In fact, instead of considering the full $2L^2 \times 2L^2$ correlation matrix describing the state we consider the $\frac{L}{2}$ correlation matrices of dimension $2L \times 2L$. The first step of the algorithm consists in partitioning the system at step $m+1$ of the partition scheme. These corresponds to taking the first $2(m+1)$ elements of each correlation matrix. One then proceed diagonalising these subsystem. This step is analogous to step 1 of the one dimensional system presented in the last section, with the difference that now we are acting on $\frac{L}{2}$ correlation matrices simultaneously. This returns $L(m+1)$ eigenvalues from the $\frac{L}{2}$ sectors. Step 2 consists in considering all these $L(m+1)$ eingenvalues *together*, selecting the $2\frac{L}{2}$ closest to 0 or 1 and then approximating them with 0 or 1 respectively in the respective correlation matrix and setting them to product state with the rest of the system analogously to what was done in step 2 of the one dimensional case. Here the difference with the one dimensional case of the last section consists in the fact that we are setting to product state $L$ modes, not just one, and that we are choosing them from all the correlation matrices. Step 3 consists in moving to the partition scheme $m+2$, enlarging the first partitions. Differently from the one dimensional case, here one has to keep track of the number of approximations performed in each sectors before diagonalising. As in the one dimensional case, the algorithm then proceed iteratively returning after at most $L$ steps, the correlation matrix $\Gamma(m)$ of a state with reduced bond dimension with the respect to the partitions along the chosen spatial direction. Considering larger values of $m$, the correlation matrix $\Gamma(m)$ converges towards $\Gamma$, the exact correlation matrix.

> **F_utilities 6.3: RBD_csectors($\vec{\Gamma}$, $L_x$, $L_y$, $m$)$\rightarrow$ $\vec{\Gamma}(m)$**
>
> This function returns the correlation matrices $\vec{\Gamma}(m)$ obtained reducing the bond dimension to $mL$ of the system on the $L_x \times L_y$ cylinder described by the correlation matrices $\vec{\Gamma}$. Each correlation matrix must contains information of two values of $k$ and must be organised following the order $\vec{\alpha} = (a_{k,1}, a_{k+\frac{L}{2},1}, a_{k,2}, a_{k+\frac{L}{2},2}, \dots)$

# 7 Conclusions

These notes are meant for the reader interested in implementing simulations and benchmarks involving the manipulations of fermionic Gaussian states. Starting from the basic definitions we introduced a set of essential and advanced concepts and techniques together with numerical examples using the library `F_utilities`. The shared code is intended both as a tool for the reader to quickly implement simple algorithms and as a technical guideline for the writing of new code that can be tailored on particular necessities and on specific languages. In the first section after the introduction we define the basic properties of fermionic Quadratic Hamiltonians. We present the Dirac and Majorana representations and we show how fermionic Quadratic Hamiltonians can be efficiently diagonalised. In the second section after the introduction we present how fermionic Gaussian states are efficiently represented and manipu-

lated. This section includes many technical details as, for example, the relation between the eigenvalues of correlation matrix in Majorana and Dirac representation and the eigenvalues in the respective density matrix representation of the state, and the expression of different information measures directly on the correlation matrices. In the third and fourth sections after the introduction we study two specific Hamiltonian, the hopping Hamiltonian and the Transverse Field Ising (TFI) Hamiltonian. The TFI Hamiltonian, being a spin Hamiltonian, is a paradigmatic example of how fermionic Gaussian states can be exploited in the study complex quantum systems not originally expressed in terms of Fermions. In the last section before the conclusion we present some advanced algorithms and ideas, connecting Femionic Guassian states with matrix product states.

# Acknowledgements

**Funding information** The authors acknowledges support from the Ramón y Cajal program RYC-2016-20594, the "Plan Nacional Generación de Conocimiento" PGC2018-095862-B-C22 and and Grant CEX2019-000918-M funded by MCIN/AEI/10.13039/501100011033. This project was supported by the European Union Regional Development Fund within the ERDF Operational Program of Catalunya, Spain (project QUASICAT/QuantumCat, ref. 001-P-001644).

# A  Appendix

## A.1  Extended calculations

### A.1.1  Eigenvalues of $\Gamma$ and $H_\alpha$

We consider the state $\rho = \frac{e^{-\vec{a}^\dagger H \vec{a}}}{Z}$, we diagonalise $H$ changing the basis to $\vec{\beta} = U^\dagger \vec{\alpha}$. Thus we have

$$\rho = \frac{e^{-\vec{\beta}^\dagger H_D \vec{\beta}}}{Z} = \frac{e^{-\sum_{k=1}^N \epsilon_k \left( b_k^\dagger b_k - b_k b_k^\dagger \right)}}{Z} \,. \tag{158}$$

We change the basis of the correlation matrix too

$$\Gamma_{i,j}^b = \left( U^\dagger \Gamma U \right)_{i,j} = Tr \left[ \rho \vec{\beta}_i \vec{\beta}_j^\dagger \right] \,. \tag{159}$$

Now we want to explicitly compute the elements of $\Gamma^b$. First of all we compute the normalisation constant

$$Z = Tr \left[ e^{-\sum_{k=1}^N \epsilon_k \left( b_k^\dagger b_k - b_k b_k^\dagger \right)} \right] = 2^N \prod_{k=1}^N \left( \cosh \left( \epsilon_k \right) \right) \,. \tag{160}$$

To compute the numerator part this equalities will result useful

- For $x \neq y$

$$
\begin{aligned}
Tr\left[e^{-\sum_{k=1}^N \epsilon_k\left(b_k^\dagger b_k - b_k b_k^\dagger\right)} b_x^\dagger b_y\right] &= \sum_{v \in \{0,1\}^N} \langle v|e^{-\sum_{k=1}^N \epsilon_k\left(b_k^\dagger b_k - b_k b_k^\dagger\right)} b_x^\dagger b_y|v\rangle \\
&= \sum_{v \in \{0,1\}^N} \langle v|e^{-\sum_{k=1}^N \epsilon_k\left(b_k^\dagger b_k - b_k b_k^\dagger\right)}|\tilde{v}\rangle \\
&= \sum_{v \in \{0,1\}^N} e^{-\sum_{k=1}^N (-1)^{v_k+1}\epsilon_k} \langle v|\tilde{v}\rangle = 0.
\end{aligned}
\tag{161}
$$

$$
Tr\left[e^{-\sum_{k=1}^N \epsilon_k\left(b_k^\dagger b_k - b_k b_k^\dagger\right)} b_x b_y^\dagger\right] = 0.
\tag{162}
$$

- $\forall x, y$

$$
Tr\left[e^{-\sum_{k=1}^N \epsilon_k\left(b_k^\dagger b_k - b_k b_k^\dagger\right)} b_x b_y\right] = 0,
\tag{163}
$$

$$
Tr\left[e^{-\sum_{k=1}^N \epsilon_k\left(b_k^\dagger b_k - b_k b_k^\dagger\right)} b_x^\dagger b_y^\dagger\right] = 0.
\tag{164}
$$

Thus the numerator can be explicitly written as

$$
\begin{aligned}
Tr&\left[e^{-\sum_{k=1}^N \epsilon_k\left(b_k^\dagger b_k - b_k b_k^\dagger\right)} \vec{\alpha}_i \vec{\alpha}_j^{\;\dagger}\right] = \\
&= \sum_{l=1}^{2N} \sum_{m=1}^{2N} U_{i,l} U_{m,j}^\dagger Tr\left[e^{-\sum_{k=1}^N \epsilon_k\left(b_k^\dagger b_k - b_k b_k^\dagger\right)} \vec{\beta}_l \vec{\beta}_m^{\;\dagger}\right] \\
&= \sum_{l=1}^{N} U_{i,l} U_{l,j}^\dagger Tr\left[e^{-\sum_{k=1}^N \epsilon_k\left(b_k^\dagger b_k - b_k b_k^\dagger\right)} b_l^\dagger b_l\right] + \sum_{l=1}^{N} U_{i,l+N} U_{l+N,j}^\dagger Tr\left[e^{-\sum_{k=1}^N \epsilon_k\left(b_k^\dagger b_k - b_k b_k^\dagger\right)} b_l b_l^\dagger\right] \\
&= \sum_{l=1}^{N} U_{i,l} U_{l,j}^\dagger e^{-\epsilon_l} \prod_{k \neq l} 2\cosh(\epsilon_k)) + \sum_{l=1}^{N} U_{i,l+N} U_{l+N,j}^\dagger e^{\epsilon_l} \prod_{k \neq l} 2\cosh(\epsilon_k)).
\end{aligned}
\tag{165}
$$

I can divide by Z and obtain

$$
\begin{aligned}
\Gamma_{i,j} &= \sum_{l=1}^{N} U_{i,l} U_{l,j}^\dagger \frac{e^{-\epsilon_l}}{e^{\epsilon_l} + e^{-\epsilon_l}} + \sum_{l=1}^{N} U_{i,l+N} U_{l+N,j}^\dagger \frac{e^{\epsilon_l}}{e^{\epsilon_l} + e^{-\epsilon_l}} \\
&= \sum_{l=1}^{N} U_{i,l} U_{l,j}^\dagger \frac{1}{1 + e^{2\epsilon_l}} + \sum_{l=1}^{N} U_{i,l+N} U_{l+N,j}^\dagger \frac{1}{1 + e^{-2\epsilon_l}} \\
&= (U\Gamma^D U^\dagger)_{i,j}.
\end{aligned}
\tag{166}
$$

So the same transformation U that moves to the free Hamiltonian $H_D$ is also the transformation that diagonalise the correlation matrix. The eigenvalues $\nu_i$ of the correlation matrix $\Gamma$ are related to the eigenvalues of the parent Hamiltonian $H$ by

$$
\nu_i = \frac{1}{1 + e^{2\epsilon_i}},
\tag{167}
$$

$$
\epsilon_i = \frac{1}{2} \ln\left(\frac{1 - \nu_i}{\nu_i}\right),
\tag{168}
$$

since $\nu_i \in [0,1]$ the eigenvalues $\epsilon_i \in [-\infty, +\infty]$.

### A.1.2 Purity

From the previous paragraph we have:

$$Z^2 = \prod_{k=1}^{N} (2\cosh(\epsilon_k))^2 \tag{169}$$

and

$$Tr\left[e^{-\sum_{k=1}^{N} \epsilon_k \left(b_k^\dagger b_k - b_k b_k^\dagger\right)}\right] = \prod_{k=1}^{N} (2\cosh(2\epsilon_k)) \, . \tag{170}$$

Thus the purity is:

$$\text{Purity} = \prod_{k=1}^{N} \frac{1}{\text{sech}(\epsilon_k) + 1} \, . \tag{171}$$

### A.1.3 Real Time Evolution

We want to compute the time evolution in the Heisenberg picture of the annihilation operator $b_k$ induced by the Hamiltonian $\hat{H} = \sum_{l=1}^{N} \epsilon_l (b_l^\dagger b_l - b_l b_l^\dagger)$. First we simplify the expression exploiting the commuting terms

$$b_k(t) = e^{i\hat{H}t} b_k e^{-i\hat{H}t} = e^{it\sum_{l=1}^{N} \epsilon_l (b_l^\dagger b_l - b_l b_l^\dagger)} b_k e^{it\sum_{l=1}^{N} \epsilon_l (b_k^\dagger b_l - b_l b_l^\dagger)} \tag{172}$$

$$= e^{it\epsilon_k (b_k^\dagger b_k - b_k b_k^\dagger)} b_k e^{it\epsilon_k (b_k^\dagger b_k - b_k b_k^\dagger)} \, . \tag{173}$$

Secondly applying B.C.H.1 (see B.C.H.1 in A.3.4) we obtain that

$$b_k(t) = \sum_{n=0}^{\infty} \frac{(ie_k t)^n}{n!} \underbrace{[b_k^\dagger b_k - b_k b_k^\dagger, \dots [v_k^\dagger b_k - b_k b_k^\dagger}_{n}, b_k \underbrace{]\dots]}_{} n \, , \tag{174}$$

and using the fact that

$$[b_k^\dagger b_k - b_k b_k^\dagger, b_k] = -2b_k \, , \tag{175}$$

we obtain

$$b_k(t) = \sum_{n=0}^{\infty} \frac{(2ie_k t)^n}{n!} b_k = e^{-i2e_k t} b_k \, . \tag{176}$$

### A.1.4 Circulant Matrices

An $N \times N$ circulant matrix $C$ is a matrix of the form

$$C = \begin{pmatrix} c_0 & c_1 & c_2 & \dots & c_{N-1} \\ c_{N-1} & c_0 & c_1 & \dots & c_{N-2} \\ c_{N-2} & c_{N-1} & c_0 & \dots & c_{N-3} \\ \vdots & \vdots & \vdots & \ddots & \vdots \\ c_1 & c_2 & c_3 & \dots & c_0 \end{pmatrix} \, . \tag{177}$$

A circulant matrix is completely specified by the *circulant vector* $\vec{c}$, that is its first row.

$$\vec{c} = (c_0, c_1, c_2, \dots, c_{N-1}) \, . \tag{178}$$

All the other rows of the matrix are cyclic permutations of $\vec{c}$ with offset increasing by one going down with the rows.

Since each descending diagonal from left to right is constant, circulant matrices are a special case of Toeplitz matrices.

Because of their special structure, circulant matrices are diagonalised by taking their Fourier transform.

Given a vector $\vec{v}$ of length $N$ its Fourier transform is expressed as $\vec{w} = W\vec{v}$, with $W$ defined as

$$W = \frac{1}{\sqrt{N}} \begin{pmatrix} \omega & \omega^2 & \omega^3 & \cdots & \omega^{N-1} & 1 \\ \omega^2 & \omega^4 & \omega^6 & \cdots & \omega^{2(N-1)} & 1 \\ \omega^3 & \omega^6 & \omega^9 & \cdots & \omega^{3(N-1)} & 1 \\ \vdots & \vdots & \vdots & \ddots & \vdots & 1 \\ \omega^{N-1} & \omega^{2(N-1)} & \omega^{3(N-1)} & \cdots & \omega^{(N-1)(N-1)} & 1 \\ 1 & 1 & 1 & 1 & \cdots & 1 \end{pmatrix}, \tag{179}$$

with $\omega = e^{-i\frac{2\pi}{N}}$.

The columns of $W$ are the normalised eigenvectors $|\lambda_i\rangle$ of every circulant matrix of dimension $N \times N$.

The corresponding eigenvalues depend on the specific circulant vector $\vec{c}$ specifying the circulant matrix and are given by

$$\lambda_j = c_0\omega^j + c_1\omega^j + c_2\omega^{j2} + \cdots + c_{N-2}\omega^{j(N-2)} + c_{N-1}\omega^{j(N-1)}. \tag{180}$$

### A.1.5  Block diagonal form of skew-symmetric matrices

Let $h$ be a $N \times N$ skew-symmetric matrix of rank $2m$, where $N \geq 2m$.

Then there exist a $N \times N$ unitary matrix $U$ such that [26]

$$U^T h U = \begin{pmatrix} 0 & \lambda_1 \\ -\lambda_1 & 0 \end{pmatrix} \oplus \begin{pmatrix} 0 & \lambda_2 \\ -\lambda_2 & 0 \end{pmatrix} \oplus \cdots \oplus \begin{pmatrix} 0 & \lambda_m \\ -\lambda_m & 0 \end{pmatrix} \oplus \hat{0}_{N-2m}, \tag{181}$$

where $\hat{0}_{N-2m}$ is a $(N-2m) \times (N-2m)$ matrix with all elements equal to zero and where the real and positive-definite $\{\lambda_i\}_{i=1,m}$ are the singular values of $h$.

Since a skew-symmetric matrix $h$ is similar to its own transpose $h^T$, then $h$ and $h^T$ must have the same eigenvalues. Thus, the eigenvalues of a skew-symmetric matrix of even dimension will always come in pairs $\pm\tilde{\lambda}$ (for the case of odd dimension there will be an unpaired eigenvalue equal to 0).

### A.1.6  Power method algorithm

The power method algorithm is based on the following idea. Suppose we want to find the biggest eigenvalue $\lambda_1$ and its associated eigenvector $|\lambda_1\rangle$ of a diagonalisable matrix $H$. We will consider the eigenvalues of $H$ to be ordered as $\lambda_1 > \lambda_2 \geq \lambda_3 \geq \ldots$. We start by choosing a random vector $|v^{[0]}\rangle$. We define the iterative algorithm

$$|v^{[n+1]}\rangle = \frac{H|v^{[n]}\rangle}{\left\||H|v^{[n]}\rangle\right\|}, \tag{182}$$

where $\||v\rangle\|$ is the norm of $|v\rangle$. Starting with $|v^{[0]}\rangle$, we expect that, if $\langle\lambda_1|v^{[0]}\rangle \neq 0$ and $\lambda_1$ is not degenerate, for $n$ sufficiently big, $|v^{[n]}\rangle \sim |\lambda_1\rangle$. The fact that this algorithm converges towards $|\lambda_1\rangle$ can be easily proved by expanding $|v^{[0]}\rangle$ on the eigenbasis $\{|\lambda_i\rangle\}_i$ of $H$

$$|v^{[0]}\rangle = c_1|\lambda_1\rangle + c_2|\lambda_2\rangle + \ldots, \tag{183}$$

with $c_i = \langle \lambda_i | v^{[0]} \rangle$ and thus $c_1 \neq 0$ because of the assumption $\langle \lambda_1 | v^{[0]} \rangle \neq 0$. Now applying $H$ to $|v^{[0]}\rangle$ for $n$ times returns

$$H^n |v^{[0]}\rangle = c_1 \lambda_1^n \left( |\lambda_1\rangle + \frac{c_2}{c_1}(\frac{\lambda_2}{\lambda_1})^2 |\lambda_2\rangle + \dots \right). \tag{184}$$

Since $\lambda_1$ is the biggest eigenvalue we have that $(\frac{\lambda_i}{\lambda_1})^n \to 0$ with $n \to \infty$ for all $i \neq 1$. Because of this, we obtain that in the limit for $n \to \infty$, taking care of the normalisation, $H^n |v^{[0]}\rangle \to |\lambda_1\rangle$.

The convergence of this method is slow (it is geometric with ratio $\left|\frac{\lambda_2}{\lambda_1}\right|$) and it becomes slower as $\lambda_2 \to \lambda_1$.

We note here the importance of the value of the difference $|\lambda_1 - \lambda_2|$.

In the field of condensed matter, one is often interested in computing the ground state energy $E_0$ of a Hamiltonian $H$, that is the smallest eigenvalue of $H$. By adding a sufficiently big number to the Hamiltonian, one obtains that the smallest eigenvalue of $H$ corresponds to the eigenvalue with the smallest magnitude. In order to compute the smallest eigenvalue in magnitude of $H$ one can use the inverse power method [51] that fundamentally is the power method applied to $H^{-1}$. In this case the algorithm will converge geometrically with ratio $\frac{E_0}{E_1}$, where $E_1$ is the second smallest eigenvalue of the Hamiltonian $H$. If $E_1 - E_0 = 0$ then the algorithm will not converge.

Because of its importance, the difference between the two lowest eigenvalues of an Hamiltonian (that is the difference between the ground state energy and the first excited state energy) has a specific name and it is called *Hamiltonian Gap* or *spectral gap* often denoted by $\Delta E$. In particular, definining a family of Hamiltonians dependent on the parameter $N$ (the dimension of the system), we call *gapless Hamiltonians* those Hamiltonians for wich the Hamiltonian Gap $\to 0$ in the thermodynamics limit $N \to \infty$, and we call *gapped Hamiltonians* those Hamiltonians for which the spectral gap remains positive in the thermodynamic limit.

### A.1.7 Equilibration of fermionic Gaussian systems

**Equilibration times and Gaussification**   For systems evolving with a quadratic fermionic Hamiltonian there exists general, and mathematically rigorous statements about the equilibration of the systems towards the GGE [52, 53].

The framework in which these theorems hold is that of a generic $1D$ fermionic system of $N$ sites with translational invariant local Hamiltonian $H$ with periodic boundary conditions, with the additional assumption that the derivative of the dispersion relation $\epsilon(k)$ have not coinciding roots (there is not a $k$ such that $\frac{d^2}{dk^2}\epsilon(k) = \frac{d^3}{dk^3}\epsilon(k) = 0$). In this context, for every initial state $\rho$ of the system with finite correlation lenght and no long-wavelength dislocations in the two points correlators of Dirac operators , there exists a constant relaxation time $t_0$ and a time of recurrence $t_R$ proportional to the system size $N$ such that, for all $t \in [t_0, t_R]$, the state locally equilibrate to a GGE with

$$|\langle O \rangle_{\rho(t)} - \langle O \rangle_{GGE}| \leq C t^{-\gamma}, \tag{185}$$

with $O$ a local observable and $C, \gamma > 0$ independent from $N$.

It is important to notice that no assumptions on the Gaussianity of the initial state have been made. It is indeed possible to choose as initial state a state that is not Gaussian, it is the quadratic form of the Hamiltonian $H$ that, through a process called *gaussification* [52], transforms the state to a state locally indistinguishable from a fermionic Gaussian state.

Gaussification is a general result conferring even more relevance to fermionic Gaussian states. Following again [53] we have that for an initial fermionic state $\rho$ with exponential decay of correlations and a non-interacting translational invariant Hamiltonian $H$ with the derivative of the

dispersion relation with not coinciding roots (there is not a $k$ such that $\frac{d^2}{dk^2}\epsilon(k) = \frac{d^3}{dk^3}\epsilon(k) = 0$), there exists a constant relaxation time $t_0$ and and a recurrence time $t_R$ proportional to $N$ such that, for all $t \in [t_0, t_R]$,

$$|\langle O \rangle_{\rho(t)} - \langle O \rangle_{\rho(\Gamma(\rho(t)))}| \leq C t^{-1/6}, \tag{186}$$

where $C > 0$. This shows that, under these conditions, the expectation value of the local observable $O$ converges with a power law in time towards the same value computed with the Gaussian projection of the state.

**Equilibration of occupations in the fermionic transverve Field Ising model**   In some cases it is possible to explicitly compute the equilibration value of some local observables.
In chapter 5.3 we compute the time evolution of the single site occupation $\langle a_1^\dagger a_1 \rangle$ during the out-of-equilibrium dynamics of a translational invariant state with Hamiltonian (122) where $g_F = -1$ and $N$ is even.
We are now interested to verify if this observable equilibrates.
In order to avoid recurrence effects we compute the limit of expression (143) in the case of the number of sites going to infinity $N \to \infty$. Defining the quantity $p = -\frac{2\pi}{N}$ we can write

$$\langle a_1^\dagger a_1 \rangle(t) =$$

$$= \lim_{N \to \infty} \frac{1}{N} \sum_{k=\frac{-N+1}{2}}^{\frac{N+1}{2}} \left[ s_k^2 \langle b_k^\dagger b_k \rangle + t_k^2 \langle b_{-k} b_{-k}^\dagger \rangle + i s_k t_k (e^{i\epsilon_k t} \langle b_{-k} b_k \rangle - e^{-i\epsilon_k t} \langle b_k^\dagger b_{-k}^\dagger \rangle) \right]$$

$$= -\int_{-\pi}^{\pi} dp \left[ s_{-p}^2 \langle b_{-p}^\dagger b_{-p} \rangle + t_{-p}^2 \langle b_p b_p^\dagger \rangle + i s_{-p} t_{-p} (e^{i\epsilon(p)t} \langle b_p b_{-p} \rangle - e^{-i\epsilon(p)t} \langle b_{-p}^\dagger b_p^\dagger \rangle) \right], \tag{187}$$

with

$$s_p = \frac{\sin(p)}{\sqrt{\epsilon_k(\epsilon_p/2 + \cot(\theta) + \cos(p))}},$$

$$t_p = \frac{\epsilon_p/2 + \cot(\theta) + cos(p)}{\sqrt{\epsilon_p(\epsilon_p/2 + \cot(\theta) + cos(p))}},$$

$$\epsilon(p) = 2\sqrt{1 + \cot(\theta)^2 - 2\cot(\theta)\cos(p)}. \tag{188}$$

The two time-dependent terms of the integral (187) have the same form, thus it suffices to study the long-time behaviour of the integral

$$I(t) = \int_{-\pi}^{\pi} dp\, e^{i\epsilon(p)t} s_{-p} t_{-p} \langle b_p b_{-p} \rangle. \tag{189}$$

In order to study the long-time beaviour of $I(t)$ we use the result about oscillatory integral in [54] chapter VII proposition 3. Having that $\left.\frac{d}{dp}\epsilon(p)\right|_{p=0} = 0$ and $\left.\frac{d^2}{dp^2}\epsilon(p)\right|_{p=0} \neq 0$, for large values of $t$, the integral can be approximated as

$$I(t) \sim t^{-1/2} \sum_{j=0}^{\infty} a_{2j} t^{-j}, \tag{190}$$

where each $a_j$ depends only on finitely many derivatives of both $\epsilon(p)$ and $s_{-p} t_{-p} \langle b_p b_{-p} \rangle$ at $p = 0$. Computing $a_j$ explicitly we find that $a_0 = 0$, thus we have that at large $t$

$$I(t) \sim t^{-3/2}. \tag{191}$$

Plugging this result into (187) we find out that the single site occupation, in the long-time regime, equilibrates as $t^{-3/2}$ to the asymptotic value of

$$\langle a_1^\dagger a_1 \rangle = -\int_{-\pi}^{\pi} dp \left[ s_{-p}^2 \langle b_{-p}^\dagger b_{-p} \rangle + t_{-p}^2 \langle b_p b_p^\dagger \rangle \right], \tag{192}$$

that is exactly the one predicted by the GGE.

### A.1.8 Jordan-Wigner transformation

The Jordan-Wigner transformation, introduced in the original paper [55], is a transformation that maps spin-$\frac{1}{2}$ systems to fermionic systems.

Suppose we have a system of $N$ spins-$\frac{1}{2}$ with the usual Pauli matrices $\sigma_j^x$, $\sigma_j^y$ and $\sigma_j^z$ acting on the $j$-th spin of the system. The Jordan-Wigner transformation defines the operator $a_j$ as

$$a_j = -\left( \otimes_{k=1}^{j-1} \sigma_k^z \right) \otimes \sigma_j^+ \left( \otimes_{k=j+1}^{N} \mathbb{I}_k \right), \tag{193}$$

where $\sigma_j^\pm = \frac{\sigma_j^x \pm i\sigma_j^y}{2}$ and $\mathbb{I}_j$ is the identity acting on the $j$-th spin. Taking the adjoint obtains

$$a_j^\dagger = -\left( \otimes_{k=1}^{j-1} \sigma_k^z \right) \otimes \sigma_j^- \left( \otimes_{k=j+1}^{N} \mathbb{I}_k \right). \tag{194}$$

Computing the anticommutator of these two operators we notice that they obey the CAR, thus using this transformation for every site $j$ we are able to build a legitimate set of Dirac fermionic operators starting from a set of Pauli matrices.

Knowing the expression for the creation and annihilation operators, we can easily find the mapping of the single site occupation operator in terms of Pauli operators:

$$a_j^\dagger a_j = \left( \otimes_{k=1}^{j-1} \mathbb{I}_k \right) \otimes \frac{\mathbb{I}_j - \sigma_j^z}{2} \left( \otimes_{k=j+1}^{N} \mathbb{I}_k \right). \tag{195}$$

Finally there are two important remarks. We notice that the mapping from spins to fermions is not local, in the sense that equation 193 maps a string of Pauli operators acting non trivially on $j$ spins to a Dirac operator local only on site $j$.

We even notice that in the definition of the annihilation operator 193 it is encoded some information on the geometrical structure of the spin system, in particular it is encoded the distance of site $j$ from the border.

When using the Jordan-Wigner transformation one has to be careful about these two observations.

In the main text we are interested in mapping the transverse field Ising Hamiltonian to a fermionic system, thus we need the inverse Jordan-Wigner transformation. We have that the Pauli operator $\sigma_j^z$ is easily mapped to fermionic annihilation and creation operators as

$$\sigma_j^z = a_j a_j^\dagger - a_j^\dagger a_j = 1 - 2a_j^\dagger a_j. \tag{196}$$

We see that for this transformation local spin operators are mapped to local fermionic operators. We know nonetheless that the Jordan-Wigner transformation does not preserve locality in general, indeed we have that the Pauli operators $\sigma_j^x$ and $\sigma_j^y$ maps to fermionic operators as

$$\sigma_j^x = -\left( \otimes_{k=1}^{j-1} \sigma_k^z \right) \otimes (a_j + a_j^\dagger) \left( \otimes_{k=j+1}^{N} \mathbb{I}_k \right),$$
$$\sigma_j^y = i \left( \otimes_{k=1}^{j-1} \sigma_k^z \right) \otimes (a_j^\dagger - a_j) \left( \otimes_{k=j+1}^{N} \mathbb{I}_k \right), \tag{197}$$

where for each $\sigma_k^z$ one should use the substitution (196).

Fortunately, if we consider the product of Pauli operators, as for example are the spin-spin interactions in the TFI model we have

$$\begin{aligned}
\sigma_j^x \sigma_{j+1}^x &= (a_j^\dagger - a_j)(a_{j+1} + a_{j+1}^\dagger), \\
\sigma_j^y \sigma_{j+1}^y &= -(a_j^\dagger + a_j)(a_{j+1}^\dagger - a_{j+1}), \\
\sigma_j^x \sigma_{j+1}^y &= i(a_j^\dagger - a_j)(a_{j+1}^\dagger - a_{j+1}), \\
\sigma_j^y \sigma_{j+1}^x &= i(a_j^\dagger + a_j)(a_{j+1}^\dagger + a_{j+1}),
\end{aligned} \tag{198}$$

nearest neighbour interactions are mapped to nearest neighbour interactions.

It easy to see that an interaction of this kind between two arbitrary spins at site $j$ and $k$ will map to a string of Dirac operators acting non trivially on all sites between $j$ and $k$.

We have seen that the Jordan-Wigner transformation defines an isomorphism from a system of $n$ fermions to a system of $n$ spins. One should ask why we cannot completely identify spin systems with fermionic systems or vice versa. To answer to this question we remind that, as specified above, the Jordan-Wigner mapping does not preserve the locality. One of the consequence of this fact is that the procedure of partial tracing does not generally commute with the Jordan-Wigner mapping [27, 28]. Consider for example a state of $N$ fermions $\rho_{AB}$ defined on a system divided in two complementary partitions $A$ and $B$. We map $\rho_{AB}$ with a Jordan-Wigner transformation to a state $\tilde{\rho}_{AB}$ of $N$ spins. Now we consider the reduced states $\rho_A$ and $\tilde{\rho}_A$ on partition $A$ of the states $\rho_{AB}$ and $\tilde{\rho}_{AB}$. If, using a Jordan-Wigner transformation, we map the state $\rho_A$ to the spin state $\tilde{\tilde{\rho}}_A$, we will generally have that $\tilde{\rho}_A \neq \tilde{\tilde{\rho}}_A$ as shown schematically in figure 16.

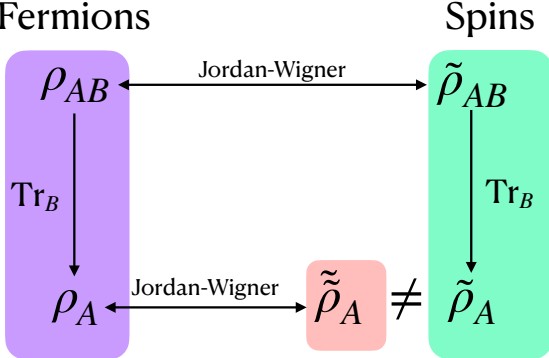

Figure 16: The mapping of the reduced state is different from the reduced state of the mapping [28]

For a detailed and very well explained treatment of this question see [28]. We end this subsection pointing out that, the fact that the well defined trace for fermionic system is not consistent with the mapping between fermions and qubits, leads to many interesting questions on entanglement in fermionic systems, see e.g. [56–61]

## A.2 Matrix product states

Let us consider an $N$ constituents systems and a vector in its Hilbert space space $\mathcal{H} = (\mathbb{C}^d)^{\otimes N}$

$$|\psi\rangle = \sum_{i_1,\dots,i_N} c_{i_1,\dots,i_N} |i_1,\dots,i_N\rangle, \tag{199}$$

where $|i_1,\ldots,i_N\rangle = |i_1\rangle \otimes \cdots \otimes |i_N\rangle$ and for each $l = 1,\ldots,N$ $_l \in \{1,\ldots,d\}$. In total we have $d^N$ different coefficients. The coefficient tensor $c_{i_1,\ldots,i_N}$ can be rewritten as

$$c_{i_1,\ldots,i_N} = A^{[1]}_{i_1} A^{[2]}_{i_2} \ldots A^{[N]}_{i_N}, \tag{200}$$

where for $l \neq 1,N$, $A^{[l]}_i$ is a $D \times D$ matrix, $A^{[1]}_i \in \mathbb{C}^{1 \times D}$ is a row vector, and $A^{[N]}_i \in \mathbb{C}^{D \times 1}$ is a column vector. We encoded all the coefficients of the state in $NdD^2$. The state can be then written as:

$$|\psi\rangle = \sum_{i_1,\ldots,i_N} A^{[1]}_{i_1} A^{[2]}_{i_2} \ldots A^{[N]}_{i_N} |i_1,\ldots,i_N\rangle. \tag{201}$$

Equation (201) is called Matrix Product State (MPS) representation of the state [62–66]. The dimension $D$ of the matrices is called *bond dimension* of the state. The amount of

## A.3 Useful relations

### A.3.1 Pauli Matrices

1. $\sigma^+ = \begin{pmatrix} 0 & 1 \\ 0 & 0 \end{pmatrix}, \quad \sigma^- = \begin{pmatrix} 0 & 0 \\ 1 & 0 \end{pmatrix}, \quad \sigma^z = \begin{pmatrix} 1 & 0 \\ 0 & -1 \end{pmatrix}, \quad \sigma^y = \begin{pmatrix} 0 & -i \\ i & 0 \end{pmatrix}, \quad \sigma^x = \begin{pmatrix} 0 & 1 \\ 1 & 0 \end{pmatrix},$

$|+\rangle_x = \frac{1}{\sqrt{2}} \begin{pmatrix} 1 \\ 1 \end{pmatrix}, \quad |-\rangle_x = \frac{1}{\sqrt{2}} \begin{pmatrix} 1 \\ -1 \end{pmatrix}, \quad |+\rangle_y = \frac{1}{\sqrt{2}} \begin{pmatrix} 1 \\ i \end{pmatrix}, \quad |-\rangle_y = \frac{1}{\sqrt{2}} \begin{pmatrix} 1 \\ -i \end{pmatrix}, \quad |0_-\rangle_z = \begin{pmatrix} 0 \\ 1 \end{pmatrix},$

$|1_+\rangle_z = \begin{pmatrix} 1 \\ 0 \end{pmatrix},$

2. $\sigma^z \sigma^- = -\sigma^-$,

3. $\sigma^z \sigma^+ = \sigma^+$,

4. $\sigma^- \sigma^z = \sigma^-$,

5. $\sigma^+ \sigma^z = -\sigma^+$,

6. $\sigma^+ \sigma^- = \frac{\sigma^z + \mathbb{I}}{2}$,

7. $\sigma^- \sigma^+ = \frac{\mathbb{I} - \sigma^z}{2}$.

### A.3.2 Operators obeying CAR

1. $\{a_i, a^\dagger_j\} = \mathbb{I}\delta_{i,j}, \qquad \{a_i, a_j\} = \{a^\dagger_i, a^\dagger_j\} = 0$,

2. $a_i a_j = -a_j a_i; \qquad a^\dagger_i a^\dagger_j = -a^\dagger_j a^\dagger_i$,

3. $a^2_i = \left(a^\dagger_j\right)^2 = 0$,

4. $a_i a^\dagger_j = \delta_{i,j} - a^\dagger_j a_i$,

5. $a_i a_j = \frac{a_i a_j - a_j a_i}{2}$,

6. $a_i a^\dagger_j = \frac{a_i a^\dagger_j - a^\dagger_j a_i}{2} + \frac{\delta_{i,j}}{2}$,

7. $a^\dagger_i a_j = \frac{a^\dagger_i a_j - a_j a^\dagger_i}{2} + \frac{\delta_{i,j}}{2}$.

Commutators

1. $[a_i^\dagger, a_j] = \delta_{i,j} - 2a_j a_i^\dagger = a_i^\dagger a_j - \delta_{i,j}$ ,

2. $[a_i, a_j^\dagger] = \delta_{i,j} - 2a_j^\dagger a_i = a_i a_j^\dagger - \delta_{i,j}$ ,

3. $[a_i, a_j] = 2a_i a_j$ ,

4. $[a_i^\dagger, a_j^\dagger] = 2a_i^\dagger a_j^\dagger$ .

Majorana operators

1. $x_i^2 = p_i^2 = \frac{1}{2}$ ,

2. $a^\dagger a = \frac{i}{2}(xp - px) + \frac{1}{2} = ixp + \frac{1}{2}$ ,

3. $aa^\dagger = \frac{i}{2}(px - xp) + \frac{1}{2} = ipx + \frac{1}{2}$ ,

4. $xp = -\frac{i}{2}(a^\dagger a - aa^\dagger) = -i(a^\dagger a - \frac{1}{2})$ .

### A.3.3 Jordan-Wigner Transformations

**spinless fermions → spins**

1. $a_j = -\bigotimes_{k=1}^{j-1} \sigma_k^z \otimes \sigma_j^- \bigotimes_{k=j+1}^{N} \mathbb{I}_k$ ,

2. $a_j^\dagger = -\bigotimes_{k=1}^{j-1} \sigma_k^z \otimes \sigma_j^+ \bigotimes_{k=j+1}^{N} \mathbb{I}_k$ ,

3. $a_j^\dagger a_j = \bigotimes_{k=1}^{j-1} \mathbb{I}_k \otimes \frac{\sigma_j^z + \mathbb{I}_j}{2} \bigotimes_{k=j+1}^{N} \mathbb{I}_k$ .

**spins → spinless fermions**

1. $\sigma_j^z = a_j a_j^\dagger - a_j^\dagger a_j$ ,

2. $\sigma_j^x = -\bigotimes_{k=1}^{j-1} \sigma_j^z \otimes (a_j + a_j^\dagger) \bigotimes_{k=j+1}^{N} \mathbb{I}_j$ ,

3. $\sigma_j^y = i \bigotimes_{k=1}^{j-1} \sigma_j^z \otimes (a_j^\dagger - a_j) \bigotimes_{k=j+1}^{N} \mathbb{I}_j$ ,

4. $\sigma_j^x \sigma_{j+1}^x = (a_j^\dagger - a_j)(a_{j+1} + a_{j+1}^\dagger)$ ,

5. $\sigma_j^y \sigma_{j+1}^y = -(a_j^\dagger + a_j)(a_{j+1}^\dagger - a_{j+1})$ ,

6. $\sigma_j^x \sigma_{j+1}^y = i(a_j^\dagger - a_j)(a_{j+1}^\dagger - a_{j+1})$ ,

7. $\sigma_j^y \sigma_{j+1}^x = i(a_j^\dagger + a_j)(a_{j+1}^\dagger + a_{j+1})$ .

### A.3.4 Formulas

1. B.C.H. 1: $e^A e^B = e^Z$
   with $Z = A + B + \frac{1}{2}[A, B] + \frac{1}{12}[A, [A, B]] + \frac{1}{12}[B, [A, B]] + \ldots$ higher commutators of A and B

2. B.C.H 2: $e^A B e^{-A} = \sum_{n=0}^{\infty} \frac{1}{n!} \underbrace{[A, \ldots [A}_{n}, B \underbrace{] \ldots]}_{n}$ where $[A, B] = AB - BA$.

3. B.C.H 3: $e^A B e^A = \sum_{n=0}^{\infty} \frac{1}{n!} \underbrace{\{A, \ldots \{A}_{n}, B \underbrace{\} \ldots\}}_{n}$ where $\{A, B\} = AB + BA$.

4. Kronecker Delta: $\delta_{n,m} = \frac{1}{N} \sum_{k=1}^{N} e^{i\frac{2\pi}{N}k(n-m)}$.

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
