# Peer review of "Fermionic Gaussian states: an introduction to numerical approaches"

_SciPost Physics Lecture Notes, doi:SciPost Phys. Lect. Notes 54 (2022)_

## Round 1 · Referee Report · Anonymous (Referee 1) · 2022-2-28

Strengths

-Clarity of the explanations
-Developed an open-source package for the numerical manipulation of Gaussian fermionic states in 1D and 2D of easy usage
-Excellent introduction of the F_utilities functions in the main text.

Weaknesses

I do not see particular weaknesses.

Report

In these Lecture notes the authors provide a clear and pedagogical review on the properties of Gaussian fermionic systems, mainly in one spatial dimension.

After a careful discussion on the basic properties of fermionic systems, the authors focus on Gaussian states, for which they list several well-known properties and introduce many useful functions of the Julia package F_utilities that they developed. In these first sections, I really appreciated the efforts that the authors made to provide a very detailed discussion in a relatively concise number of pages. Afterwards, they focus on some cases of particular interest (hopping models and transverse field Ising chain).
Section 6 aims to connect the techniques available for free fermions to algorithms with MPS. While I have appreciated 6.1 and 6.2, I have found sec. 6.3 and 6.4 very different from the rest of these notes : in my opinion these subsections appear much less pedagogical and in some parts even a bit confusing for the reader (see the detailed comments below).

As a general comment, I think that this manuscript is a valuable and interesting guide on the topic. It fully reflects the authors intention (that I read in the abstract) as well as the acceptance criteria of SciPost Physics - Lecture Notes and therefore it deserves publication after some minor changes (see below).

Requested changes

1) pg 6 – Please define the Hamming distance.

2) pg 9-10 – I would move the discussion about the mapping to Dirac fermions (eq 33-35) after the discussion on the diagonalization of Majorana fermions. In my opinion, this change would improve the clarity of the discussion about the diagonalization.

3) pg 10 - I would move the box of F_utilities 2.1 after point 3 (at the bottom of pg 10): indeed, at that stage the reader does not know how to get $U$ and $H_D$ yet.

4) pg 12 – Could it be useful to define the Pfaffian for the unexpert reader?

5) pg 17, after eq.(68) it could be useful to write down the case $N=2$ explicitly.

6) F_utilities 3.8 – The explanation of the function Inject_gamma is clear from the illustration but the variable $i$ is not commented in the dedicated box.

7) Figure 3: according to the authors convention in eq.(22), the plot of the first $N$ eigenvalues has y-label $-\epsilon_k$ instead of $\epsilon_k$, please correct.

8) pg 10, code frame : the function Fu.Build_hopping_hamiltonian(N,PBC=true) is commented only later in the text (end of pg 31). I have noticed a similar issue other times in these notes. I do understand that the boxes of F_utilities are placed in convenient locations (or automatically?) but sometimes these are found really far away from their comment in the main text (and without tags to them in in the main text) making the read a bit difficult. If possible, I would ask the authors to insert some tags in the main text in reference to the boxes of F_utilities or at least to improve the positions of some of them in the main text.

9) pg 32 – before the coding frame it would be useful to write explicitly which $\Gamma_{i,j}$ the authors consider (specified only in the code).

10) Table 1 – The authors may expand on the correspondence of the spin/fermions boundary conditions a bit more in the text/caption: PBC/APBC may be exchanged depending on the number $N$ of particles in the chain.

11) pg 39 – code frame: The authors can add more comments to help the unexpert reader to follow the lines of the code. For instance “#Numerical results for the energies with different boundary conditions” and similarly for the rest.

12) Figure 9 – It is not specified in the y-label and/or in the figure caption that the authors are plotting $\log_{10}(\Delta E)$ and not $\Delta E$. Please correct.

13) Sec 6 pg 43, what is an experimental algorithm?

14) Pg 45 -The authors write: “[…] approximate imaginary time evolution stops at some plateaux and thus it tricks the algorithm in believing in a false convergence to the ground state.” – Perhaps the authors can add a short sentence on a possible escape trick (for instance, taking a combination of the previous Trotter steps).

15) It seems to me that Figure 11 is not called in the main text.

16) pg 47 : “In figure 14 we plot [...]” I suppose it is figure 15 instead, please check.

17) pg 48: In my opinion it would be useful to write the discussion of pg 47 in a more schematic way, for instance by following the steps 1-6 of Figure 12 and 13, and perhaps add these panels at each of the steps. In any case, I would recommend to merge Figure 12 and 13 together.

18) The authors could have a problem with the tags of the figures in sec 6.3/ 6.4. In pg. 49 “In figure 15 we plot [...]” I suppose they refer to figure 16 instead. Please check. Similarly, in pg 50 “[…] along the y direction as in figure 6.4.”, I suppose it is figure 14.

19) pg 51 – I would move the box of F_utilities 6.2 in sec 6.3 where the 1D case is discussed. Also, I do not understand whether it is a desidered layout or it is due to an automatic page formatting but I do not really like that the code frame and the associated figures 15,16 are inserted at the end of the sec 6.4 and not at the end of sec 6.3 since these refer to the 1D case.

20) Some typos that I have found (please check):

pg 3 “ Fee systems constitute […]” pg 17 “To tensor product […]” F_utilities 3.14: “Generate a random f.q.h. Hamiltonian […]” pg 30 : “[…] the hopping Hamiltonian using functions of F-utilities.” pg 35 : “We notice that becuase […] ” pg 39 : “In the thermodynamic limits [...]” Figure 8 : “[...] periodic and free boundary conditions.” pg 59 : “A $N\times N$ circulant matrix $C$ is a matric [...]” pg 59: “[…] to its own transpose$h^T$” missing space pg 60: “In condensed matter one is often interested in [...]” it seems as if the authors missed a word after "condensed matter" pg 61: “In some cases it is possible to explicitly compute the equilbration [...]” pg 62: after (192) “[…] that is exactly the one predicted by the GDE .” Do the authors mean GGE? *pg 62: “the system The Jordan-Wigner” missing full stop

---

## Round 1 · Referee Report · Anonymous (Referee 2) · 2022-3-31

Strengths

  • clear and pedagogical presentation
  • numerical implementation of the routines

Weaknesses

  • presentation of sec. 6.3

Report

This lecture notes presents a very nice and pedagogical introduction
to fermionic Gaussian states, focusing on numerical approaches.
The manuscript gives a detailed overview of the basics of free-fermion
techniques supplemented by numerical examples. The lecture notes
comes with a Julia implementation of the various routines.

I believe that this material could be extremely useful for beginners
who'd like to have a kickstart in working with free-fermion systems.
The text is well written and the examples are instructive.
Hence the manuscript clearly deserves to be published.

The main shortcoming of the manuscript is the presentation of sec. 6.3.
This part shows a very sharp contrast with the rest of the manuscript
in terms of clarity and pedagogical approach. In particular, in describing
the compression method for correlation matrices, the authors would like to
stress some analogies to matrix product states. However, they do not even
define what an MPS is. Without such a definition, it is hard to start a discussion
about bond dimensions. In my opinion, this part of the manuscript definitely
needs some improvement. Other than that, I found only minor points and
typos as discussed below in detail.

Requested changes

  • Since the authors discuss entanglement extensively, it would be useful to include some previous reviews on the topic for further reference

  • On p.3 the "rainbow chain" is mentioned but no references are given.

Sec. 6.3:

  • definition of an MPS as well as corresponding bond dimension should be introduced and discussed to make the analogy with the correlation matrix compression method clear

  • Fig. 10 and 11 are not referenced in the text. The caption of Fig. 11 should say "eigenvalues that are NOT 0 or 1". The claim "entanglement saturates" is not true, it grows logarithmically.

  • on p. 46, the correlation matrix $\Lambda$ is incorrectly referenced as "$\Lambda$ is a pure state" and "let $\Lambda$ be the ground state". It is also not clear what "it makes $\nu$ also an approximate eigenvalue of $\Lambda$" means, since $\Lambda$ has a much larger dimension than $\Lambda_\ell$.

  • in the last paragraph on p. 47 the notation is inconsistent: one should either use $m+1$ or $\ell$ for the length of the block, using both is confusing

  • in Fig 12c the label m+1 looks like a matrix element, same in Fig. 13

  • in Fig. 15 the notation changes from $\Lambda$ -> $\Gamma$, which is confusing.

  • in Fig 16. a random Hamiltonian is considered, however, the result is not properly discussed. In particular, one should make clear that in such a situation one has an extensive scaling of the entropy with subsystem size, hence the logarithmic plot. In fact, it would actually be useful and more reasonable to show a plot with a larger m value.

Typos:

  • I believe the word "fermionic" should not be capitalized
  • in term of -> in terms of (various times in the manuscript)
  • Eq. (13): $i^{2N}$
  • second line of p. 8: transformation
  • eq. below (102): last index $i+\nu$ -> $i+N_A$
  • p. 31, in F_utilities box: of dimension
  • Fig. 6 last line of caption: farther
  • line above Fig. 8: $g_F=-1$
  • Fig. 9: one should mention that y axis has log-scale
  • p. 47: "the diagonal elements of $D^1_{1,\dots,m+1}$"
  • same paragraph on p. 47: $\Lambda^{(N-m+1)}$
  • bottom of p. 47: Fig. 14 should be Fig. 15
  • Fig. 12c: correlation, in caption: one obtains
  • below Fig. 13: bound dimension -> bond dimension
  • on p. 49: Fig. 15 should be Fig. 16
  • p. 50 bottom: perpendicularly
  • p. 51, in F_utilities 6.3 box: contains
  • Fig. 15 and 16 caption: analogous

---

## Round 2 · Referee Report · Anonymous (Referee 2) · 2022-4-20

Report

The authors have implemented all the recommendations, the manuscript can now be published.

Requested changes

I just noticed a typo in the new Eqs. (200) and (201), the second tensor should have index $i_2$ and superscript $[2]$.

---

## Round 2 · Referee Report · Anonymous (Referee 1) · 2022-4-20

Strengths

-Clarity of the explanations
-Developed an open-source package for the numerical manipulation of Gaussian fermionic states in 1D and 2D of easy usage
-Excellent introduction of the F_utilities functions in the main text.

Weaknesses

I do not see particular weaknesses.

Report

In the revised version of the manuscript, the authors fixed the main issues regarding sec. 6.3/6.4 and properly implemented the other small changes that I suggested. Therefore, I recommend the manuscript for publication in SciPost Physics Lecture Notes.

Requested changes

-

---

## Round 2 · Author Response

We really thanks the Referee 1 for their report and in-depth analysis of section 6.3. We implemented the suggested changes as follows:

  • Added reference http://doi.org/10.1103/RevModPhys.80.517 and references therein as a review on the topic of entanglement in many-body systems.
  • Added references https://doi.org/10.1088/1742-5468/2015/06/p06002 and https://doi.org/10.1088/1751-8121/aa6268 for the rainbow chain.

Sec.6.3:

  • Added a section on the definition of MPS in the appendix A.2 and references https://doi.org/10.1016/j.aop.2014.06.013, https://doi.org/10.1103/RevModPhys.93.045003 and https://doi.org/10.1016/j.aop.2010.09.012.
  • Fig. 10 and 11 are now referenced in the text (p.44). The captions have been corrected.
  • The text has been updated to make it more clear, in particular we corrected to ""it makes $\vec{v}$ also an approximate eigenvalue of Λ" with the sentence "This makes $\vec{v}$ also an approximate eingenvector of Λ. In fact we know that the eigenvalues of Λ are 0 or 1, thus it is sufficient to extend the dimension of the eigenvector $\vec{v}$ adding to it N − l zeros to obtain an eigenvector of Λ." Changed "Since $\Lambda$ is a pure state" with "Since $\Lambda$ corresponds to a pure state". Changed "let $\Lambda$ be the ground state" with "Let $\Lambda$ be the correlation matrix of the ground state of a 1D local Hamiltonian."
  • The notation has been uptated to make it consistent.
  • The figures have been updated to make it clear that the label $m+1$ referers to the dimension of the block.
  • The notation has been updated to make it consistent.
  • We created new plots for bigger sizes and for a bigger value of $m$ as suggested by the referee. We updated the caption making clear that in such a situation one has an extensive scaling of the entropy with subsystem size.

Typos:

The typos have been corrected.

We gratefully thanks the Referee 2 for their careful and in-depth report. We implemented the changes suggested as follows:

1) We agree with the referee. We added a footnote with the definition of Hamming distance. 2) We agree with the referee. We moved the discussion where suggested. 3) We agree with the referee. We moved the box where suggested. 4) We thought about the possibility of defining the Pfaffian in the text. We believe that the Pfaffian is not so relevant to require its cumberson definition in the main text or in the appendix. Adding the definition of Pfaffian as a footnote is not going to help much either, as in order to define the Pfaffian one also have to introduce other elements such as the Levi-Civita symbol of groups of permutations. In the explicit examples following the introduction of the Pfaffian, some simple expressions of the Pfaffian are explicitly presented. 5) We agree with the referee. We wrote down the case for $N=2$ explicitly. 6) The variable $i$ was commented at the end of the box. Nevertheless, it was not discussed explicitly as an argument of the function. For this reason we added a comment about $i$ in the first sentence of the box as requested by the referee. 7) Added the minus sign. 8) We agree with the referee, we improved the position of the boxes. 9) Before the coding frame we wrote "in the following program we numerically compute the time evolution induced by a hopping Hamiltonian on a random translational invariant gaussian state with exponentially decaying correlation functions." The $\Gamma$ represents this state. 10) We agree with the referee. We expanded the caption of the table as requested. 11) We agree with the referee. We added more comments to the code. 12) Corrected the label of the figure. 13) We removed the word "experimental". 14) Added "Possible escape tricks exists. Perhaps one can slightly perturb the state and then compute the value of the energy after another step of the evolution, this can help in escaping from local minima. In general such tricks have to be adapted to the particular needs. " as requested. 15) Figure 11 is now called in the main text. 16) We corrected the figure numbering. 17) We agree with the referee. We merged figure 12 and 13. 18) There was a general problem with the tags of the figures. We corrected it. 19) We agree with the referee. We moved the box, as well as the code frames and the associated figures in the appropiate section. 20) We corrected the typos.

---

## Round 2 · List of Changes

We really thanks the Referee 1 for their report and in-depth analysis of section 6.3. We implemented the changes suggested as follows:

  • Added reference http://doi.org/10.1103/RevModPhys.80.517 and references therein as a review on the topic of entanglement in many-body systems.
  • Added references https://doi.org/10.1088/1742-5468/2015/06/p06002 and https://doi.org/10.1088/1751-8121/aa6268 for the rainbow chain.

Sec.6.3:

  • Added a section on the definition of MPS in the appendix A.2 and references https://doi.org/10.1016/j.aop.2014.06.013, https://doi.org/10.1103/RevModPhys.93.045003 and https://doi.org/10.1016/j.aop.2010.09.012.
  • Fig. 10 and 11 are now referenced in the text (p.44). The captions have been corrected.
  • The text has been updated to make it more clear, in particular we corrected to ""it makes $\vec{v}$ also an approximate eigenvalue of Λ" with the sentence "This makes $\vec{v}$ also an approximate eingenvector of Λ. In fact we know that the eigenvalues of Λ are 0 or 1, thus it is sufficient to extend the dimension of the eigenvector $\vec{v}$ adding to it N − l zeros to obtain an eigenvector of Λ." Changed "Since $\Lambda$ is a pure state" with "Since $\Lambda$ corresponds to a pure state". Changed "let $\Lambda$ be the ground state" with "Let $\Lambda$ be the correlation matrix of the ground state of a 1D local Hamiltonian."
  • The notation has been uptated to make it consistent.
  • The figures have been updated to make it clear that the label $m+1$ referers to the dimension of the block.
  • The notation has been updated to make it consistent.
  • We created new plots for bigger sizes and for a bigger value of $m$ as suggested by the referee. We updated the caption making clear that in such a situation one has an extensive scaling of the entropy with subsystem size.

Typos:

The typos have been corrected.

We really thanks the Referee 2 for their careful and in-depth report. We implemented the changes suggested as follows:

1) We agree with the referee. We added a footnote with the definition of Hamming distance. 2) We agree with the referee. We moved the discussion where suggested. 3) We agree with the referee. We moved the box where suggested. 4) We thought about the possibility of defining the Pfaffian in the text. We believe that the Pfaffian is not so relevant to require its cumberson definition in the main text or in the appendix. Adding the definition of Pfaffian as a footnote is not going to help much either, as in order to define the Pfaffian one also have to introduce other elements such as the Levi-Civita symbol of groups of permutations. In the explicit examples following the introduction of the Pfaffian, some simple expressions of the Pfaffian are explicitly presented. 5) We agree with the referee. We wrote down the case for $N=2$ explicitly. 6) The variable $i$ was commented at the end of the box. Nevertheless, it was not discussed explicitly as an argument of the function. For this reason we added a comment about $i$ in the first sentence of the box as requested by the referee. 7) Added the minus sign. 8) We agree with the referee, we improved the position of the boxes. 9) Before the coding frame we wrote "in the following program we numerically compute the time evolution induced by a hopping Hamiltonian on a random translational invariant gaussian state with exponentially decaying correlation functions." The $\Gamma$ represents this state. 10) We agree with the referee. We expanded the caption of the table as requested. 11) We agree with the referee. We added more comments to the code. 12) Corrected the label of the figure. 13) We removed the word "experimental". 14) Added "Possible escape tricks exists. Perhaps one can slightly perturb the state and then compute the value of the energy after another step of the evolution, this can help in escaping from local minima. In general such tricks have to be adapted to the particular needs. " as requested. 15) Figure 11 is now called in the main text. 16) We corrected the figure numbering. 17) We agree with the referee. We merged figure 12 and 13. 18) There was a general problem with the tags of the figures. We corrected it. 19) We agree with the referee. We moved the box, as well as the code frames and the associated figures in the appropiate section. 20) We corrected the typos.

---

## Editorial Decision

published